# GeoEvo: Identity-Aware Potential Game with Geometric Evolution for Personalized Multimodal Federated Learning

**Chen Wang** [1]    **Yongli Hu** [1]    **Huajie Jiang** [1]    **Kan Guo** [1]    **Tengfei Liu** [1]    **Junbin Gao** [2]    **Yanfeng Sun** [1]    **Baocai Yin** [1]

## Abstract

We reconceptualize Personalized Multimodal Federated Learning (PMFL) by treating missing modalities as intrinsic structural identities that constrain each client to a distinct Riemannian submanifold, rather than as deficiencies to be compensated. To reconcile the tension between identity preservation and cross-client collaboration, we cast PMFL as an identity-aware potential game and seek a geometry-consistent equilibrium instead of a monolithic full-modality optimum. We propose GeoEvo, a federated approximate solver that combines curvature-adaptive Fisher descent with manifold-lifted evolutionary search: Natural Evolution Strategies for basin escape, and Particle Swarm updates anchored to a server-broadcast Fréchet prototype for cross-client transfer. A monotone acceptance rule drives per-step potential dissipation, yielding an $\mathcal{O}(1/\sqrt{T})$ stationarity rate and convergence toward first-order Nash equilibria in non-convex regimes. Empirically, GeoEvo improves personalization and robustness across diverse modality-missing patterns.

## 1. Introduction

Federated Learning (FL) (McMahan et al., 2017) has emerged as a promising paradigm for collaborative model training across distributed clients without exposing raw data. However, conventional FL frameworks (Li et al., 2020) predominantly operate under a restrictive unimodal assumption, neglecting the multimodal nature of real-world scenarios. To bridge this gap, Multimodal Federated Learning (MMFL) (Xiong et al., 2022) extends FL to exploit cross-modal synergies, learning richer multimodal representations while preserving strict privacy guarantees.

Existing MMFL frameworks (Sun et al., 2024b; Singha et al., 2025) presume full modality availability, treating clients as homogeneous entities that differ only in data distribution. In practice, this presupposition collapses under heterogeneous hardware, divergent privacy constraints, or asymmetric sensors. The resulting modality incompleteness shifts the core challenge from managing statistical discrepancies to coordinating a fragmented federation with structurally incompatible observation domains. Such fragmentation creates a persistent performance gap; although Personalized Federated Learning (PFL) (Tan et al., 2022a) seeks to address client-specific characteristics, it remains constrained by a shared input space. Unlike standard statistical shifts, this incompleteness introduces *structural heterogeneity*—a more profound barrier where divergent hypothesis-space dimensions render conventional personalization ill-posed.

Prevailing approaches (Yu et al., 2023; Nguyen et al., 2026) treat missing modalities as deficiencies to be rectified, relying on imputation or robust training to enforce alignment with a modality-complete reference. This compensation-centric view, however, conflates structural voids with distributional shifts, pursuing correspondence where none exists. We argue that a client's modality configuration constitutes its *intrinsic structural identity*—not a defect, but a defining characteristic constraining optimization to a distinct Riemannian submanifold. This conceptual shift motivates **Personalized Multimodal Federated Learning (PMFL)**, which realizes cross-client collaboration via geometry-aware adaptation across heterogeneous manifolds, circumventing representational distortions of forced global alignment.

This perspective exposes a coordination paradox: *how can clients on distinct manifolds collaborate without collapsing into an identity-erasing consensus?* We resolve this tension by casting PMFL as an **Identity-Aware Potential Game**, where each client optimizes on its Riemannian manifold, coupled through a shared exact potential. This formulation reduces decentralized objective conflicts to principled descent on a potential landscape, reconciling collaborative alignment with preservation of heterogeneous equilibria.

[1] Beijing Key Laboratory of Multimodal Cognitive Computing and Intelligent Software Technology, Beijing Institute of Artificial Intelligence, School of Artificial Intelligence, Beijing University of Technology, Beijing, China. [2] The Discipline of Business Analytics, The University of Sydney Business School, The University of Sydney, Camperdown, NSW, Australia. Correspondence to: Yongli Hu <huyongli@bjut.edu.cn>.

*Proceedings of the 43rd International Conference on Machine Learning*, Seoul, South Korea. PMLR 306, 2026. Copyright 2026 by the author(s).

Exact minimization of potential is hindered by structural barriers: *communication infeasibility*, as pairwise couplings require neighbor synchronization; and *multimodal entrapment*, as non-convex potential confines first-order descent to disconnected basins. We propose **GeoEvo** (Geometric Evolution), a federated approximate solver dissolving both via complementary server-client operations. On the server, Fréchet prototypes compress the $\mathcal{O}(K^2)$ pairwise couplings into one message per client. On each client, a Fisher-preconditioned retracted update adapts descent to identity-induced curvature, while a manifold-lifted evolutionary search—combining Natural Evolution Strategies (NES) for stochastic basin escape with Particle Swarm Optimization (PSO) anchored to the broadcast prototype—explores beyond the local basin. A monotone selection rule over the prototype-based surrogate enforces per-step monotone descent, driving the global potential to dissipate in expectation.

Our contributions are threefold: (i) **Conceptual Reframing.** We reconceptualize PMFL by treating modality incompleteness as an intrinsic structural identity that constrains each client to a distinct Riemannian submanifold, shifting the paradigm from compensatory alignment to geometry-aware adaptation. (ii) **Game-Theoretic Foundation.** We formulate PMFL as an Identity-Aware Potential Game, reducing decentralized multi-objective coordination to principled descent on a scalar potential landscape with guaranteed convergence to geometry-consistent Nash equilibria. (iii) **Algorithmic Realization.** We develop GeoEvo, a federated solver that pairs curvature-adaptive Fisher descent with a manifold-lifted evolutionary search. Server-broadcast Fréchet prototypes reduce per-round communication to $\mathcal{O}(K)$, while a monotone selection rule enforces per-step monotone descent and drives global-potential dissipation, yielding an $\mathcal{O}(1/\sqrt{T})$ stationarity rate.

## 2. Problem Formulation

### 2.1. Personalized Federated Learning

Standard FL seeks a single global model $\theta_g$ by minimizing the weighted empirical risk across $K$ clients: $\theta_g = \arg\min_\theta \sum_{k=1}^{K} w_k F_k(\theta)$, where $F_k(\theta) = \mathbb{E}_{(x,y)\sim\mathcal{D}_k}[\ell(f_\theta(x), y)]$ is the local objective, $\mathcal{D}_k$ denotes the local data distribution, and $w_k \geq 0$ is the aggregation weight with $\sum_{k=1}^{K} w_k = 1$. Under statistical heterogeneity, a universal $\theta_g$ generally fails to attain each client's local optimum $\theta_k^\star$, incurring a strictly positive optimality gap $F_k(\theta_g) - F_k(\theta_k^\star) > 0$ (Hanzely & Richtárik, 2020). To mitigate this, PFL aims to learn client-specific models $\{\theta_k\}_{k=1}^{K}$ by jointly leveraging collaborative training and local distribution adaptation, a problem commonly cast as multi-task optimization: $\min_{\{\theta_k\}_{k=1}^{K}} \sum_{k=1}^{K} w_k F_k(\theta_k)$. A classical solution introduces a proximal regularizer (Li et al., 2021): $\min_{\theta_k} F_k(\theta_k) + \frac{\lambda}{2}\|\theta_k - \theta_g\|^2$.

However, existing PFL methods implicitly assume (i) homogeneous client parameterization and (ii) a shared input space $\mathcal{X}$. Both assumptions break down in multimodal settings, where clients operate on distinct feature spaces and often demand heterogeneous architectures.

### 2.2. Multimodal Federated Learning

The collapse of standard PFL premises in multimodal scenarios reveals a deeper barrier: *structural heterogeneity*, as clients observe data from divergent modality subsets over incompatible domains. Let $\mathcal{M} = \{m_1, \ldots, m_M\}$ denote the universe of modalities, with idealized full input space $\mathcal{X} = \prod_{m\in\mathcal{M}} \mathcal{X}^{(m)}$. Each client $k$ observes only $\mathcal{M}_k \subseteq \mathcal{M}$, inducing a client-specific input space $\mathcal{X}_k = \prod_{m\in\mathcal{M}_k} \mathcal{X}^{(m)}$. Given this client-specific input $\mathbf{x}_{\mathcal{M}_k} \in \mathcal{X}_k$, the standard MMFL objective is $\theta_g = \arg\min_\theta \sum_{k=1}^{K} w_k F_k(\theta; \mathcal{M}_k)$, where $F_k(\theta; \mathcal{M}_k) = \mathbb{E}_{(\mathbf{x}_{\mathcal{M}_k}, y)\sim\mathcal{D}_k}[\ell(f_\theta(\mathbf{x}_{\mathcal{M}_k}), y)]$, exposing the structural mismatch: when $\mathcal{M}_i \neq \mathcal{M}_j$, the optimization landscapes are defined on dimensionally incompatible product spaces. Consequently, global parameter averaging becomes semantically ill-posed, as no canonical bijection exists between disparate sensor configurations that would justify a shared Euclidean coordinate system.

**Optimization Divergence.** This structural mismatch exacerbates gradient heterogeneity, quantified by $\Gamma(\theta) = \sum_{k=1}^{K} w_k \|\nabla_\theta F_k(\theta; \mathcal{M}_k) - \nabla_\theta \bar{F}(\theta)\|^2$, with $\bar{F}(\theta) = \sum_k w_k F_k(\theta; \mathcal{M}_k)$. Unlike unimodal FL where $\Gamma$ reflects purely distributional shifts $\mathcal{D}_i \neq \mathcal{D}_j$ (Karimireddy et al., 2020), in MMFL, $\Gamma$ captures structural incompatibility: even with identical distributions, gradients inhabit different effective subspaces, pointing to irreconcilable optima.

**Proposition 2.1** (Structural Divergence of Local Optima). *Let clients $i$ and $j$ share the same underlying full-modality distribution, i.e., $\mathcal{D}_i|_\mathcal{M} = \mathcal{D}_j|_\mathcal{M}$, but observe different modality subsets $\mathcal{M}_i \neq \mathcal{M}_j$. Whenever the modalities in $\mathcal{M}_j \setminus \mathcal{M}_i$ (or $\mathcal{M}_i \setminus \mathcal{M}_j$) carry predictive information about $Y$ not contained in their counterpart, the local optima diverge in the shared parameter space $\Theta$: $\arg\min_{\theta\in\Theta} F_i(\theta; \mathcal{M}_i) \neq \arg\min_{\theta\in\Theta} F_j(\theta; \mathcal{M}_j)$.*

Intuitively, under any strictly proper loss, the optimal predictor on a restricted modality set is the conditional projection of the full-modality Bayes predictor onto the corresponding $\sigma$-field, inducing an irreducible bias on parameter-space stationary points.

### 2.3. Limitations of Vanilla Extensions

We first formalize a vanilla extension of PFL to the multimodal domain as our baseline:

$$\min_{\theta_g, \{\theta_k\}} \sum_{k=1}^{K} w_k F_k(\theta_k; \mathcal{M}_k) + \frac{\lambda}{2} \sum_{k=1}^{K} \|\theta_k - \theta_g\|^2. \quad (1)$$

Eq. (1) admits client-specific parameters $\theta_k$, yet treats modality absence as a statistical deficiency of $\mathcal{D}_k$ rather than a structural property of client $k$ itself.

**Definition 2.2** (Static Absence as Structural Identity). For client $k$, static absence denotes the permanent unavailability of modalities outside its fixed configuration $\mathcal{M}_k$, imposed by time-invariant physical constraints and distinct from transient dynamic occlusion. We refer to $\mathcal{M}_k$ as client $k$'s *structural identity*: an invariant of the observation space $\mathcal{X}_k$ that is irreducible to a statistical deficiency of $\mathcal{D}_k$.

**Irreducible Optimality Gap.** To compare population risks across modality subsets, we assume an underlying full-modality distribution $\mathcal{D}_k|_{\mathcal{M}}$ whose marginalization onto $\mathcal{M}_k$ recovers $\mathcal{D}_k$; for any $\mathcal{M}' \subseteq \mathcal{M}$, define $F_k(\theta; \mathcal{M}') \triangleq \mathbb{E}_{(\mathbf{x},y) \sim \mathcal{D}_k|_{\mathcal{M}'}}[\ell(f_\theta(\mathbf{x}_{\mathcal{M}'}), y)]$ and $F_k^\star(\mathcal{M}') \triangleq \min_{\theta \in \Theta} F_k(\theta; \mathcal{M}')$. Since restricting predictors to the smaller $\sigma$-field $\sigma(X_{\mathcal{M}_k})$ cannot decrease the Bayes risk, $F_k^\star(\mathcal{M}_k) \geq F_k^\star(\mathcal{M})$, with strict inequality whenever the omitted modalities carry predictive information about $Y$ not contained in $\mathcal{X}_k$. In such cases, clients with richer modalities pull $\theta_g$ toward $F_k^\star(\mathcal{M})$, while a modality-limited client cannot go below $F_k^\star(\mathcal{M}_k)$, forcing it to forfeit personalization for an unreachable consensus.

**Gradient Misalignment.** Beyond mis-targeting the optimum, Euclidean coupling injects unidentifiable directions.

**Proposition 2.3** (Incompatible Update Directions under Structural Mismatch). *Let clients $i$ and $j$ have nested modality subsets $\mathcal{M}_i \subset \mathcal{M}_j$, and let $\mathcal{S}_k(\theta) \subseteq \Theta$ denote the* identifiable subspace *of client $k$ at $\theta$, defined as the range of the local Fisher information $\mathbb{E}_{\mathcal{D}_k|_{\mathcal{M}_k}}[\nabla_\theta \ell \, \nabla_\theta \ell^\top]$. Then $\mathcal{S}_i(\theta) \subseteq \mathcal{S}_j(\theta)$, and the directions in $\mathcal{S}_j(\theta) \cap \mathcal{S}_i(\theta)^\perp$ are intrinsically unidentifiable from $\mathcal{M}_i$. Consequently, the proximal pull $\lambda(\theta_g - \theta_i)$ in Eq. (1), with $\theta_g$ influenced by client $j$, generically carries nonzero components along $\mathcal{S}_i(\theta)^\perp$ that contribute zero first-order gradient to $F_i(\theta; \mathcal{M}_i)$ yet inflate update variance, inducing negative transfer for client $i$.*

**Critique of the Robustness Paradigm.** A natural alternative is worst-case robust optimization over modality subsets: $\min_{\theta \in \Theta} \max_{\mathcal{M}' \subseteq \mathcal{M}} F_k(\theta; \mathcal{M}')$. While appropriate for dynamic modality failures, where any subset may manifest at test time, this objective is mis-specified under static constraints: it conflates transient corruption with the client's permanent structural identity, sacrificing personalization for robustness against absence patterns that, by Definition 2.2, never materialize for that client.

### 2.4. PMFL: Identity-Aware Potential Game

Motivated by the structural failures of Eq. (1), we reformulate PMFL as a *potential game* (Monderer & Shapley, 1996) over identity-constrained Riemannian manifolds (Absil et al., 2008), in which each client optimizes within its feasible geometry while contributing to a common global potential that encodes only structurally compatible transfer.

**Definition 2.4** (Client Identity and Feasible Geometry). Extending the structural identity $\mathcal{M}_k$ of Definition 2.2 with the local distribution, we define the *client identity* as the pair $\mathcal{I}_k \triangleq (\mathcal{D}_k, \mathcal{M}_k)$. This identity restricts client $k$'s local optimization to a feasible geometry $\Theta_k \subset \mathbb{R}^{d_k}$, assumed to be either a smooth embedded manifold or a closed constraint set admitting a retraction, endowed with a metric $g_k$ intrinsically determined by $\mathcal{I}_k$.

**Definition 2.5** (Transferable Structure Space). To enable principled comparison between heterogeneous feasible geometries, let $(\mathcal{Z}, d_\mathcal{Z})$ be a common metric space, the *transferable structure space*. For each client $k$, an extraction map $\mathcal{T}_k : \Theta_k \to \mathcal{Z}$ maps local parameters to structural representation $z_k \triangleq \mathcal{T}_k(\theta_k)$. For any modality subset $\mathcal{M}' \subseteq \mathcal{M}$, a restriction map $\pi_{\mathcal{M}'} : \mathcal{Z} \to \mathcal{Z}_{\mathcal{M}'}$ projects onto the subspace supported on $\mathcal{M}'$, with $d_\mathcal{Z}$ inducing a metric on each $\mathcal{Z}_{\mathcal{M}'}$ by restriction.

**Assumption 2.6** (Structural Compatibility). We assume: **(i) Regularity:** For each client $k$, $\mathcal{T}_k$ is measurable and locally Lipschitz on $\Theta_k$. **(ii) Subspace Comparability.** For any $\mathcal{M}' \subseteq \mathcal{M}$, $\pi_{\mathcal{M}'}$ is well-defined. For any pair $(i, j)$ with $S_{ij} \triangleq \mathcal{M}_i \cap \mathcal{M}_j \neq \emptyset$, $\pi_{S_{ij}}(z_i)$ and $\pi_{S_{ij}}(z_j)$ lie in the same metric space $(\mathcal{Z}_{S_{ij}}, d_\mathcal{Z})$ for all $\theta_i \in \Theta_i$, $\theta_j \in \Theta_j$. **(iii) Sparse Interaction:** Symmetric couplings satisfy $\omega_{ij} = \omega_{ji} \geq 0$, with $\omega_{ij} > 0$ if and only if $S_{ij} \neq \emptyset$.

**Game Formulation.** Let $W \triangleq (\theta_1, \ldots, \theta_K) \in \prod_{k=1}^K \Theta_k$ denote the joint strategy profile and $W_{-k}$ the profile with $\theta_k$ excluded. We formulate PMFL as a $K$-player non-cooperative game where each client $k$ minimizes a payoff combining local fitness with pairwise structural alignment:

$$J_k(\theta_k; W_{-k}) \triangleq w_k F_k(\theta_k; \mathcal{I}_k)$$
$$+ \frac{\lambda}{2} \sum_{j \in \mathcal{N}_k} \omega_{kj} \, d_\mathcal{Z}^2 \left(\pi_{S_{kj}}(z_k), \pi_{S_{kj}}(z_j)\right), \tag{2}$$

where $\mathcal{N}_k \triangleq \{j \in [K] \setminus \{k\} : S_{kj} \neq \emptyset\}$ collects $k$'s structurally compatible neighbors. A profile $W^\star = (\theta_1^\star, \ldots, \theta_K^\star)$ is a local Nash equilibrium if $\theta_k^\star \in \arg\min_{\theta \in \Theta_k} J_k(\theta; W_{-k}^\star)$ for all $k \in [K]$.

**Exact Potential Function.** Computing Nash equilibria is PPAD-hard (Daskalakis et al., 2006); the symmetric coupling $\omega_{ij} = \omega_{ji}$ in Eq. (2), by contrast, collapses these per-client neighbor sums into a single exact potential $V : \prod_{k=1}^K \Theta_k \to \mathbb{R}$, counting each unordered pair $\{i, j\}$ once:

$$V(W) \triangleq \sum_{k=1}^K w_k F_k(\theta_k; \mathcal{I}_k)$$
$$+ \frac{\lambda}{2} \sum_{1 \leq i < j \leq K} \omega_{ij} \, d_\mathcal{Z}^2 \left(\pi_{S_{ij}}(z_i), \pi_{S_{ij}}(z_j)\right). \tag{3}$$

**Proposition 2.7** (Exact Potential Game). *Under Assumption 2.6(iii), $V(W)$ in Eq. (3) is an exact potential for the game in Eq. (2): for any client $k$, profile $W_{-k}$, and unilateral deviation $\theta_k' \in \Theta_k$,*

$$J_k(\theta_k'; W_{-k}) - J_k(\theta_k; W_{-k}) = V(\theta_k', W_{-k}) - V(\theta_k, W_{-k}).$$
(4)

*Consequently, every local minimizer of $V$ on $\prod_{k=1}^{K} \Theta_k$ is a local Nash equilibrium of the game.*

**Stability via Riemannian Gradient Flow.** Beyond ensuring Nash equilibria coincide with stationary points of $V$, the potential structure yields a clean stability analysis via gradient flow. On the product manifold $\mathcal{W} \triangleq \prod_{k=1}^{K} \Theta_k$, consider the continuous-time dynamics

$$\dot{\theta}_k(t) = -\text{grad}_{g_k, \theta_k} V(W(t)), \qquad k = 1, \dots, K, \quad (5)$$

where $\text{grad}_{g_k, \theta_k} V$ denotes the Riemannian gradient of $V$ on $\Theta_k$ under $g_k$. Along Eq. (5), $V$ acts as a Lyapunov function:

$$\frac{d}{dt} V(W(t)) = -\sum_{k=1}^{K} \left\| \text{grad}_{g_k, \theta_k} V(W(t)) \right\|_{g_k}^2 \le 0, \quad (6)$$

with equality if and only if $\text{grad}_{g_k, \theta_k} V(W(t)) = 0$ for all $k$, *i.e.*, $W(t)$ is a first-order stationary point of $V$, equivalently a Nash equilibrium of the game by Proposition 2.7. By LaSalle's invariance principle, every bounded trajectory therefore converges to the set of first-order Nash equilibria, regardless of the clients' geometric heterogeneity.

**Decentralized Realization.** By Proposition 2.7, $\text{grad}_{g_k, \theta_k} J_k(\theta_k; W_{-k}) = \text{grad}_{g_k, \theta_k} V(W)$ for all $k \in [K]$. Hence the gradient flow in Eq. (5) is equivalent to the decentralized dynamics

$$\dot{\theta}_k(t) = -\text{grad}_{g_k, \theta_k} J_k(\theta_k(t); W_{-k}(t)), \qquad k = 1, \dots, K,$$

whereby each client independently descending its own payoff $J_k$ monotonically decreases the global potential $V$ and drives $W(t)$ toward a first-order Nash equilibrium, without explicit coordination on $V$.

*Full proofs and technical details are deferred to Appendix B.*

## 3. Methodology

### 3.1. Reference Solver: Exact Potential Minimization

By Proposition 2.7, solving the identity-aware PMFL game in Eq. (2) reduces to minimizing the exact potential $V(W)$ over the product space $\mathcal{W} = \prod_{k=1}^{K} \Theta_k$:

$$\min_{W \in \mathcal{W}} V(W) = \sum_{k=1}^{K} w_k F_k(\theta_k; \mathcal{I}_k)$$
$$+ \frac{\lambda}{2} \sum_{1 \le i < j \le K} \omega_{ij} d_{\mathcal{Z}}^2 \big( \pi_{S_{ij}}(z_i), \pi_{S_{ij}}(z_j) \big).$$
(7)

**Identity-Consistent Fisher Metric.** The minimization landscape of $V$ is highly anisotropic across clients: coordinates affecting frequently-observed modalities differ in sensitivity from those affecting rare ones. Concretely, we realize $\Theta_k$ (Definition 2.4) as a submanifold of an ambient modality-factorized parameterization in which the blocks of absent modalities $\mathcal{M} \setminus \mathcal{M}_k$ are inactive. To equip each $\Theta_k$ with an identity-aware metric $g_k$, we observe that, for such models, the score $\nabla_{\theta_k} \log p(y \mid \mathbf{x}_{\mathcal{M}_k}; \theta_k)$ vanishes along these inactive coordinates. The empirical Fisher information matrix (FIM) (Kunstner et al., 2019) thereby inherits the block-sparse structure of $\mathcal{M}_k$ without explicit masking:

$$\mathbf{F}_k(\theta_k) \triangleq \frac{1}{|\widehat{\mathcal{D}}_k|} \sum_{(\mathbf{x}, y) \in \widehat{\mathcal{D}}_k} \big( \nabla_{\theta_k} \log p(y \mid \mathbf{x}_{\mathcal{M}_k}; \theta_k) \big)$$
$$\big( \nabla_{\theta_k} \log p(y \mid \mathbf{x}_{\mathcal{M}_k}; \theta_k) \big)^\top,$$
(8)

where $\widehat{\mathcal{D}}_k$ denotes the local sample set drawn from $\mathcal{D}_k$ and $p(y \mid \mathbf{x}_{\mathcal{M}_k}; \theta_k)$ is the predictive distribution induced by $f_{\theta_k}$. Restricted to the tangent space $T_{\theta_k} \Theta_k$, $\mathbf{F}_k$ instantiates the intrinsic metric $g_k$.

*Remark* 3.1 (Damped Natural-Gradient Preconditioning). For numerical stability of inversion, we use the damped variant $\widetilde{\mathbf{F}}_k(\theta_k) \triangleq \mathbf{F}_k(\theta_k) + \epsilon \mathbf{I}$ with $\epsilon > 0$, and apply $\widetilde{\mathbf{F}}_k(\theta_k)^{-1}$ as the natural-gradient preconditioner.

**Block Riemannian Descent.** By the decentralized realization of Section 2.4, each client can update independently while collectively minimizing $V$. We discretize the gradient flow in Eq. (5) via block-wise retracted steps:

$$\theta_k^{t+1} = \text{Retr}_{\theta_k^t} \big( -\eta_t \, \text{grad}_{g_k, \theta_k} V(W^t) \big), \qquad k = 1, \dots, K,$$
(9)

where adopting the Fisher metric instantiates $\text{grad}_{g_k, \theta_k} V = \widetilde{\mathbf{F}}_k(\theta_k)^{-1} \nabla_{\theta_k} V(W^t)$, with $\nabla_{\theta_k} V$ computed by automatic differentiation through the composed map $\theta_k \mapsto z_k \mapsto \pi_{S_{kj}}(z_k)$.

**Lemma 3.2** (Per-Step Sufficient Decrease). *Fix $W^t$ and let $W^{t+1}$ follow the simultaneous update in Eq. (9). If $V$ is $L$-smooth w.r.t. retraction on $(\mathcal{W}, g)$ (Assumption 4.2) and $0 < \eta_t \le 1/L$, then*

$$V(W^{t+1}) \le V(W^t) - \frac{\eta_t}{2} \left\| \text{grad} \, V(W^t) \right\|_g^2$$

$$= V(W^t) - \frac{\eta_t}{2} \sum_{i=1}^{K} \left\| \text{grad}_{g_i, \theta_i} V(W^t) \right\|_{g_i}^2.$$
(10)

Telescoping this bound and using $V \ge V_{\text{inf}}$ (with $\inf_t \eta_t > 0$) yields $\lim_{t \to \infty} \left\| \text{grad} \, V(W^t) \right\|_g = 0$, hence $\left\| \text{grad}_{g_k, \theta_k} V(W^t) \right\|_{g_k} \to 0$ for every $k$; every limit point of $\{W^t\}$ is therefore a first-order Nash equilibrium. Equivalently, by Proposition 2.7, every client's payoff is simultaneously stationary ($\text{grad}_{g_k, \theta_k} J_k \to 0$).

## 3.2. GeoEvo: Federated Geometric Evolution

Despite the Nash-equilibrium convergence of the reference solver, two structural barriers preclude its federated realization. *(i) Communication Infeasibility.* The pairwise term in $V$ couples $\theta_i$ and $\theta_j$ across clients, yet each owns only its local parameter; exact computation of $\text{grad}_{g_k, \theta_k} V$ thus demands per-step neighbor synchronization, contradicting the federated premise. *(ii) Multimodal Entrapment.* Composing non-convex losses $\{F_k\}$ with pairwise distances over heterogeneous manifolds renders $V$ deeply multimodal; block Riemannian descent (Eq. (9)) is confined to its initial basin, and no first-order method can cross disconnected attractors. GeoEvo dissolves both via complementary server-client operations: at the server, per-round Fréchet prototypes compress $\mathcal{O}(K^2)$ pairwise couplings into $\mathcal{O}(K)$ per-client messages; at each client, population-level evolutionary search traverses disconnected attractors. Algorithm 1 formalizes the alternating procedure.

### Server: Environment Field Construction.

Let $z_k^t \triangleq \mathcal{T}_k(\theta_k^t)$ denote the structural representation of client $k$ at round $t$, and let $\mathcal{S}^t \subseteq [K]$ be the set of clients participating in round $t$. The server collects $\{z_k^t : k \in \mathcal{S}^t\}$ and for each client $k$, forms the neighbor representation set $\mathcal{E}_k^t \triangleq \{z_j^t : j \in \mathcal{N}_k^t\}, \mathcal{N}_k^t \triangleq \mathcal{N}_k \cap \mathcal{S}^t$, where $\mathcal{N}_k$ and the coupling sparsity $\{\omega_{kj}, S_{kj}\}$ follow Assumption 2.6(iii).

### Fréchet Prototype.

To compress the per-pair projections of $V$ in Eq. (3) while evaluating each neighbor only on its active overlap, we collapse the $|\mathcal{N}_k^t|$ pairwise distance terms into a single Fréchet objective whose minimizer is the round-$t$ prototype:

$$Z_k^t \in \arg\min_{z \in \mathcal{Z}} \sum_{j \in \mathcal{N}_k^t} \omega_{kj} \, d_{\mathcal{Z}}^2 \Big( \pi_{S_{kj}}(z), \, \pi_{S_{kj}}(z_j^t) \Big). \quad (11)$$

We summarize the neighborhood by its union support $S_k^t \triangleq \bigcup_{j \in \mathcal{N}_k^t} S_{kj} \subseteq \mathcal{M}_k$ and aggregate weight $\bar{\omega}_k^t \triangleq \sum_{j \in \mathcal{N}_k^t} \omega_{kj}$, both used client-side below and well-defined there since $S_k^t \subseteq \mathcal{M}_k$. The minimizer $Z_k^t$ is the weighted Fréchet mean of $k$'s neighbors on the identity-induced metric space $(\mathcal{Z}, d_{\mathcal{Z}}^2)$, compressing all pairwise structural couplings $\{(S_{kj}, \omega_{kj})\}_{j \in \mathcal{N}_k^t}$ into a single neighborhood centroid. Unlike class-centric prototypes, $Z_k^t$ is a per-client *neighborhood centroid* driven by the game potential $V$.

### Single-Message Broadcast.

The server transmits the triple $(Z_k^t, S_k^t, \bar{\omega}_k^t)$ as a single message to client $k$, achieving: (i) *Communication Compression:* $\mathcal{O}(K^2)$ pairwise traffic is reduced to $\mathcal{O}(K)$ aggregate-broadcast; (ii) *Privacy Preservation:* the server observes only structural representations $\{z_j^t\}$, never raw parameters $\{\theta_j^t\}$; (iii) *Identity-Aware Aggregation:* the Fréchet mean on $(\mathcal{Z}, d_{\mathcal{Z}}^2)$ inherits the identity-induced geometry rather than enforcing a Euclidean mean.

---

**Algorithm 1** GeoEvo: Federated Geometric Evolution

---

**Require:** Rounds $T$, local steps $E$, stepsizes $(\eta, \eta_{\text{nes}})$, samples $m$, coefficients $(c_0, c_1, c_2)$, damping $\epsilon$

1: **Initialize:** for all $k \in [K]$: $\theta_k \in \Theta_k$, $u_k \leftarrow \mathbf{0}$, $z_k^{\text{best}} \leftarrow \mathcal{T}_k(\theta_k)$
2: **for** $t = 0, 1, \ldots, T-1$ **do**
3:    **Server:** sample $\mathcal{S}^t \subseteq [K]$; broadcast
     $\{(Z_k^t, S_k^t, \bar{\omega}_k^t)\}_{k \in \mathcal{S}^t}$      via Eq. (11)
4:    **for** each $k \in \mathcal{S}^t$ **in parallel do**
5:      **for** $s = 1, \ldots, E$ **do**
6:        $z_k \leftarrow \mathcal{T}_k(\theta_k)$
7:        **Descent:** compute $\theta_k^{\text{base}}$    via Eq. (13)
8:        **NES:** compute $\theta_k^{\text{NES}}$      via Eq. (15)
9:        **PSO:** compute $\theta_k^{\text{PSO}}$      via Eq. (16)
10:       **Select:** $\theta_k \leftarrow \theta_k^+$      via Eq. (17)
11:       **if** $\phi_k(\mathcal{T}_k(\theta_k)) < \phi_k(z_k^{\text{best}})$ **then**
        $z_k^{\text{best}} \leftarrow \mathcal{T}_k(\theta_k)$
12:      **end for**
13:      Upload $z_k^{t+1} \leftarrow \mathcal{T}_k(\theta_k)$ to server
14:    **end for**
15: **end for**

---

### Client: Manifold-Constrained Trajectory Search.

**Prototype-Based Surrogate.** The portion of $V$ in Eq. (3) depending on $\theta_k$ couples it to every neighbor through the pairwise sum $\sum_{j \in \mathcal{N}_k^t} \omega_{kj} \, d_{\mathcal{Z}}^2(\pi_{S_{kj}}(z_k), \pi_{S_{kj}}(z_j^t))$, which is prohibitive under federated communication: each per-pair projection $\pi_{S_{kj}}$ demands a separate neighbor exchange. This many-to-many coupling collapses into a one-to-prototype surrogate:

$$\tilde{J}_k^t(\theta_k) \triangleq w_k F_k(\theta_k; \mathcal{I}_k) + \frac{\lambda}{2} \bar{\omega}_k^t \, d_{\mathcal{Z}}^2\big(\pi_{S_k^t}(z_k), \pi_{S_k^t}(Z_k^t)\big). \quad (12)$$

Since $\tilde{J}_k^t$ depends only on the broadcast triple, each client minimizes it through purely local computation.

**Geometry-Aware Descent.** The Fisher-preconditioned retracted update of Eq. (9), originally defined for the exact potential $V$, transfers directly to the surrogate $\tilde{J}_k^t$ by substituting $\nabla_{\theta_k} \tilde{J}_k^t$ for $\nabla_{\theta_k} V$. This inheritance preserves the identity-induced block-sparsity of $\widetilde{\mathbf{F}}_k$ at no additional cost, ensuring the descent direction remains supported on $\mathcal{M}_k$-relevant coordinates. The resulting client-side Fisher-retracted step is given by:

$$\theta_k^{\text{base}} \leftarrow \text{Retr}_{\theta_k}\Big( -\eta \, \widetilde{\mathbf{F}}_k(\theta_k)^{-1} \, \nabla_{\theta_k} \tilde{J}_k^t(\theta_k) \Big). \quad (13)$$

**Manifold-Lifted Evolutionary Search.** $\tilde{J}_k^t$ inherits $V$'s multimodality, confining the descent in Eq. (13) to the basin of $\theta_k^{\text{cur}}$. This intrinsic limitation of first-order methods motivates evolutionary exploration in the structure space $\mathcal{Z}$—a lower-dimensional, identity-aligned domain whose global

topology is more tractable than $\Theta_k$—with proposals lifted back to $\Theta_k$ via the structure-matching operator

$$\text{Lift}_k(z; \theta_k) \triangleq \arg\min_{\theta \in \Theta_k} d_{\mathcal{Z}}^2(\mathcal{T}_k(\theta), z), \qquad (14)$$

which returns the parameter whose structural embedding is closest to $z$; in practice, a few warm-started gradient steps from $\theta_k$ suffice.

*NES Proposal.* Let $\phi_k(z) \triangleq \tilde{J}_k^t(\text{Lift}_k(z; \theta_k))$. The Gaussian-smoothed gradient estimator is $\widehat{\nabla}_z \phi_{k,\sigma_z}(z_k) = \frac{1}{m\sigma_z} \sum_{i=1}^m \phi_k(z_k + \sigma_z \varepsilon_i) \varepsilon_i$ with $\varepsilon_i \sim \mathcal{N}(0, \mathbf{I})$, yielding the NES proposal

$$\theta_k^{\text{NES}} \leftarrow \text{Lift}_k\Big(z_k - \eta_{\text{nes}} \widehat{\nabla}_z \phi_{k,\sigma_z}(z_k); \theta_k\Big). \qquad (15)$$

*PSO Proposal.* Client $k$ maintains a velocity $u_k \in \mathcal{Z}$ and personal best $z_k^{\text{best}}$; setting the broadcast prototype $Z_k^t$ as the swarm's global attractor embeds cross-client structural coordination directly into the swarm dynamics.

$$u_k \leftarrow c_0 u_k + c_1 r_1(z_k^{\text{best}} - z_k) + c_2 r_2(Z_k^t - z_k),$$
$$\theta_k^{\text{PSO}} \leftarrow \text{Lift}_k(z_k + u_k; \theta_k).$$
$$(16)$$

**Monotone Selection.** At each local step, GeoEvo evaluates the descent candidate $\theta_k^{\text{base}}$ alongside the evolutionary proposals $\{\theta_k^{\text{NES}}, \theta_k^{\text{PSO}}\}$, retaining the surrogate-minimizer:

$$\theta_k^+ \leftarrow \arg\min_{\theta \in \{\theta_k^{\text{cur}}, \theta_k^{\text{base}}, \theta_k^{\text{NES}}, \theta_k^{\text{PSO}}\}} \tilde{J}_k^t(\theta), \qquad (17)$$

and updates the personal best $z_k^{\text{best}} \leftarrow \mathcal{T}_k(\theta_k^+)$ when this strictly decreases $\phi_k$. Including the current iterate $\theta_k^{\text{cur}}$ in the candidate pool guarantees that $\tilde{J}_k^t(\theta_k^+) \leq \tilde{J}_k^t(\theta_k^{\text{cur}})$. This condition transfers the per-step decrease of Lemma 3.2 from $V$ to the surrogate, remaining robust to any inexactness in the structural $\text{Lift}_k$ operator since selection always evaluates the exact surrogate objective $\tilde{J}_k^t$ across all candidates.

*Full proofs and technical details are deferred to Appendix C.*

# 4. Theoretical Analysis

Our theoretical analysis bridges the idealized centralized reference solver and its communication-feasible federated realization. For the exact potential $V$ in Eq. (3), local minimizers are local Nash equilibria (Proposition 4.4) and retracted block descent on $(\mathcal{W}, g)$ yields sufficient decrease and first-order stationarity (Theorem 4.5). For its federated realization, the server prototype is Fréchet-optimal for the exact per-pair coupling (Proposition 4.6); under partial participation and bounded prototype drift, the GeoEvo client update attains an $\mathcal{O}(1/\sqrt{T})$ surrogate-stationarity rate despite its monotone selection (Theorem 4.10); and NES/PSO proposals preserve monotone surrogate descent within each round (Proposition 4.12).

## 4.1. Preliminaries and Assumptions

We endow the product space $\mathcal{W} = \prod_{k=1}^K \Theta_k$ with the product metric $g \triangleq \oplus_{k=1}^K g_k$, and write $\text{grad}\, V(W)$ for the Riemannian gradient on $(\mathcal{W}, g)$ and $\text{grad}_{g_k, \theta_k} V(W)$ for its $k$-th block. All retractions are denoted by $\text{Retr}$.

**Assumption 4.1** (Geometry and Regularity). For each client $k$, $\Theta_k$ admits a second-order retraction $\text{Retr}$, and $F_k(\cdot; \mathcal{I}_k)$ and $\mathcal{T}_k$ are twice continuously differentiable. For each edge $\{k, j\}$ with $S_{kj} \neq \emptyset$, the map $\theta_k \mapsto d_{\mathcal{Z}}^2(\pi_{S_{kj}}(\mathcal{T}_k(\theta_k)), \pi_{S_{kj}}(\mathcal{T}_j(\theta_j)))$ is continuously differentiable in $\theta_k$ for any fixed $\theta_j$.

**Assumption 4.2** (Smoothness on Manifolds). There exists $L > 0$ such that the potential $V$ on $(\mathcal{W}, g)$ *and* every round-$t$ client surrogate $\tilde{J}_k^t$ on $(\Theta_k, g_k)$ are $L$-smooth w.r.t. $\text{Retr}$. For any $W \in \mathcal{W}$ and $\xi \in T_W \mathcal{W}$,

$$V(\text{Retr}_W(\xi)) \leq V(W) + \langle \text{grad}\, V(W), \xi \rangle_g + \frac{L}{2}\|\xi\|_g^2,$$

and analogously $\tilde{J}_k^t(\text{Retr}_{\theta_k}(\xi_k)) \leq \tilde{J}_k^t(\theta_k) + \langle \text{grad}_{g_k} \tilde{J}_k^t(\theta_k), \xi_k \rangle_{g_k} + \frac{L}{2}\|\xi_k\|_{g_k}^2$ for $\xi_k \in T_{\theta_k} \Theta_k$. The surrogate bound is inherited from the twice continuous differentiability of $F_k$ and $\mathcal{T}_k$ together with the second-order retraction (Assumption 4.1) on the compact sublevel set (Assumption 4.3(ii)).

**Assumption 4.3** (Lower Boundedness and Compactness). (i) $V$ is bounded below on $\mathcal{W}$, *i.e.*, $V(W) \geq V_{\text{inf}} > -\infty$. (ii) (Optional, for limit-point statements) the sublevel set $\mathcal{L}(V(W^0)) = \{W \in \mathcal{W} : V(W) \leq V(W^0)\}$ is compact.

## 4.2. Exact Potential and First-Order Nash Equilibria

**Proposition 4.4** (Exact Potential Game (Restated)). *Under Assumption 2.6(iii) and the symmetric edge convention ($\omega_{kj} = \omega_{jk}$ and the same $S_{kj}$ and $\pi_{S_{kj}}$ are used on edge $\{k, j\}$), $V$ in Eq. (3) is an exact potential for the game in Eq. (2), i.e., Eq. (4) holds for any unilateral deviation. Consequently, any local minimizer of $V$ on $\mathcal{W}$ is a local Nash equilibrium.*

*Proof Sketch.* Fix client $k$ and $W_{-k}$, and consider a unilateral deviation $\theta_k \to \theta_k'$. The terms of $V(W)$ affected by $\theta_k$ are the local term $w_k F_k(\theta_k; \mathcal{I}_k)$ and the pairwise terms on edges $\{k, j\}$. Since $V$ counts each undirected edge once, the induced change in $V$ equals

$$w_k\big(F_k(\theta_k'; \mathcal{I}_k) - F_k(\theta_k; \mathcal{I}_k)\big)$$
$$+ \frac{\lambda}{2} \sum_{j \in \mathcal{N}_k} \omega_{kj}\Big(d_{\mathcal{Z}}^2(\pi_{S_{kj}}(\mathcal{T}_k(\theta_k')), \pi_{S_{kj}}(\mathcal{T}_j(\theta_j)))$$
$$- d_{\mathcal{Z}}^2(\pi_{S_{kj}}(\mathcal{T}_k(\theta_k)), \pi_{S_{kj}}(\mathcal{T}_j(\theta_j)))\Big), \qquad (18)$$

which is exactly $J_k(\theta_k'; W_{-k}) - J_k(\theta_k; W_{-k})$ by the symmetry $\omega_{kj} = \omega_{jk}$ and the use of the same overlap operator $\pi_{S_{kj}}$ on each edge. $\qquad \square$

**First-Order Nash Equilibrium (FONE).** If $\operatorname{grad} V(W^\star) = 0$ (equivalently, $\operatorname{grad}_{g_k, \theta_k} V(W^\star) = 0$ for every $k$), we call $W^\star$ a *first-order Nash equilibrium* (FONE), *i.e.*, each player satisfies a first-order stationarity condition under its feasible geometry. FONE is necessary but not sufficient for local Nash optimality.

### 4.3. Retracted Block Descent and Stationarity

**Theorem 4.5** (Stationarity Convergence (Simultaneous Blocks))**.** *Under Assumptions 4.2–4.3, for the iterates $\{W^t\}$ of the block descent in Eq. (9) with stepsizes $0 < \eta_t \leq 1/L$, the sequence $\{V(W^t)\}$ is non-increasing and $\sum_{t=0}^{\infty} \eta_t \|\operatorname{grad} V(W^t)\|_g^2 < \infty$. In particular, if $\inf_t \eta_t > 0$, then $\lim_{t \to \infty} \|\operatorname{grad} V(W^t)\|_g = 0$; under Assumption 4.3(ii) and the continuity of $\operatorname{grad} V$ (implied by the second-order retraction and $C^2$ data of Assumption 4.1), every limit point of $\{W^t\}$ is a FONE.*

**From the Reference Solver to GeoEvo.** Theorem 4.5 characterizes the *idealized centralized* reference solver, which evaluates the exact gradient $\operatorname{grad} V$ and therefore requires per-step neighbor synchronization. GeoEvo replaces this exact coupling by stale broadcast prototypes under partial participation $\mathcal{S}^t$; deterministic monotone decrease of $V$ is therefore not guaranteed in general for coupled nonconvex problems. The federated guarantees relevant to GeoEvo are the per-round surrogate stationarity of Theorem 4.10 and the monotone selection of Proposition 4.12.

### 4.4. Optimal Surrogate Approximation

Fix round $t$ and recall $\mathcal{N}_k^t = \mathcal{N}_k \cap \mathcal{S}^t$, $S_k^t = \bigcup_{j \in \mathcal{N}_k^t} S_{kj} \subseteq \mathcal{M}_k$, and $\bar\omega_k^t = \sum_{j \in \mathcal{N}_k^t} \omega_{kj}$. We assume $\mathcal{Z}$ is a product space and $d_{\mathcal{Z}}$ acts coordinatewise under masking, so that $\pi_S(\pi_{S'}(\cdot)) = \pi_S(\cdot)$ whenever $S \subseteq S'$. The per-pair coupling of $V$ attributable to client $k$ is $\Phi_k^t(z) \triangleq \frac{\lambda}{2} \sum_{j \in \mathcal{N}_k^t} \omega_{kj} d_{\mathcal{Z}}^2(\pi_{S_{kj}}(z), \pi_{S_{kj}}(z_j^t))$.

**Proposition 4.6** (Variational Optimality of Prototypes)**.** *The server-computed prototype $Z_k^t$ in Eq. (11) satisfies $\Phi_k^t(Z_k^t) \leq \Phi_k^t(z)$ for all $z \in \mathcal{Z}$, i.e., $Z_k^t$ minimizes the exact per-pair coupling $\Phi_k^t$. By coordinatewise separation, $Z_k^t$ is, on each modality $m \in S_k^t$, the weighted Fréchet mean over the neighbors that possess $m$.*

To reduce communication, GeoEvo then replaces $\Phi_k^t$ with the single-prototype surrogate that collapses the $|\mathcal{N}_k^t|$ per-pair terms into a single prototype $Z_k^t$:

$$\widetilde{\Phi}_k^t(\theta_k; Z_k^t) \triangleq \frac{\lambda}{2} \bar\omega_k^t \, d_{\mathcal{Z}}^2\left(\pi_{S_k^t}(\mathcal{T}_k(\theta_k)), \pi_{S_k^t}(Z_k^t)\right). \quad (19)$$

**Assumption 4.7** (Surrogate Approximation Error)**.** There exists $\delta_k^t \geq 0$ such that for all $\theta_k \in \Theta_k$, $|\Phi_k^t(\mathcal{T}_k(\theta_k)) - \widetilde{\Phi}_k^t(\theta_k; Z_k^t)| \leq \delta_k^t$. This bound captures two effects: the

prototype-compression error (neighbor dispersion on $S_k^t$), and the residual from collapsing the per-pair projections $\{\pi_{S_{kj}}\}_j$ and weights $\{\omega_{kj}\}_j$ onto the single union mask $\pi_{S_k^t}$ and aggregate weight $\bar\omega_k^t$.

### 4.5. Stochastic Convergence to Surrogate Stationarity

For practical federated settings, we use stochastic gradient estimators for local optimization of $\tilde{J}_k^t$. Define the round-$t$ surrogate value over participating clients by $\widetilde{J}^t \triangleq \frac{1}{|\mathcal{S}^t|} \sum_{k \in \mathcal{S}^t} \tilde{J}_k^t(\theta_k^t)$ and $\widetilde{J}_{\inf} \triangleq \inf_t \mathbb{E}[\widetilde{J}^t]$.

**Assumption 4.8** (Stochastic Gradients)**.** Each participating client employs an unbiased stochastic gradient estimator $\widehat{\operatorname{grad}}_{g_k} \tilde{J}_k$ with bounded variance:

$$\mathbb{E}[\widehat{\operatorname{grad}}_{g_k, \theta_k} \tilde{J}_k \mid \theta_k, Z_k^t] = \operatorname{grad}_{g_k, \theta_k} \tilde{J}_k, \quad (20)$$

$$\mathbb{E}[\|\widehat{\operatorname{grad}}_{g_k, \theta_k} \tilde{J}_k - \operatorname{grad}_{g_k, \theta_k} \tilde{J}_k\|_{g_k}^2 \mid \theta_k, Z_k^t] \leq \sigma^2. \quad (21)$$

**Assumption 4.9** (Bounded Prototype Drift)**.** The per-round change of the surrogate induced by prototype updates is summable in expectation: $\sum_{t=1}^{T-1} \mathbb{E}\left[\frac{1}{|\mathcal{S}^t|} \sum_{k \in \mathcal{S}^t} \left(\tilde{J}_k^t(\theta_k^t) - \tilde{J}_k^{t-1}(\theta_k^t)\right)\right] \leq \Delta_T$, where $\Delta_T$ bounds cumulative prototype drift $\sum_t \|\pi_{S_k^t}(Z_k^t) - \pi_{S_k^{t-1}}(Z_k^{t-1})\|$.

**Theorem 4.10** (Non-Convex Convergence Rate (Surrogate Stationarity))**.** *Under Assumptions 4.2, 4.8 and 4.9, consider the iterates generated by the GeoEvo client update (17) with stepsize $\eta \leq 1/L$, whose candidate pool deterministically contains the unbiased stochastic base step $\theta_k^{\mathrm{base}}$. Then there exists an absolute constant $C > 0$ such that*

$$\min_{t < T} \mathbb{E}\left[\frac{1}{|\mathcal{S}^t|} \sum_{k \in \mathcal{S}^t} \|\operatorname{grad}_{g_k, \theta_k^t} \tilde{J}_k^t\|_{g_k}^2\right] \leq \frac{C(\mathbb{E}[\bar{J}^0] - \widetilde{J}_{\inf} + \Delta_T)}{\eta T} + CL\eta\sigma^2. \quad (22)$$

*The summability in Assumption 4.9 renders the cumulative prototype drift uniformly bounded in $T$, i.e. $\Delta_T = \mathcal{O}(1)$. Setting $\eta = \Theta(1/\sqrt{T})$ with summable drift yields an $\mathcal{O}(1/\sqrt{T})$ rate; if $\Delta_T = o(\sqrt{T})$ the bound is $o(1)$, and $\Delta_T = \Theta(\sqrt{T})$ leaves an $\mathcal{O}(1)$ stationarity neighborhood.*

*Proof Sketch.* The candidate pool in Eq. (17) contains the unbiased base step $\theta_k^{\mathrm{base}}$, so $\tilde{J}_k^t(\theta_k^{t+1}) \leq \tilde{J}_k^t(\theta_k^{\mathrm{base}})$ for every noise realization; in expectation the filtered iterate thus inherits the Riemannian-SGD descent of $\theta_k^{\mathrm{base}}$ (Assumptions 4.2, 4.8). Telescoping over the $E$ local steps and $T$ rounds with inter-round drift absorbed by $\Delta_T$ gives Eq. (22). $\square$

*Remark* 4.11 (Expected Dissipation of $V$)**.** Lifting the surrogate guarantees to $V$ via the error bound $\delta_k^t$, partial-participation gap $\beta_k^t$, and bounded drift $\Delta_T$, the global potential dissipates in expectation per round up to an $\mathcal{O}(\delta_k^t + \beta_k^t + \Delta_T)$ residual (Proposition D.9). This residual vanishes asymptotically as the compression becomes exact, participation becomes full ($\beta_k^t \to 0$), and the drift remains summable.

*Table 1.* Performance comparison on CREMA-D across varying missing rates in terms of accuracy (%).

| Method | 30% | 50% | 70% |
|---|---|---|---|
| **Baselines** | | | |
| Centralized | $54.10 \pm 0.80$ | $52.30 \pm 0.95$ | $50.95 \pm 1.40$ |
| IID | $52.07 \pm 2.06$ | $51.29 \pm 1.99$ | $49.60 \pm 2.02$ |
| **FL (Global & Local Trade-off)** | | | |
| FedProx (Li et al., 2020) | $47.30 \pm 0.94$ | $46.86 \pm 1.38$ | $48.59 \pm 2.07$ |
| SCAFFOLD (Karimireddy et al., 2020) | $50.51 \pm 3.00$ | $48.67 \pm 2.02$ | $47.14 \pm 2.67$ |
| HetPFL (Ye & Tang, 2025) | $48.47 \pm 1.52$ | $47.38 \pm 1.05$ | $46.50 \pm 3.71$ |
| FedMoSWA (Liu et al., 2025) | $50.68 \pm 1.21$ | $49.61 \pm 1.37$ | $48.56 \pm 3.65$ |
| **PFL (Euclidean-Constrained)** | | | |
| Ditto (Li et al., 2021) | $47.57 \pm 0.94$ | $46.94 \pm 0.99$ | $43.14 \pm 2.46$ |
| HB-PFL (Thapa & Li, 2025) | $51.18 \pm 0.88$ | $48.40 \pm 1.22$ | $46.95 \pm 2.21$ |
| pFedMoAP (Luo et al., 2025) | $51.91 \pm 1.54$ | $50.37 \pm 1.05$ | $47.50 \pm 2.90$ |
| FedMGP (Bo et al., 2026) | $51.60 \pm 0.97$ | $49.55 \pm 2.15$ | $47.10 \pm 4.48$ |
| **MMFL (Compensation-oriented)** | | | |
| FedMultimodal (Feng et al., 2023) | $48.88 \pm 0.68$ | $45.30 \pm 1.88$ | $44.82 \pm 2.15$ |
| $M^3$Fed (Li et al., 2024) | $50.64 \pm 0.88$ | $49.85 \pm 1.58$ | $47.42 \pm 2.30$ |
| FedMEMA (Dai et al., 2024) | $51.15 \pm 0.90$ | $50.10 \pm 2.25$ | $49.35 \pm 2.32$ |
| FED-PRIME (Phung et al., 2025) | $50.95 \pm 1.38$ | $49.07 \pm 1.40$ | $48.08 \pm 2.54$ |
| PEPSY (Nguyen et al., 2026) | $52.02 \pm 0.85$ | $50.85 \pm 1.44$ | $48.90 \pm 3.05$ |
| **PMFL (Proposed Identity-Aware)** | | | |
| PMFL (Vanilla) | $50.31 \pm 1.26$ | $49.69 \pm 1.28$ | $47.54 \pm 2.46$ |
| Oracle (Exact) | $56.40 \pm 0.85$ | $56.85 \pm 1.10$ | $54.10 \pm 1.95$ |
| GeoEvo (Prototype) | $55.82 \pm 0.92$ | $53.90 \pm 1.40$ | $52.05 \pm 3.50$ |
| GeoEvo (Ours) | $\mathbf{56.06} \pm 1.03$ | $\mathbf{57.56} \pm 1.63$ | $\mathbf{54.67} \pm 2.14$ |

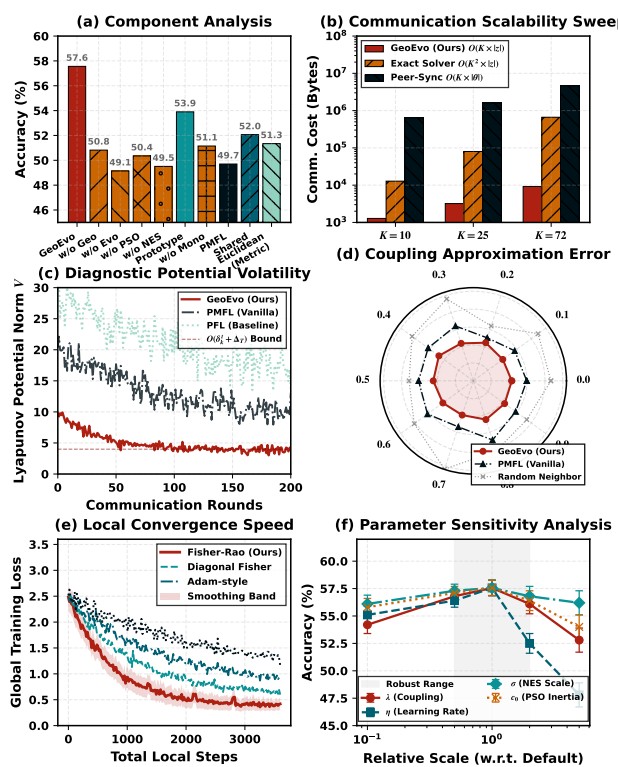

*Figure 1.* Ablation of GeoEvo on CREMA-D at 50% missing rate.

## 4.6. Monotone Surrogate Descent and Exploration

**Proposition 4.12** (Evolutionary Proposals Preserve Monotonicity). *Fix round $t$ and fields $\{Z_k^t\}$. Let $\theta_k'$ be any proposal (gradient-based or evolutionary). If the client uses the acceptance rule in Eq. (17), which retains the current iterate $\theta_k^t$ (denoted $\theta_k^{\mathrm{cur}}$ therein) in the candidate pool, then $\tilde{J}_k(\theta_k^{t+1}; Z_k^t) \leq \tilde{J}_k(\theta_k^t; Z_k^t)$. This holds unconditionally: the filter evaluates the exact parameter-space surrogate $\tilde{J}_k^t(\cdot)$, so it is immune to any optimization inexactness in the structural $\mathrm{Lift}_k$ operator, which at worst yields a weaker candidate. Under Assumption 4.7, this yields an approximate decrease of the corresponding coupling contribution to $V$ up to $\delta_k^t$.*

*Remark* 4.13 (Saddle Avoidance in Practice). The stochastic exploratory nature of NES/PSO proposals, combined with the monotone acceptance rule, facilitates identifying descent directions in regions of negative curvature, empirically mitigating stagnation at saddle points.

*Full proofs and supporting lemmas are in Appendix D.*

## 5. Experiments

**Experimental Setup.** We evaluate GeoEvo on three multimodal benchmarks: CREMA-D (Cao et al., 2014) (audiovisual emotion), CrisisMMD (Alam et al., 2018) (image-text crisis), and the large-scale UPMC Food-101 (Wang et al., 2015) (image-text food). To simulate federated heterogeneity, we partition CREMA-D by speaker identity, while the other two datasets are split using a Dirichlet prior $\mathrm{Dir}(\alpha)$. Modality incompleteness is introduced by two patterns: proportional (dataset-native) and binomial (uniformly random). Following (Feng et al., 2023; Phung et al., 2025), we employ heterogeneous backbones (*e.g.*, MobileNetV2, MobileBERT) for small-scale tasks and ViLT for UPMC Food-101. Each client reserves a portion of its local data as a private test set to evaluate the model's alignment with its personalized distribution. We compare against 15 representative baselines spanning centralized, FL, PFL, and MMFL paradigms, as well as our proposed PMFL variants.

**Overall Performance Superiority.** GeoEvo consistently outperforms all baselines across the three benchmarks. On CREMA-D (Table 1), it achieves 54.67% under a 70% missing rate, surpassing even the Oracle as incompleteness deepens; this advantage extends to CrisisMMD (Figure 2) and UPMC Food-101 (Figure 3), where GeoEvo remains top-tier across heterogeneity levels and degrades far more gracefully than FL, PFL, and MMFL methods as missing rates rise. This superiority stems from a mechanistic shift: modality incompleteness is an intrinsic structural identity to be respected, not a deficiency to be repaired, thereby guiding geometry-aware adaptation along each client's identifiable coordinates.

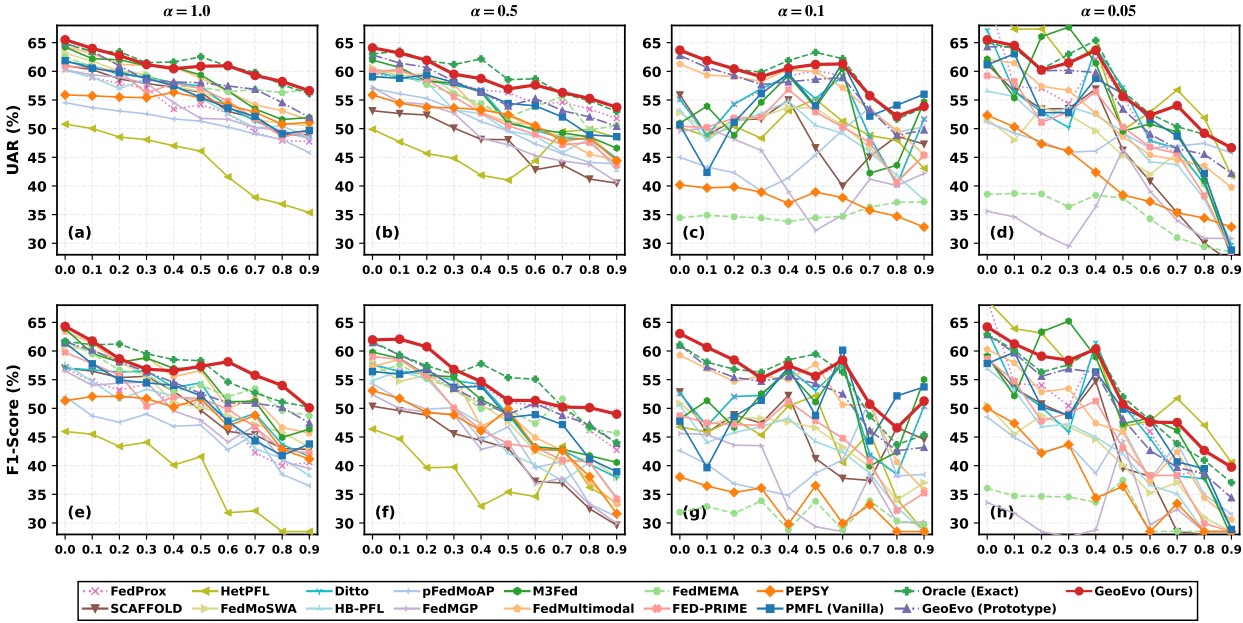

*Figure 2.* Comparative performance on CrisisMMD under binomial modality simulation across varying heterogeneity levels.

By casting PMFL as an identity-aware potential game, Geo-Evo replaces semantically ill-posed shared-coordinate averaging with adaptation on client-specific Riemannian submanifolds, whose Fisher metric inherits each client's modality block-sparsity to nullify unobserved-dimension updates and eliminate the imputation noise capping compensation-oriented MMFL. This is why the baselines forfeit ground: FL fuses dimensionally incompatible optima into a single model, Euclidean PFL injects unidentifiable directions that drive negative transfer, and MMFL mistakes a permanent structural void for a transient shift, incurring an irreducible optimality gap. Conversely, GeoEvo coordinates via server-broadcast Fréchet prototypes that preserve heterogeneous equilibria, with a monotone selection rule turning evolutionary search into safe basin escape.

**Ablation and Diagnostic Analysis.** Figure 1(a) isolates each component: removing evolutionary search (*w/o Evo*) or geometric coupling (*w/o Geo*) causes drops of 8.5 and 6.8 pp from the full 57.6%. Ablating either evolutionary operator, the monotone rule, or the Fisher metric, each costs accuracy, confirming that every design choice is necessary rather than incidental. These benefits incur no communication overhead: Figure 1(b) shows GeoEvo's traffic scaling linearly as $\mathcal{O}(K|z|)$, far below the exact solver's quadratic $\mathcal{O}(K^2|z|)$ coupling, so the prototype compression scales to large populations. The potential trajectory (Figure 1(c)) shows GeoEvo contracting $V$ into the residual band and fluctuating within it, as expected from Proposition D.9 under stale prototypes and partial participation; the monotone rule (Proposition 4.12) bounds exploration volatility, whereas

the uncoupled vanilla PMFL and PFL baselines exhibit persistent, non-vanishing fluctuations. Across missing-rate regimes, Figure 1(d) keeps GeoEvo's coupling error lowest, validating the variational optimality of the Fréchet prototype (Proposition 4.6) over vanilla and random-neighbor surrogates; Figure 1(e) shows the Fisher–Rao update converging faster than diagonal-Fisher and Adam-style preconditioners; and Figure 1(f) confirms robustness across hyperparameters, with sensitivity concentrated on the learning rate as predicted by stepsize condition $\eta \leq 1/L$.

*Full configurations and results are deferred to Appendix E.*

## 6. Conclusion

This paper reconceptualizes PMFL by treating modality incompleteness as intrinsic structural identity, formalizing cross-client collaboration as an Identity-Aware Potential Game realized through GeoEvo, which integrates curvature-adaptive Fisher descent with manifold-lifted evolutionary search under Fr'echet prototypes. GeoEvo attains monotone surrogate descent and an $\mathcal{O}(1/\sqrt{T})$ stationarity rate at $\mathcal{O}(K)$ communication, empirically improving personalization and robustness across missing-modality regimes. By design, GeoEvo's single-prototype compression trades a minor surrogate-approximation residual inherent to distributed non-convex landscapes for $\mathcal{O}(K)$ communication efficiency. Tightening this controlled residual via variance-reduced prototypes, and extending the potential-game framework to asynchronous client populations, represent highly promising future directions.

## Acknowledgement

This research is supported by the National Natural Science Foundation of China under Grant Nos. 62572017, U21B2038, U19B2039, 62206007, and 52302397, the National Key R&D Program of China under Grant No. 2021ZD0111902, the R&D Program of Beijing Municipal Education Commission under Grant No. KZ202210005008, the Beijing Municipal Fund General Projects under Grant No. 4262037, the Beijing Natural Science Foundation under Grant No. 4262034, the China Postdoctoral Science Foundation under Grant No. 2025M781451, and the Fundamental Research Funds for Beijing Municipal Universities under Grant No. 312000546325001.

## Impact Statement

This paper presents work whose goal is to advance the field of Machine Learning. There are many potential societal consequences of our work, none of which we feel must be specifically highlighted here.

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

# A. Related Works

### A.1. Federated Learning

Federated learning (FL) enables collaborative training without centralizing raw data, but client heterogeneity often causes biased optimization and unstable convergence. Classic FedAvg (McMahan et al., 2017) and its heterogeneity-robust variants address client drift via regularization (FedProx (Li et al., 2020)) or control variates (SCAFFOLD (Karimireddy et al., 2020)). However, these approaches assume homogeneous input spaces and model architectures, which becomes restrictive in multimodal settings where clients suffer permanent structural missingness. Our work builds on FL by explicitly modeling each client's structural identity and enabling geometry-consistent coupling across heterogeneous submanifolds.

### A.2. Personalized Federated Learning

Personalized Federated Learning (PFL) addresses statistical heterogeneity by learning client-adaptive models rather than enforcing a single global solution. Existing PFL methods are commonly grouped into: **(i) Regularization-based** approaches that constrain local drift from a reference model (*e.g.*, Ditto (Li et al., 2021)); **(ii) Model decoupling or partial sharing** methods that separate globally shared representations from client-specific components (*e.g.*, FedPer (Arivazhagan et al., 2019), FedRep (Collins et al., 2021)); **(iii) Meta-learning** frameworks that learn an initialization enabling fast client adaptation (*e.g.*, Per-FedAvg (Fallah et al., 2020)); and **(iv) Prototype or distillation-based** techniques that exchange lightweight knowledge to improve personalization without sharing full parameters (*e.g.*, FedProto (Tan et al., 2022b)). While effective in unimodal settings, these paradigms typically assume a shared parameter geometry and a comparable notion of distance across clients. In contrast, our setting treats permanent modality absence as an intrinsic structural identity, which induces heterogeneous feasible geometries and makes naive global consensus or modality-agnostic aggregation prone to representational distortion.

### A.3. Multimodal Federated Learning

Multimodal Federated Learning (MMFL) extends FL to scenarios where each client observes multiple modalities, often with severe modality imbalance and missingness. Most existing solutions treat missing modalities as a deficiency to be compensated toward a full-modality template, which can be broadly grouped into three paradigms: **(i) Imputation-based** methods that reconstruct missing modalities using external resources (*e.g.*, public datasets (Yu et al., 2023), prototype libraries (Saha et al., 2025), reconstruction models (Wang et al., 2024), or pre-trained foundation models (Qiao et al., 2025)); **(ii) Disentanglement-based** methods that partition architectures into modality-specific branches and coordinate them via auxiliary alignment or distillation (Peng et al., 2024; Ouyang et al., 2023; Sun et al., 2024a; Chen & Zhang, 2024); and **(iii) Aggregation-based** methods that design structured message passing or block-wise aggregation among multimodal clients (Chen & Zhang, 2022; Qi & Li, 2024; Borazjani et al., 2024). While effective when missingness is transient or recoverable, these compensation-centric objectives can conflate permanent client-specific absence with distribution shift, encouraging artificial correspondence and amplifying aggregation bias. In contrast, we formalize modality configurations as identities and couple clients only through shared modality supports in a common structure space, avoiding alignment on unsupported dimensions—thereby preserving each client's intrinsic optimization geometry. This stands in sharp contrast to prior MMFL methods that implicitly enforce a shared Euclidean template.

### A.4. Game-Theoretic and Evolutionary Perspectives in Federated Optimization

Game-theoretic formulations model strategic interactions among heterogeneous agents; potential games are particularly attractive because equilibria can be characterized via a single scalar potential function (Monderer & Shapley, 1996). This perspective aligns naturally with personalization: clients may converge to distinct solutions while retaining structured coupling. In parallel, evolutionary computation provides gradient-free or hybrid exploration mechanisms for non-convex optimization, including Natural Evolution Strategies (NES) (Wierstra et al., 2014) and swarm-based methods such as Particle Swarm Optimization (PSO) (Kennedy & Eberhart, 1995). These tools have been adopted in distributed learning and FL to improve robustness, exploration–exploitation balance, and resilience to noisy objectives (*e.g.*, population-based training (Jaderberg et al., 2017)). Our method connects these threads by (i) casting identity-aware Personalized Multimodal Federated Learning (PMFL) as an exact potential game under symmetric coupling on shared supports, (ii) designing a geometry-aware evolutionary optimizer that respects client-specific information geometry while enforcing monotone decrease of a local surrogate objective via an acceptance rule, and (iii) compressing the $\mathcal{O}(K^2)$ pairwise coupling to $\mathcal{O}(K)$ communication via server-broadcast Fréchet prototypes.

## B. Appendix for the Problem Formulation

**Notation.** $\mathcal{M} = \{m_1, \ldots, m_M\}$ is the modality universe. Client $k$ has a fixed modality set $\mathcal{M}_k \subseteq \mathcal{M}$ and identity $\mathcal{I}_k = (\mathcal{D}_k, \mathcal{M}_k)$ (Definition 2.4). Its feasible geometry is the manifold constraint set $\Theta_k$ with intrinsic metric $g_k$. The joint strategy profile is $W = (\theta_1, \ldots, \theta_K) \in \mathcal{W} \triangleq \prod_{k=1}^{K} \Theta_k$, and $W_{-k}$ omits the $k$-th block. For a pair $(k, j)$ the shared support is $S_{kj} \triangleq \mathcal{M}_k \cap \mathcal{M}_j$ and $\mathcal{N}_k \triangleq \{j \in [K] \setminus \{k\} : S_{kj} \neq \emptyset\}$. The structure-extraction map is $\mathcal{T}_k : \Theta_k \to \mathcal{Z}$ with $z_k \triangleq \mathcal{T}_k(\theta_k)$, the restriction map is $\pi_S : \mathcal{Z} \to \mathcal{Z}_S$, and $d_{\mathcal{Z}}$ denotes the metric on each $\mathcal{Z}_S$. Coupling weights satisfy $\omega_{kj} = \omega_{jk} \geq 0$ with $\omega_{kj} > 0 \iff S_{kj} \neq \emptyset$ (Assumption 2.6(iii)). We write $\langle \cdot, \cdot \rangle_{g_k}, \| \cdot \|_{g_k}$ for the metric on $T_{\theta_k} \Theta_k$ and $\mathrm{grad}_{g_k, \theta_k}$ for the Riemannian gradient in the $k$-th block. Throughout, "$Y \not\perp X \mid Z$" denotes conditional dependence.

### B.1. Static vs. Dynamic Absence: Justifying the Structural Identity

Definition 2.2 promotes the modality set $\mathcal{M}_k$ from a statistical accident of $\mathcal{D}_k$ to a permanent attribute of the client. The following regime distinction makes precise the constraint that licenses this promotion.

**Definition B.1** (Modality-Absence Regimes). Let $\mathcal{M}_k^{(t)}$ be the modalities available to client $k$ at round $t$.

- **Dynamic Absence:** $\exists t \neq t'$ with $\mathcal{M}_k^{(t)} \neq \mathcal{M}_k^{(t')}$ (transient occlusion, sensor dropout).

- **Static Absence:** $\mathcal{M}_k^{(t)} = \mathcal{M}_k$ for all $t \geq 0$, reflecting time-invariant hardware, policy, or cost constraints.

**Proposition B.2** (Mis-Specification of Full-Modality Consensus under Static Absence). *Assume static absence and let $\mathcal{D}_k$ be supported on $\mathcal{X}_k = \prod_{m \in \mathcal{M}_k} \mathcal{X}^{(m)}$. Any objective that, for client $k$, scores a predictor on a coordinate $m \in \mathcal{M} \setminus \mathcal{M}_k$ (e.g., a consensus term enforcing agreement on full-modality outputs) is evaluated on an event of $\mathcal{D}_k$-probability zero, hence is uninformative for $k$ and cannot be reduced by any admissible $\theta_k \in \Theta_k$.*

*Proof.* Under static absence the inputs $X_{\mathcal{M} \setminus \mathcal{M}_k}$ are never realized for client $k$: $\mathcal{D}_k(X_{\mathcal{M} \setminus \mathcal{M}_k} \in \cdot)$ places all mass on the masked null configuration. Consequently $f_{\theta_k}$ is, by construction of $\Theta_k$, a function of $\mathbf{x}_{\mathcal{M}_k}$ only, and any term depending on absent coordinates is constant in $\theta_k$ on $\mathrm{supp}(\mathcal{D}_k)$, contributing zero gradient. Asking $k$ to match information it can never observe is therefore mis-specified. $\square$

This is precisely the failure of the vanilla extension Eq. (1): it treats $\mathcal{M} \setminus \mathcal{M}_k$ as missing data of $\mathcal{D}_k$ rather than as a structural invariant, and couples clients through a Euclidean consensus that ignores $\mathcal{M}_k$. The remedy adopted in the main text is to couple clients only through the shared supports $S_{kj}$ and to keep $\mathcal{M}_k$ invariant.

### B.2. The Irreducible Optimality Gap

We formalize the inequality $F_k^\star(\mathcal{M}_k) \geq F_k^\star(\mathcal{M})$ asserted in the "Irreducible Optimality Gap" paragraph. For a modality subset $\mathcal{M}' \subseteq \mathcal{M}$ write $F_k(\theta; \mathcal{M}') \triangleq \mathbb{E}_{\mathcal{D}_k|_{\mathcal{M}'}}[\ell(f_\theta(\mathbf{x}_{\mathcal{M}'}), Y)]$ and $F_k^\star(\mathcal{M}') \triangleq \inf_{\theta \in \Theta} F_k(\theta; \mathcal{M}')$.

**Lemma B.3** (Monotonicity of Bayes Risk in the Observation $\sigma$-Field). *Let $\ell$ be a strictly proper loss whose Bayes act given a $\sigma$-field $\mathcal{G}$ is the $\mathcal{G}$-conditional functional of $Y$ (e.g., squared loss with $\mathbb{E}[Y \mid \mathcal{G}]$, or log loss with $P(Y \mid \mathcal{G})$). If the hypothesis class $\{f_\theta\}_{\theta \in \Theta}$ is expressive enough to realize these conditional functionals, then for $\mathcal{M}_k \subseteq \mathcal{M}$,*

$$F_k^\star(\mathcal{M}_k) \geq F_k^\star(\mathcal{M}), \tag{23}$$

*with strict inequality whenever $Y \not\perp X_{\mathcal{M} \setminus \mathcal{M}_k} \mid X_{\mathcal{M}_k}$.*

*Proof.* Let $\mathcal{G}_k \triangleq \sigma(X_{\mathcal{M}_k}) \subseteq \sigma(X_{\mathcal{M}}) \triangleq \mathcal{G}$. By the defining Bayes optimality of a strictly proper loss, $F_k^\star(\mathcal{M}')$ equals the Bayes risk $\mathcal{R}(\mathcal{G}')$ attained by the $\sigma(X_{\mathcal{M}'})$-measurable Bayes act. The Bayes risk is non-increasing under refinement of the conditioning $\sigma$-field (the tower property: conditioning on a finer field can only reduce the expected divergence), hence $\mathcal{R}(\mathcal{G}_k) \geq \mathcal{R}(\mathcal{G})$, which is Eq. (23). For strict proper losses, the inequality is strict exactly when the finer field changes the Bayes act on a positive-measure event, *i.e.*, when $Y \not\perp X_{\mathcal{M} \setminus \mathcal{M}_k} \mid X_{\mathcal{M}_k}$. $\square$

Lemma B.3 explains why a single consensus $\theta_g$ is corrosive: clients with richer modalities drive $\theta_g$ toward the lower floor $F_k^\star(\mathcal{M})$, which a modality-limited client provably cannot reach, so consensus forces it to forfeit personalization for an unattainable target.

## B.3. Proof of Proposition 2.1 (Structural Divergence of Local Optima)

We give the argument for squared loss to make the Bayes-optimal step explicit; the conclusion extends to any strictly proper loss by the same conditional-functional argument used in Lemma B.3.

**Proposition B.4** (Structural Divergence of Local Optima (Expanded)). *Let an underlying full-modality vector $X_{\mathcal{M}}$ and label $Y$ have joint law $P$, and let clients $i, j$ share it, $\mathcal{D}_i|_{\mathcal{M}} = \mathcal{D}_j|_{\mathcal{M}} = P$, but observe $\mathcal{M}_i \neq \mathcal{M}_j$. Under squared loss define $F(\theta; \mathcal{M}') = \mathbb{E}[(f_\theta(X_{\mathcal{M}'}) - Y)^2]$. If $Y \not\perp\!\!\!\perp X_{\mathcal{M}_j \setminus \mathcal{M}_i} \mid X_{\mathcal{M}_i}$ (the extra modalities carry predictive information absent from $\mathcal{M}_i$), then the Bayes predictors differ,*

$$f_i^\star(x_{\mathcal{M}_i}) = \mathbb{E}[Y \mid X_{\mathcal{M}_i} = x_{\mathcal{M}_i}] \neq \mathbb{E}[Y \mid X_{\mathcal{M}_j} = x_{\mathcal{M}_j}] = f_j^\star(x_{\mathcal{M}_j}) \quad (a.s.), \tag{24}$$

*and the minimizers in a shared parameterization generally differ:* $\arg\min_{\theta \in \Theta} F(\theta; \mathcal{M}_i) \neq \arg\min_{\theta \in \Theta} F(\theta; \mathcal{M}_j)$.

*Proof.* Under squared loss, the Bayes predictor given a $\sigma$-field $\mathcal{G}$ is $\mathbb{E}[Y \mid \mathcal{G}]$. Put $\mathcal{G}_i \triangleq \sigma(X_{\mathcal{M}_i})$, $\mathcal{G}_j \triangleq \sigma(X_{\mathcal{M}_j})$, and $f_i^\star = \mathbb{E}[Y \mid \mathcal{G}_i]$, $f_j^\star = \mathbb{E}[Y \mid \mathcal{G}_j]$. If $\mathcal{M}_i \subsetneq \mathcal{M}_j$ then $\mathcal{G}_i \subsetneq \mathcal{G}_j$, and by the tower property

$$\mathbb{E}[f_j^\star \mid \mathcal{G}_i] = \mathbb{E}[\mathbb{E}[Y \mid \mathcal{G}_j] \mid \mathcal{G}_i] = \mathbb{E}[Y \mid \mathcal{G}_i] = f_i^\star, \tag{25}$$

so $f_i^\star$ is the $\mathcal{G}_i$-measurable projection of $f_j^\star$. The dependence hypothesis means $\mathbb{E}[Y \mid \mathcal{G}_j]$ is *not* $\mathcal{G}_i$-measurable, hence $f_j^\star \neq f_i^\star$ on a positive-measure event.

For general (possibly non-nested) $\mathcal{M}_i \neq \mathcal{M}_j$ the same conclusion holds without nesting: the Bayes predictor $f_i^\star = \mathbb{E}[Y \mid \mathcal{G}_i]$ is $\sigma(X_{\mathcal{M}_i})$-measurable and hence cannot depend on $X_{\mathcal{M}_j \setminus \mathcal{M}_i}$, whereas the hypothesis $Y \not\perp\!\!\!\perp X_{\mathcal{M}_j \setminus \mathcal{M}_i} \mid X_{S_{ij}}$ (with $S_{ij} = \mathcal{M}_i \cap \mathcal{M}_j$) forces $f_j^\star = \mathbb{E}[Y \mid \mathcal{G}_j]$ to depend nontrivially on $X_{\mathcal{M}_j \setminus \mathcal{M}_i}$, so it is not $\mathcal{G}_i$-measurable. A $\mathcal{G}_i$-measurable random variable cannot a.s. equal one that genuinely depends on $X_{\mathcal{M}_j \setminus \mathcal{M}_i}$; therefore $f_i^\star \neq f_j^\star$ on a positive-measure event. The symmetric hypothesis on $\mathcal{M}_i \setminus \mathcal{M}_j$ handles the other direction.

Finally, if $\{f_\theta\}$ is rich enough to approximately realize these Bayes predictors, the population minimizers are the parameters realizing them; since the predictors differ, so do the minimizers. $\square$

Thus, the restricted-modality optimum is a conditional projection of the full-modality optimum, and projecting onto *different* $\sigma$-fields relocates the stationary points in the shared parameter space—precisely the divergence asserted in the main text and the population-level cause of the gradient heterogeneity $\Gamma(\theta)$.

## B.4. Proof of Proposition 2.3 (Incompatible Update Directions)

This subsection provides the proof omitted from the main text, completing the symmetry: three formal claims (Proposition 2.1—2.7), three proofs. Whereas Proposition 2.1 shows the targets diverge, here we show that the Euclidean coupling of Eq. (1) injects update directions that client $i$ cannot even identify, producing negative transfer.

**Setup.** For client $k$ write the per-sample score and population gradient

$$s_k(\theta; \mathbf{x}, y) \triangleq \nabla_\theta \ell(f_\theta(\mathbf{x}_{\mathcal{M}_k}), y), \qquad g_k(\theta) \triangleq \nabla_\theta F_k(\theta; \mathcal{M}_k) = \mathbb{E}_{\mathcal{D}_k|_{\mathcal{M}_k}}[s_k(\theta; \mathbf{x}, y)], \tag{26}$$

and the Fisher and its identifiable subspace

$$\mathbf{F}_k(\theta) \triangleq \mathbb{E}_{\mathcal{D}_k|_{\mathcal{M}_k}}[s_k s_k^\top], \qquad \mathcal{S}_k(\theta) \triangleq \text{range } \mathbf{F}_k(\theta). \tag{27}$$

We use the modality-factorized realization of $\Theta_k$ adopted in Section 3: the parameter splits as $\theta = (\theta^{\text{shr}}, \{\theta^{(m)}\}_{m \in \mathcal{M}})$, where $\theta^{(m)}$ influences the output only through modality $m$'s input branch, and the absent blocks $\{m \notin \mathcal{M}_k\}$ are inactive for client $k$. Let $\mathcal{A}_k$ index the active coordinates of $k$ (*i.e.*, $\theta^{\text{shr}}$ and $\{\theta^{(m)} : m \in \mathcal{M}_k\}$) and $E_{\mathcal{A}_k} \triangleq \text{span}\{e_a : a \in \mathcal{A}_k\}$. We assume the mild *marginal consistency* of a shared backbone: masking the inputs of $\mathcal{M}_j \setminus \mathcal{M}_i$ reduces client $j$'s model to client $i$'s, so

$$s_i(\theta; \mathbf{x}_{\mathcal{M}_i}, y) = \Pi_{\mathcal{A}_i} s_j(\theta; \mathbf{x}_{\mathcal{M}_j}, y)\Big|_{\mathcal{M}_j \setminus \mathcal{M}_i \text{ masked}}, \tag{28}$$

where $\Pi_{\mathcal{A}_i}$ is the coordinate projector onto $E_{\mathcal{A}_i}$. This is exactly the modality-dropout-consistent, shared-encoder parameterization used by the method.

**Lemma B.5** (Score Support and Gradient Containment). *For every $k$ and $\theta$: (a) $s_k(\theta; \cdot) \in E_{\mathcal{A}_k}$ almost surely, hence $\mathcal{S}_k(\theta) \subseteq E_{\mathcal{A}_k}$; (b) $g_k(\theta) \in \mathcal{S}_k(\theta)$.*

*Proof.* (a) Coordinates outside $\mathcal{A}_k$ index inactive branches, so $f_\theta(\mathbf{x}_{\mathcal{M}_k})$ does not depend on them; their entries of $s_k$ vanish a.s., giving $s_k \in E_{\mathcal{A}_k}$ and therefore $\mathbf{F}_k = \mathbb{E}[s_k s_k^\top]$ has range in $E_{\mathcal{A}_k}$. (b) Since $\mathcal{S}_k = \operatorname{range} \mathbf{F}_k = \overline{\operatorname{span}} \operatorname{supp}(s_k)$, each realization satisfies $s_k \in \mathcal{S}_k$; as $\mathcal{S}_k$ is a closed subspace, $g_k = \mathbb{E}[s_k] \in \mathcal{S}_k$. $\qquad\square$

**Proposition B.6** (Incompatible Update Directions (Expanded)). *Let $\mathcal{M}_i \subset \mathcal{M}_j$ and assume Eq. (28). Then at every $\theta$:*

(i) ***Nesting:*** *$\mathcal{S}_i(\theta) \subseteq \mathcal{S}_j(\theta)$, and $\mathcal{S}_j(\theta) \cap \mathcal{S}_i(\theta)^\perp \neq \{0\}$ whenever $Y \not\perp\!\!\!\perp X_{\mathcal{M}_j \setminus \mathcal{M}_i} \mid X_{\mathcal{M}_i}$.*

(ii) ***First-order unidentifiability:*** *every $v \in \mathcal{S}_i(\theta)^\perp$ satisfies $v^\top \mathbf{F}_i(\theta)\, v = 0$ and $\langle g_i(\theta), v \rangle = 0$; i.e., $v$ is invisible to both the curvature and the first-order objective of client $i$.*

(iii) ***Conflicted proximal pull:*** *the proximal force $\lambda(\theta_g - \theta_i)$ of Eq. (1), with $\theta_g$ influenced by client $j$, has a generically nonzero component in $\mathcal{S}_i(\theta)^\perp$ that produces no first-order decrease of $F_i$.*

*Proof.* (i) By Eq. (28), $\operatorname{supp}(s_i) = \Pi_{\mathcal{A}_i} \operatorname{supp}(s_j) \subseteq \operatorname{supp}(s_j)$ as subsets of $\mathbb{R}^d$, whence $\mathcal{S}_i = \overline{\operatorname{span}} \operatorname{supp}(s_i) \subseteq \overline{\operatorname{span}} \operatorname{supp}(s_j) = \mathcal{S}_j$. For the strict gap, note that under the dependence hypothesis the extra branch $\mathcal{M}_j \setminus \mathcal{M}_i$ changes the Bayes predictor (Lemma B.3), so $s_j$ excites at least one coordinate of $\theta^{(m)}$, $m \in \mathcal{M}_j \setminus \mathcal{M}_i$, that $s_i$ never excites; that direction lies in $\mathcal{S}_j \cap \mathcal{S}_i^\perp$.

(ii) For $v \in \mathcal{S}_i^\perp$, $v^\top \mathbf{F}_i v = \mathbb{E}[(s_i^\top v)^2] = 0$ because $s_i \in \mathcal{S}_i \perp v$ a.s.; the same a.s. orthogonality gives $\langle g_i, v \rangle = \mathbb{E}[s_i^\top v] = 0$, and indeed $g_i \in \mathcal{S}_i$ by Lemma B.5(b).

(iii) Decompose $\theta_g - \theta_i = P_{\mathcal{S}_i}(\theta_g - \theta_i) + P_{\mathcal{S}_i^\perp}(\theta_g - \theta_i)$ with $P$ the orthogonal projectors. Because $\theta_g$ aggregates client $j$, whose identifiable subspace strictly contains $\mathcal{S}_i$ by (i), $\theta_g$ generically carries mass in $\mathcal{S}_j \cap \mathcal{S}_i^\perp$, so $P_{\mathcal{S}_i^\perp}(\theta_g - \theta_i) \neq 0$. By (ii) this component is orthogonal to $g_i$ and lies in the null curvature of $F_i$, hence contributes nothing to the first-order decrease of $F_i$. $\qquad\square$

*Remark* B.7 (Negative Transfer via Conflicted Pull: Variance Inflation Mechanism). Consider client $i$ running stochastic Riemannian descent on the regularized objective $F_i(\theta; \mathcal{M}_i) + \frac{\lambda}{2}\|\theta - \theta_g\|^2$ with unbiased gradient noise of covariance $\Sigma$. Project the recursion onto $\mathcal{S}_i^\perp$ via $P \triangleq P_{\mathcal{S}_i^\perp}$. By Proposition B.6(ii), $P g_i = 0$, so the in-subspace iterate $u^t \triangleq P(\theta_i^t - \theta_g)$ obeys the linear recursion

$$u^{t+1} = (1 - \eta\lambda)\, u^t - \eta\, P\, \xi^t, \qquad \mathbb{E}[\xi^t] = 0,\ \operatorname{Cov}(\xi^t) = \Sigma, \tag{29}$$

whose stationary second moment is $\mathbb{E}\|u^\infty\|^2 = \dfrac{\eta}{2\lambda - \eta\lambda^2}\operatorname{tr}(P\Sigma P) > 0$ whenever noise leaks into $\mathcal{S}_i^\perp$. Thus, the consensus pull keeps $\theta_i$ persistently fluctuating along directions that reduce no loss for $i$ (zero signal) while accumulating variance (nonzero $\mathbb{E}\|u^\infty\|^2$). This is the precise mechanism of the negative transfer, and it motivates replacing the Euclidean consensus by the overlap-restricted coupling $d_{\mathcal{Z}}^2(\pi_{S_{ij}}(z_i), \pi_{S_{ij}}(z_j))$ of Eq. (2), which by construction acts only on shared—hence mutually identifiable—supports.

## B.5. Well-Posedness of the Edge Couplings

When $\mathcal{M}_i \neq \mathcal{M}_j$, the two landscapes live on incompatible product spaces; the role of $\mathcal{T}_k$ and $\pi_S$ are to manufacture a single comparable object per edge, so that each distance in Eqs. (2)–(3) is well-defined.

**Lemma B.8** (Edge-wise Comparability via Restriction). *Fix $k \neq j$ with $S_{kj} \neq \emptyset$. Under Assumption 2.6(ii), the restricted representations*

$$\pi_{S_{kj}}\big(\mathcal{T}_k(\theta_k)\big), \ \ \pi_{S_{kj}}\big(\mathcal{T}_j(\theta_j)\big) \ \in \ \mathcal{Z}_{S_{kj}} \tag{30}$$

*lie in the same metric space $(\mathcal{Z}_{S_{kj}}, d_{\mathcal{Z}})$ for all $\theta_k \in \Theta_k$, $\theta_j \in \Theta_j$, so the edge distance $d_{\mathcal{Z}}^2(\pi_{S_{kj}}(z_k), \pi_{S_{kj}}(z_j))$ is well-defined and symmetric in $(k, j)$. If $S_{kj} = \emptyset$ then $\omega_{kj} = 0$ (Assumption 2.6(iii)) and the edge contributes no term, so no undefined restriction is ever evaluated.*

*Proof.* By Assumption 2.6(ii), the transformations $\mathcal{T}_k$ and $\mathcal{T}_j$ map the local parameters into a common ambient space $\mathcal{Z}$, and the restriction $\pi_{S_{kj}}$ projects onto the shared coordinate block $\mathcal{Z}_{S_{kj}}$. Consequently, $\pi_{S_{kj}}(\mathcal{T}_k(\theta_k))$ and $\pi_{S_{kj}}(\mathcal{T}_j(\theta_j))$ both lie in the same metric space $(\mathcal{Z}_{S_{kj}}, d_{\mathcal{Z}})$, rendering the edge distance $d_{\mathcal{Z}}^2(\cdot, \cdot)$ well-defined and symmetric in $(k, j)$. Assumption 2.6(iii) guarantees that whenever $S_{kj} = \emptyset$, the corresponding weight $\omega_{kj} = 0$, so the edge contributes no term and no undefined restriction is ever evaluated. The symmetry follows because both endpoints are processed by the same operator $\pi_{S_{kj}}$ on the undirected edge $\{k, j\}$ and $d_{\mathcal{Z}}$ is a metric. $\square$

Lemma B.8 is the structural prerequisite for the potential property below: it guarantees that the term shared between $J_k$ and $J_j$ on edge $\{k, j\}$ is one and the same quantity.

## B.6. Proof of Proposition 2.7 (Exact Potential Game)

**Proposition B.9** (Exact Potential Game (Expanded)). *Consider the payoffs $J_k$ of Eq. (2) and the candidate potential $V$ of Eq. (3). Assume (i) $\omega_{kj} = \omega_{jk}$ for all $k \neq j$; (ii) the same restriction $\pi_{S_{kj}}$ is used on each undirected edge $\{k, j\}$; (iii) $\omega_{kj} > 0 \iff S_{kj} \neq \emptyset$. Then $V$ is an* exact potential*: for every $k$, every fixed $W_{-k}$, and all $\theta_k, \theta_k' \in \Theta_k$,*

$$J_k(\theta_k'; W_{-k}) - J_k(\theta_k; W_{-k}) = V(\theta_k', W_{-k}) - V(\theta_k, W_{-k}). \tag{31}$$

*Proof.* Define the edge discrepancy, well-defined whenever $\omega_{kj} > 0$ by Lemma B.8,

$$D_{kj}(\theta_k, \theta_j) \triangleq d_{\mathcal{Z}}^2\Big(\pi_{S_{kj}}(\mathcal{T}_k(\theta_k)), \pi_{S_{kj}}(\mathcal{T}_j(\theta_j))\Big), \qquad D_{kj} = D_{jk}. \tag{32}$$

Fix $k$ and hold $W_{-k}$ constant. Split the potential into the $k$-local term, the other clients' local terms, and the pairwise sum:

$$V(W) = \underbrace{w_k F_k(\theta_k; \mathcal{I}_k)}_{\text{depends on } \theta_k} + \underbrace{\sum_{i \neq k} w_i F_i(\theta_i; \mathcal{I}_i)}_{\text{constant in } \theta_k} + \frac{\lambda}{2} \sum_{1 \leq i < j \leq K} \omega_{ij} D_{ij}(\theta_i, \theta_j). \tag{33}$$

Partition the unordered-pair sum into edges incident to $k$ and edges avoiding $k$:

$$\sum_{1 \leq i < j \leq K} \omega_{ij} D_{ij} = \sum_{j \neq k} \omega_{\min\{k,j\} \max\{k,j\}} D_{kj}(\theta_k, \theta_j) + \sum_{\substack{1 \leq i < j \leq K \\ i \neq k, \, j \neq k}} \omega_{ij} D_{ij}(\theta_i, \theta_j). \tag{34}$$

Because the edge set is undirected, $\omega_{kj} = \omega_{jk}$, and $\omega_{kj} = 0$ for non-neighbors, the first sum collapses to $\sum_{j \in \mathcal{N}_k} \omega_{kj} D_{kj}(\theta_k, \theta_j)$, while the second is constant in $\theta_k$. Hence

$$V(W) = \left( w_k F_k(\theta_k; \mathcal{I}_k) + \frac{\lambda}{2} \sum_{j \in \mathcal{N}_k} \omega_{kj} D_{kj}(\theta_k, \theta_j) \right) + \Phi(W_{-k}), \tag{35}$$

where $\Phi(W_{-k})$ gathers every term independent of $\theta_k$. The bracket is exactly $J_k(\theta_k; W_{-k})$ of Eq. (2). Subtracting the values at $\theta_k'$ and $\theta_k$ cancels $\Phi(W_{-k})$ and yields Eq. (31). $\square$

The factor $\frac{1}{2}$ in $V$ and the once-per-unordered-pair summation are exactly what convert the per-client neighbor sums of $\{J_k\}$ into a single shared function; symmetry $\omega_{kj} = \omega_{jk}$ together with the common $\pi_{S_{kj}}$ (Lemma B.8) is what makes the cancellation exact rather than approximate. Consequently, every local minimizer of $V$ admits no profitable unilateral deviation and is a local Nash equilibrium, as stated below Eq. (4).

## B.7. Gradient-Flow Stability: Lyapunov Descent and LaSalle Convergence

We discharge the stability claims attached to the Riemannian gradient flow Eq. (5), namely the dissipation identity Eq. (6) and convergence to first-order Nash equilibria.

**Definition B.10** (First-Order Nash Equilibrium). Assume each $\Theta_k$ is a smooth manifold (or a retraction-compatible constraint set) and each $J_k(\cdot; W_{-k})$ is differentiable. A profile $W^\star \in \mathcal{W}$ is a *first-order Nash equilibrium* (FONE) if $\mathrm{grad}_{g_k, \theta_k} J_k(\theta_k^\star; W_{-k}^\star) = 0$ for every $k$.

**Proposition B.11** (Lyapunov Property of $V$ and LaSalle Convergence). *Let $V$ be continuously differentiable on the product manifold $(\mathcal{W}, g)$ with $g \triangleq \oplus_k g_k$, and let $W(\cdot)$ solve the flow Eq. (5), $\dot{\theta}_k = -\text{grad}_{g_k, \theta_k} V(W)$. Then along any solution,*

$$\frac{d}{dt} V(W(t)) = -\sum_{k=1}^{K} \left\| \text{grad}_{g_k, \theta_k} V(W(t)) \right\|_{g_k}^2 \leq 0, \tag{36}$$

*with equality iff $W(t)$ is a FONE. If moreover the sublevel set $\{W : V(W) \leq V(W(0))\}$ is compact, then every solution is bounded and its $\omega$-limit set is a nonempty, connected, invariant subset of the FONE set; hence $\text{grad}\, V(W(t)) \to 0$ and $W(t)$ converges to the set of first-order Nash equilibria, irrespective of the clients' geometric heterogeneity.*

*Proof.* On the product manifold the gradient decomposes blockwise, $\text{grad}\, V = (\text{grad}_{g_k, \theta_k} V)_k$, and $\langle \cdot, \cdot \rangle_g = \sum_k \langle \cdot, \cdot \rangle_{g_k}$. By the chain rule along the flow,

$$\frac{d}{dt} V(W(t)) = \sum_k \left\langle \text{grad}_{g_k, \theta_k} V, \dot{\theta}_k \right\rangle_{g_k} = -\sum_k \left\langle \text{grad}_{g_k, \theta_k} V, \text{grad}_{g_k, \theta_k} V \right\rangle_{g_k} = -\sum_k \left\| \text{grad}_{g_k, \theta_k} V \right\|_{g_k}^2, \tag{37}$$

which is Eq. (36); the sum vanishes iff every block gradient vanishes, *i.e.* $W(t)$ is a FONE. Under compactness of the sublevel set, monotonicity of $V$ along the flow keeps $W(t)$ in that compact set, so the solution is bounded and $V(W(t))$ decreases to a finite limit. LaSalle's invariance principle then forces the $\omega$-limit set into the largest invariant set contained in $\{W : \frac{d}{dt} V = 0\} = \{W : \text{grad}\, V(W) = 0\}$, which is exactly the FONE set, giving $\text{grad}\, V(W(t)) \to 0$ and convergence to that set. $\qquad\square$

**B.8. Gradient Alignment and the Decentralized Realization**

Finally, we justify the "Decentralized Realization" claim that each client may descend its *own* payoff $J_k$ while collectively minimizing $V$.

**Lemma B.12** (Blockwise Gradient Alignment). *If the game is an exact potential game with a differentiable potential $V$ (Proposition B.9), then for every $k$ and every $W \in \mathcal{W}$,*

$$\text{grad}_{g_k, \theta_k} J_k(\theta_k; W_{-k}) = \text{grad}_{g_k, \theta_k} V(W). \tag{38}$$

*Consequently, $W^\star$ is a first-order stationary point of $V$ ($\text{grad}_{g_k, \theta_k} V(W^\star) = 0$ for all $k$) if and only if $W^\star$ is a FONE.*

*Proof.* By the exact-potential identity Eq. (31), $J_k(\cdot; W_{-k})$ and $V(\cdot, W_{-k})$ differ by the additive constant $\Phi(W_{-k})$ that is independent of $\theta_k$. Taking the directional derivative along any tangent direction $\xi_k \in T_{\theta_k} \Theta_k$ therefore gives identical directional derivatives in the $k$-th block; by Riesz representation under the metric $g_k$ the Riemannian gradients coincide. The stated equivalence is then immediate from Definition B.10. $\qquad\square$

Lemma B.12 is the formal content of the main text's decentralized dynamics: replacing $\text{grad}_{g_k, \theta_k} V$ by $\text{grad}_{g_k, \theta_k} J_k$ in Eq. (5) leaves the flow—and hence the Lyapunov descent of Proposition B.11—unchanged, so each client descending its local payoff monotonically decreases the global potential $V$ without explicit coordination on $V$.

# C. Appendix for the Methodology

This appendix discharges the mechanical claims of Section 3 for both solvers: the idealized *reference solver* (Part C) and its communication-feasible realization *GeoEvo* (Part C.3). We establish per-round, per-client, deterministic structural properties: the identity-aware geometry of the descent, the well-posedness of every operator, and a proven majorization that bridges the global potential $V$ and the local prototype surrogate.

**Notation.** $z_k = \mathcal{T}_k(\theta_k) \in \mathcal{Z}$; $\pi_S : \mathcal{Z} \to \mathcal{Z}_S$ restricts to support $S \subseteq \mathcal{M}$; $S_{kj} = \mathcal{M}_k \cap \mathcal{M}_j$; $\mathcal{N}_k^t = \mathcal{N}_k \cap \mathcal{S}^t$ are the participating neighbors at round $t$; $S_k^t = \bigcup_{j \in \mathcal{N}_k^t} S_{kj} \subseteq \mathcal{M}_k$ and $\bar{\omega}_k^t = \sum_{j \in \mathcal{N}_k^t} \omega_{kj}$; $\widetilde{\mathbf{F}}_k(\theta_k) = \mathbf{F}_k(\theta_k) + \epsilon \mathbf{I}$ is the damped Fisher metric (Remark 3.1); $\text{grad}_{g_k, \theta_k}$ is the Riemannian gradient under $g_k$.

**Part I. Reference Solver: Exact Potential Minimization**

### C.1. Identity-Consistent Fisher Metric and Its Block-Sparsity

The main text equips each $\Theta_k$ with the empirical Fisher metric of Eq. (8) and asserts that it "inherits the block-sparse structure of $\mathcal{M}_k$ without explicit masking." We make this precise. Adopt the modality-factorized realization of Section 3: $\theta_k = (\theta_k^{\mathrm{shr}}, \{\theta_k^{(m)}\}_{m \in \mathcal{M}})$, where $\theta_k^{(m)}$ acts on the output only through modality $m$'s input branch, and for client $k$ the branches $m \notin \mathcal{M}_k$ are inactive.

**Lemma C.1** (Score Sparsity $\Rightarrow$ Fisher Block-Sparsity). *Let $s_k(\theta_k; \mathbf{x}, y) \triangleq \nabla_{\theta_k} \log p(y \mid \mathbf{x}_{\mathcal{M}_k}; \theta_k)$ be the score. For every coordinate block $m \notin \mathcal{M}_k$, the partial score $\nabla_{\theta_k^{(m)}} \log p(y \mid \mathbf{x}_{\mathcal{M}_k}; \theta_k) \equiv 0$. Consequently the empirical Fisher $\mathbf{F}_k(\theta_k) = \frac{1}{|\mathcal{D}_k|} \sum s_k s_k^\top$ has zero rows and columns on all blocks $m \notin \mathcal{M}_k$; i.e., $\mathbf{F}_k$ is supported on the active index set $\mathcal{A}_k = \{\theta_k^{\mathrm{shr}}\} \cup \{\theta_k^{(m)} : m \in \mathcal{M}_k\}$. The damped metric $\widetilde{\mathbf{F}}_k = \mathbf{F}_k + \epsilon \mathbf{I}$ is therefore block-diagonal across the active/inactive split, so its inverse $\widetilde{\mathbf{F}}_k^{-1}$ preserves this split (acting as $\frac{1}{\epsilon}\mathbf{I}$ on the inactive block); since the descent objective ($F_k$ and $\mathcal{T}_k$) does not depend on the inactive coordinates, $\nabla_{\theta_k} V$ vanishes there and the preconditioned direction $\widetilde{\mathbf{F}}_k^{-1} \nabla_{\theta_k} V$ stays supported on $\mathcal{A}_k$.*

*Proof.* Since branch $m \notin \mathcal{M}_k$ is inactive, $p(y \mid \mathbf{x}_{\mathcal{M}_k}; \theta_k)$ does not depend on $\theta_k^{(m)}$, so $\partial_{\theta_k^{(m)}} \log p \equiv 0$ pointwise; hence those entries of $s_k$ vanish for every sample. An outer product $s_k s_k^\top$ with zero entries on block $m$ has zero rows or columns there, and the property is preserved under averaging; adding $\epsilon \mathbf{I}$ keeps $\widetilde{\mathbf{F}}_k$ block-diagonal across the active/inactive split, so $\widetilde{\mathbf{F}}_k^{-1}$ does not mix the two blocks. Restricting $\mathbf{F}_k$ to $T_{\theta_k}\Theta_k$ yields the intrinsic metric $g_k$. $\square$

Lemma C.1 is what makes the metric *identity-consistent*: the geometry never spends curvature on modalities that the client cannot observe, so the natural-gradient preconditioner $\widetilde{\mathbf{F}}_k^{-1}$ acts only on $\mathcal{M}_k$-relevant coordinates, matching the structural identity of Definition 2.4.

### C.2. Block Descent as Steepest Descent under the Fisher Metric

We justify the preconditioned step shared by Eq. (9) (reference solver) and Eq. (13) (GeoEvo).

**Lemma C.2** (Metric Steepest Descent). *Let $h : \Theta_k \to \mathbb{R}$ be differentiable, $g_k(\theta_k) \succ 0$ a metric, and $\eta > 0$. The trust-region model*

$$\Delta^\star = \arg \min_{\Delta \in T_{\theta_k}\Theta_k} \langle \nabla_{\theta_k} h(\theta_k), \Delta \rangle + \frac{1}{2\eta} \|\Delta\|_{g_k(\theta_k)}^2 \tag{39}$$

*has the closed-form minimizer $\Delta^\star = -\eta\, g_k(\theta_k)^{-1} \nabla_{\theta_k} h(\theta_k) = -\eta \operatorname{grad}_{g_k, \theta_k} h(\theta_k)$, and the retracted update $\theta_k^+ = \operatorname{Retr}_{\theta_k}(\Delta^\star)$ is a first-order approximation of the Riemannian gradient flow $\dot{\theta}_k = -\operatorname{grad}_{g_k, \theta_k} h$.*

*Proof.* Eq. (39) is a strictly convex quadratic in $\Delta$ (as $g_k \succ 0$); its stationarity condition $\nabla h + \frac{1}{\eta} g_k \Delta = 0$ gives $\Delta^\star = -\eta\, g_k^{-1} \nabla h$. By definition of the Riemannian gradient, $g_k^{-1} \nabla h = \operatorname{grad}_{g_k, \theta_k} h$. A retraction satisfies $\operatorname{Retr}_{\theta_k}(\tau \xi) = \theta_k + \tau \xi + \mathcal{O}(\tau^2)$, so $\theta_k^+ = \theta_k - \eta \operatorname{grad}_{g_k, \theta_k} h + \mathcal{O}(\eta^2)$, *i.e.* an explicit-Euler step of the gradient flow. $\square$

Instantiating $g_k = \widetilde{\mathbf{F}}_k$ in Lemma C.2 yields the natural-gradient direction $-\eta \widetilde{\mathbf{F}}_k^{-1} \nabla_{\theta_k} h$, which is steepest descent measured in the model's KL geometry. With $h = V$ this is exactly Eq. (9); with $h = \tilde{J}_k^t$ (Part C.3) it is Eq. (13). By Lemma C.1, the direction is automatically supported on $\mathcal{A}_k$, so no explicit masking is required.

### C.3. Per-Step Sufficient Decrease

We prove Lemma 3.2 in the *simultaneous* form used throughout the paper: at each iteration *all* blocks move along their own Riemannian gradient, $W^{t+1} = \left( \operatorname{Retr}_{\theta_k^t}(-\eta_t \operatorname{grad}_{g_k, \theta_k} V(W^t)) \right)_{k=1}^{K}$. This matches Eq. (9) and avoids any single-block serial convention.

**Lemma C.3** (Per-step Sufficient Decrease (Simultaneous Blocks)). *Let $V$ be $L$-smooth w.r.t. the retraction on the product manifold $(\mathcal{W}, g)$, $g = \oplus_k g_k$ (Assumption 4.2). For the simultaneous update with $0 < \eta_t \leq 1/L$,*

$$V(W^{t+1}) \leq V(W^t) - \frac{\eta_t}{2} \left\| \operatorname{grad} V(W^t) \right\|_g^2 = V(W^t) - \frac{\eta_t}{2} \sum_{k=1}^{K} \left\| \operatorname{grad}_{g_k, \theta_k} V(W^t) \right\|_{g_k}^2. \tag{40}$$

*Proof.* The simultaneous step is the single product-manifold tangent vector $\xi = -\eta_t \operatorname{grad} V(W^t) \in T_{W^t}\mathcal{W}$, with $W^{t+1} = \operatorname{Retr}_{W^t}(\xi)$. The product structure gives the orthogonal block decomposition $\|\xi\|_g^2 = \sum_k \|\xi_k\|_{g_k}^2$ and $\langle \operatorname{grad} V, \xi \rangle_g = \sum_k \langle \operatorname{grad}_{g_k, \theta_k} V, \xi_k \rangle_{g_k}$. Applying the $L$-smooth descent inequality of Assumption 4.2,

$$V(W^{t+1}) \leq V(W^t) + \langle \operatorname{grad} V(W^t), \xi \rangle_g + \frac{L}{2}\|\xi\|_g^2 = V(W^t) - \left( \eta_t - \frac{L\eta_t^2}{2} \right) \left\| \operatorname{grad} V(W^t) \right\|_g^2. \tag{41}$$

For $\eta_t \leq 1/L$ we have $\eta_t - \frac{L\eta_t^2}{2} \geq \frac{\eta_t}{2}$, giving Eq. (40); the second equality is the block decomposition of the product norm. $\square$

**Corollary C.4** (Telescoped Stationarity of the Reference Solver). *Summing Eq. (40) and using $V \geq V_{\inf}$ (Assumption 4.3(i)),*

$$\sum_{t=0}^{\infty} \frac{\eta_t}{2} \left\| \operatorname{grad} V(W^t) \right\|_g^2 \leq V(W^0) - V_{\inf} < \infty. \tag{42}$$

*Hence if $\inf_t \eta_t > 0$ then $\|\operatorname{grad} V(W^t)\|_g \to 0$, so $\|\operatorname{grad}_{g_k, \theta_k} V(W^t)\|_{g_k} \to 0$ for every $k$.*

## Part II.  GeoEvo: Federated Geometric Evolution

### C.4. Masked Structure Space and a Monotonicity Lemma

We take $\mathcal{Z} = \bigoplus_{m \in \mathcal{M}} \mathcal{Z}^{(m)}$ to be a Euclidean product, the restriction $\pi_S$ to keep the blocks indexed by $S$, and $d_{\mathcal{Z}}$ to act coordinatewise:

$$d_{\mathcal{Z}}^2 \big( \pi_S(z), \pi_S(z') \big) = \sum_{m \in S} \left\| z^{(m)} - z'^{(m)} \right\|^2, \qquad \pi_S \circ \pi_{S'} = \pi_S \ (S \subseteq S'). \tag{43}$$

By the convention $\omega_{kj} > 0 \iff S_{kj} \neq \emptyset$, no empty-overlap distance is ever evaluated.

**Lemma C.5** (Masked-Distance Monotonicity). *For $S \subseteq S' \subseteq \mathcal{M}$ and any $z, z' \in \mathcal{Z}$, $d_{\mathcal{Z}}^2(\pi_S(z), \pi_S(z')) \leq d_{\mathcal{Z}}^2(\pi_{S'}(z), \pi_{S'}(z'))$, with gap $\sum_{m \in S' \setminus S} \|z^{(m)} - z'^{(m)}\|^2 \geq 0$.*

*Proof.* Immediate from Eq. (43): enlarging the index set only adds non-negative squared terms. $\square$

Lemma C.5 formalizes the intuition that evaluating a coupling on the *union* support $S_k^t$ (rather than on a smaller per-pair overlap $S_{kj} \subseteq S_k^t$) can only *increase* the measured discrepancy—this is the geometric seed of the majorization used in §C.6.

### C.5. Server Prototype: Existence and Coordinatewise Form

The server prototype solves the Fréchet problem Eq. (11): $Z_k^t \in \arg\min_{z \in \mathcal{Z}} \sum_{j \in \mathcal{N}_k^t} \omega_{kj} d_{\mathcal{Z}}^2 \big( \pi_{S_{kj}}(z), \pi_{S_{kj}}(z_j^t) \big)$. For each $m \in S_k^t$ write the neighbors that possess $m$ and their aggregate weight as

$$\mathcal{N}_k^t(m) \triangleq \{j \in \mathcal{N}_k^t : m \in S_{kj}\}, \qquad \Omega_m^t \triangleq \sum_{j \in \mathcal{N}_k^t(m)} \omega_{kj} > 0. \tag{44}$$

**Lemma C.6** (Existence and Coordinatewise Weighted Mean). *Under Eq. (43), the Fréchet objective separates across coordinates, the minimizer exists and, on each $m \in S_k^t$, is the weighted mean of the participating neighbors,*

$$Z_k^{t,(m)} = \frac{1}{\Omega_m^t} \sum_{j \in \mathcal{N}_k^t(m)} \omega_{kj} z_j^{t,(m)}, \qquad m \in S_k^t. \tag{45}$$

*Proof.* Interchanging the finite sums, the objective equals $\sum_{m \in S_k^t} \sum_{j \in \mathcal{N}_k^t(m)} \omega_{kj} \|z^{(m)} - z_j^{t,(m)}\|^2$, a sum of independent strictly convex coordinate problems. Each is a weighted least-squares problem minimized at the weighted mean Eq. (45); coordinates outside $S_k^t$ are unconstrained and may be fixed arbitrarily. $\square$

In particular, $Z_k^t$ minimizes the exact per-pair coupling, recovering the variational optimality stated as Proposition 4.6; the present lemma supplies the constructive coordinatewise form used below.

*Remark* C.7 (Per-Round Communication Complexity). Each round the server returns to client $k$ a single triple $(Z_k^t, S_k^t, \bar{\omega}_k^t)$ whose payload is one structural vector supported on $S_k^t$ plus a scalar weight, *i.e.* $\mathcal{O}(|S_k^t|) \leq \mathcal{O}(|\mathcal{M}_k|)$ reals in *one* message, and the uplink is symmetric (each participating client sends only $z_k^{t+1} = \mathcal{T}_k(\theta_k)$). Hence the aggregate per-round traffic is $\mathcal{O}(|\mathcal{S}^t|) = \mathcal{O}(K)$ messages, independent of the neighborhood sizes $\{|\mathcal{N}_k^t|\}$, whereas the reference solver of Part C must exchange a representation along every active edge, incurring $\sum_k |\mathcal{N}_k^t| = \mathcal{O}(K^2)$ messages in the dense-coupling regime.

## C.6. Prototype Compression as a Proven Geometric Majorization

This subsection establishes why optimizing the single-prototype surrogate serves as a principled proxy for the true many-neighbor coupling. We prove a majorization inequality that decomposes the approximation gap into two distinct non-negative sources: neighbor dispersion and weight over-counting. Consequently, the deviation between the "per-pair mask vs. union mask" is fully accounted for.

Define the *exact per-pair coupling* (the $\theta_k$-dependent block of $V$ with neighbors frozen at their round-$t$ uploads; matching $\Phi_k^t$ of Sec. 4) and the *prototype surrogate coupling* (matching $\widetilde{\Phi}_k^t$, the coupling part of Eq. (12)):

$$\Phi_k^t(z) \triangleq \frac{\lambda}{2} \sum_{j \in \mathcal{N}_k^t} \omega_{kj} \, d_{\mathcal{Z}}^2 \big( \pi_{S_{kj}}(z), \pi_{S_{kj}}(z_j^t) \big), \tag{46}$$

$$\widetilde{\Phi}_k^t(z) \triangleq \frac{\lambda}{2} \bar{\omega}_k^t \, d_{\mathcal{Z}}^2 \big( \pi_{S_k^t}(z), \pi_{S_k^t}(Z_k^t) \big). \tag{47}$$

**Lemma C.8** (Bias–Variance (Parallel-Axis) Decomposition of the Exact Coupling). *With $Z_k^t$ from Eq. (45),*

$$\Phi_k^t(z) = \underbrace{\frac{\lambda}{2} \sum_{m \in S_k^t} \Omega_m^t \big\| z^{(m)} - Z_k^{t,(m)} \big\|^2}_{\text{z-dependent (bias)}} + \underbrace{\frac{\lambda}{2} \sum_{m \in S_k^t} \nu_m^t}_{\text{z-independent (dispersion) } =: C_k^t}, \qquad \nu_m^t \triangleq \sum_{j \in \mathcal{N}_k^t(m)} \omega_{kj} \big\| z_j^{t,(m)} - Z_k^{t,(m)} \big\|^2. \tag{48}$$

*Proof.* Coordinatewise separation as in Lemma C.6, then the weighted parallel-axis identity $\sum_j \omega_j \|x - a_j\|^2 = \Omega \|x - \bar{a}\|^2 + \sum_j \omega_j \|a_j - \bar{a}\|^2$ with $\bar{a} = \Omega^{-1} \sum_j \omega_j a_j$, applied per coordinate with $x = z^{(m)}$, $a_j = z_j^{t,(m)}$, $\Omega = \Omega_m^t$, $\bar{a} = Z_k^{t,(m)}$. $\square$

**Theorem C.9** (Prototype Majorization). *For every $z \in \mathcal{Z}$ (hence in particular on $\text{Im}(\mathcal{T}_k)$, for $z = \mathcal{T}_k(\theta_k)$),*

$$\Phi_k^t(z) \leq \widetilde{\Phi}_k^t(z) + C_k^t, \qquad C_k^t = \frac{\lambda}{2} \sum_{m \in S_k^t} \nu_m^t \geq 0, \tag{49}$$

*with $C_k^t$ independent of $z$. Thus, the union-mask prototype coupling $\widetilde{\Phi}_k^t$ majorizes the exact per-pair coupling $\Phi_k^t$ up to the neighbor-dispersion constant $C_k^t$. The bound is tight ($C_k^t = 0$ and equality of the z-dependent parts) iff all neighbors agree on every shared coordinate and share the full union support, i.e. $z_j^{t,(m)} = Z_k^{t,(m)}$ and $\Omega_m^t = \bar{\omega}_k^t$ for all $m \in S_k^t$.*

*Proof.* By Lemma C.8, $\Phi_k^t(z) - C_k^t = \frac{\lambda}{2} \sum_{m \in S_k^t} \Omega_m^t \|z^{(m)} - Z_k^{t,(m)}\|^2$. Each per-coordinate weight obeys $\Omega_m^t = \sum_{j : m \in S_{kj}} \omega_{kj} \leq \sum_{j \in \mathcal{N}_k^t} \omega_{kj} = \bar{\omega}_k^t$, because $\mathcal{N}_k^t(m) \subseteq \mathcal{N}_k^t$. Hence, term by term, $\sum_m \Omega_m^t \|z^{(m)} - Z_k^{t,(m)}\|^2 \leq \bar{\omega}_k^t \sum_m \|z^{(m)} - Z_k^{t,(m)}\|^2 = d_{\mathcal{Z}}^2(\pi_{S_k^t}(z), \pi_{S_k^t}(Z_k^t)) \cdot 1$, so $\Phi_k^t(z) - C_k^t \leq \widetilde{\Phi}_k^t(z)$. Equality forces $\Omega_m^t = \bar{\omega}_k^t$ on every active coordinate (no weight over-counting) and $C_k^t = 0$ (no dispersion). $\square$

Theorem C.9 establishes the precise relationship between the exact per-pair coupling and its single-prototype surrogate. The gap $\widetilde{\Phi}_k^t - (\Phi_k^t - C_k^t) = \frac{\lambda}{2} \sum_m (\bar{\omega}_k^t - \Omega_m^t) \|z^{(m)} - Z_k^{t,(m)}\|^2$ decomposes into (i) a *weight-over-counting* part $\bar{\omega}_k^t - \Omega_m^t \geq 0$

(the union mask applies the aggregate weight to coordinates owned by only some neighbors) and (ii) the *dispersion* constant $C_k^t$ (neighbor disagreement on shared coordinates). Both are non-negative, so $\widetilde{\Phi}_k^t$ is a conservative (majorizing) compression—never an under-estimate of the true coupling beyond the additive constant.

**Proposition C.10** (Surrogate Descent Controls the Exact Per-Pair Objective). *Let* $g_k^t(\theta) \triangleq w_k F_k(\theta; \mathcal{I}_k) + \Phi_k^t(\mathcal{T}_k(\theta))$ *be the exact per-pair objective and* $\tilde{J}_k^t(\theta) = w_k F_k(\theta; \mathcal{I}_k) + \widetilde{\Phi}_k^t(\mathcal{T}_k(\theta))$ *the prototype surrogate (Eq.* (12)*). If an update satisfies* $\tilde{J}_k^t(\theta_k^+) \leq \tilde{J}_k^t(\theta_k^{\mathrm{cur}})$, *then*

$$g_k^t(\theta_k^+) \ \leq \ g_k^t(\theta_k^{\mathrm{cur}}) + \rho_k^t, \qquad \rho_k^t \triangleq \frac{\lambda}{2} \sum_{m \in S_k^t} (\bar{\omega}_k^t - \Omega_m^t) \big\| z_k^{\mathrm{cur},(m)} - Z_k^{t,(m)} \big\|^2 \ \geq \ 0, \tag{50}$$

*where* $z_k^{\mathrm{cur}} = \mathcal{T}_k(\theta_k^{\mathrm{cur}})$. *If overlaps are homogeneous* ($S_{kj} = S_k^t$ *for all* $j \in \mathcal{N}_k^t$, *hence* $\Omega_m^t = \bar{\omega}_k^t$), *then* $\rho_k^t = 0$ *and surrogate descent is* exact *descent of* $g_k^t$.

*Proof.* Using majorization at $\theta_k^+$, then surrogate descent, then the decomposition Eq. (48) at $\theta_k^{\mathrm{cur}}$:

$$g_k^t(\theta_k^+) \leq w_k F_k(\theta_k^+) + \widetilde{\Phi}_k^t(\mathcal{T}_k(\theta_k^+)) + C_k^t = \tilde{J}_k^t(\theta_k^+) + C_k^t \ \leq \ \tilde{J}_k^t(\theta_k^{\mathrm{cur}}) + C_k^t$$
$$= w_k F_k(\theta_k^{\mathrm{cur}}) + \widetilde{\Phi}_k^t(z_k^{\mathrm{cur}}) + C_k^t = g_k^t(\theta_k^{\mathrm{cur}}) + \Big[ \widetilde{\Phi}_k^t(z_k^{\mathrm{cur}}) + C_k^t - \Phi_k^t(z_k^{\mathrm{cur}}) \Big] = g_k^t(\theta_k^{\mathrm{cur}}) + \rho_k^t,$$

where the bracket equals $\rho_k^t$ by Lemma C.8 and Theorem C.9. $\qquad \square$

*Remark* C.11 (Closure of the $V$-to-Surrogate Gap). Proposition C.10 is a genuine majorize–minimize statement: it does not claim surrogate descent equals descent of $g_k^t$ in general, but bounds the slippage by an explicit residual $\rho_k^t \geq 0$ that vanishes under homogeneous overlaps. Since $g_k^t$ is exactly the $\theta_k$-block of $V$ with neighbors frozen at $\{z_j^t\}$, this provides the per-round, per-client link to the global potential.

## C.7. Structure-Matching Lift: Well-Posedness and Robustness

The lift Eq. (14) is $\mathrm{Lift}_k(z; \theta_k) \in \arg\min_{\theta \in \Theta_k} d_{\mathcal{Z}}^2(\mathcal{T}_k(\theta), z)$.

**Lemma C.12** (Existence of the Lift). *If* $\mathcal{T}_k$ *is continuous and* $\Theta_k$ *is compact (or* $d_{\mathcal{Z}}^2(\mathcal{T}_k(\cdot), z)$ *is coercive on* $\Theta_k$*), then the lift minimizer exists for every target* $z$.

*Proof.* $\theta \mapsto d_{\mathcal{Z}}^2(\mathcal{T}_k(\theta), z)$ is continuous (composition of continuous maps) and is minimized over a compact set, or is coercive with closed sublevel sets; in either case, the Weierstrass extreme-value theorem guarantees a minimizer. $\qquad \square$

*Remark* C.13 (Coercivity, Compactness, and the Role of Approximate Lifts). The coercivity alternative requires careful handling. When $\mathcal{T}_k$ has bounded range (*e.g.*, a squashing output layer), $\theta \mapsto d_{\mathcal{Z}}^2(\mathcal{T}_k(\theta), z)$ is bounded and hence not coercive; on a non-compact $\Theta_k$ the infimum may not be attained. The reliable hypothesis is therefore compactness of $\Theta_k$, as already stated in Definition 2.4. Crucially, GeoEvo never computes the exact minimizer: the lift is solved approximately by a few warm-started retraction steps, and by Proposition C.14 the monotone-acceptance guarantee—and thus all subsequent convergence results—holds for any such approximate lift. The existence of an exact $\arg\min$ is a property of the idealized operator, not a prerequisite for the analysis.

**Proposition C.14** (Approximate Lift Preserves Monotone Acceptance). *Let* $\theta_k^{\mathrm{NES}}, \theta_k^{\mathrm{PSO}}$ *be produced by any lift, e.g. a few warm-started retracted steps from* $\theta_k$. *If the client selects* $\theta_k^+$ *by Eq.* (17)*, then* $\tilde{J}_k^t(\theta_k^+) \leq \tilde{J}_k^t(\theta_k^{\mathrm{cur}})$.

*Proof.* Selection evaluates the *exact* surrogate $\tilde{J}_k^t$ on the already-lifted candidates and includes $\theta_k^{\mathrm{cur}}$ in the pool; the $\arg\min$ over a set containing $\theta_k^{\mathrm{cur}}$ cannot exceed $\tilde{J}_k^t(\theta_k^{\mathrm{cur}})$, regardless of how the candidates were generated. Inexactness of the lift only affects candidate *quality*, never the monotonicity guarantee. $\qquad \square$

## C.8. NES in Structure Space: Unbiased Smoothed Gradient

With $\phi_k(z) \triangleq \tilde{J}_k^t(\mathrm{Lift}_k(z; \theta_k))$ and the Gaussian-smoothed objective $\phi_{k,\sigma_z}(z) \triangleq \mathbb{E}_{\varepsilon \sim \mathcal{N}(0,\mathbf{I})}[\phi_k(z + \sigma_z \varepsilon)]$, the estimator in Eq. (15) is $\widehat{\nabla}_z \phi_{k,\sigma_z}(z) = \frac{1}{m\sigma_z} \sum_{i=1}^m \phi_k(z + \sigma_z \varepsilon_i) \varepsilon_i$.

**Lemma C.15** (Unbiasedness of the NES Estimator). *If $\phi_k$ is measurable with $\mathbb{E}[|\phi_k(z + \sigma_z \varepsilon)| \, \|\varepsilon\|] < \infty$, then $\mathbb{E}[\widehat{\nabla}_z \phi_{k,\sigma_z}(z)] = \nabla_z \phi_{k,\sigma_z}(z)$.*

*Proof.* By Stein's identity for $\varepsilon \sim \mathcal{N}(0, \mathbf{I})$, $\mathbb{E}[f(\varepsilon)\varepsilon] = \mathbb{E}[\nabla f(\varepsilon)]$. Taking $f(\varepsilon) = \phi_k(z + \sigma_z \varepsilon)$ gives $\nabla_\varepsilon f = \sigma_z \nabla_z \phi_k(z + \sigma_z \varepsilon)$, hence $\mathbb{E}[\phi_k(z + \sigma_z \varepsilon)\varepsilon] = \sigma_z \mathbb{E}[\nabla_z \phi_k(z + \sigma_z \varepsilon)] = \sigma_z \nabla_z \phi_{k,\sigma_z}(z)$. Dividing by $\sigma_z$ and averaging over $m$ samples gives $\mathbb{E}[\widehat{\nabla}_z \phi_{k,\sigma_z}(z)] = \nabla_z \phi_{k,\sigma_z}(z)$. $\qquad\square$

*Remark* C.16. The smoothing radius $\sigma_z$ trades estimator variance against bias relative to $\nabla_z \phi_k$ ($\phi_{k,\sigma_z} \to \phi_k$ as $\sigma_z \downarrow 0$). NES proposes candidates in the lower-dimensional, identity-aligned space $\mathcal{Z}$, enabling moves outside the basin of the exploitation step; monotonicity is never at risk because such proposals are only *accepted* through Eq. (17).

## C.9. PSO in Structure Space: A Proposal Generator

The PSO update Eq. (16) sets the broadcast prototype $Z_k^t$ as the global attractor, embedding cross-client structural coordination into the swarm.

*Remark* C.17 (Role and Safety). We do not require PSO to be a convergent or contractive dynamical system. In GeoEvo, it is purely a *proposal mechanism* in $\mathcal{Z}$: candidates are lifted to $\Theta_k$ and then filtered by the surrogate. By Proposition C.14, any PSO proposal leaves the monotone acceptance guarantee intact, so global coordination toward $Z_k^t$ is obtained without imposing stability conditions on the swarm parameters $(c_0, c_1, c_2)$.

## C.10. Monotone Selection and Round-wise Surrogate Dissipation

**Proposition C.18** (Surrogate Monotonicity under Acceptance). *Fix round $t$ and treat $(Z_k^t, S_k^t, \bar\omega_k^t)$ as constants, so $\tilde J_k^t$ is a fixed function. With $\theta_k^+$ chosen by Eq. (17) from a pool containing $\theta_k^{\mathrm{cur}}$, $\tilde J_k^t(\theta_k^+) \le \tilde J_k^t(\theta_k^{\mathrm{cur}})$.*

*Proof.* $\theta_k^+$ minimizes $\tilde J_k^t$ over a finite candidate set that contains $\theta_k^{\mathrm{cur}}$; the minimum over a set is no larger than the value at any of its members. $\qquad\square$

**Theorem C.19** (Round-Wise Surrogate Dissipation). *Fix round $t$ and client $k \in \mathcal{S}^t$, and hold $(Z_k^t, S_k^t, \bar\omega_k^t)$ fixed during $E$ local steps. If each step applies Eq. (17), the surrogate sequence is non-increasing: $\tilde J_k^t(\theta_k^{t,s+1}) \le \tilde J_k^t(\theta_k^{t,s})$ for $s = 0, 1, \ldots, E - 1$, where $\theta_k^{t,0}$ is the round-$t$ initialization and $\theta_k^{t,E}$ the uploaded model. $\tilde J_k^t(\theta_k^{t,E}) \le \tilde J_k^t(\theta_k^{t,0})$, and by Proposition C.10 the exact per-pair objective obeys $g_k^t(\theta_k^{t,E}) \le g_k^t(\theta_k^{t,0}) + \sum_{s=0}^{E-1} \rho_k^{t,s}$ with $\rho_k^{t,s} \ge 0$ given by Eq. (50) at $\theta_k^{t,s}$.*

*Proof.* Apply Proposition C.18 at each local step with $\theta_k^{\mathrm{cur}} = \theta_k^{t,s}$ and $\theta_k^+ = \theta_k^{t,s+1}$; telescoping over $s$ gives the round inequality. The per-pair consequence chains Proposition C.10 over the $E$ steps, so the residuals add. $\qquad\square$

# D. Appendix for the Theoretical Analysis

This appendix proves the convergence results of Section 4, covering both the idealized reference solver and its federated realization. The analysis includes stationarity of simultaneous descent, the surrogate error budget $\delta_k^t$, the $\mathcal{O}(1/\sqrt{T})$ convergence rate, and the lift to the global potential $V$.

**Notation.** $\mathcal{W} = \prod_k \Theta_k$ carries the product metric $g = \oplus_k g_k$; $\operatorname{grad} V$ is the Riemannian gradient on $(\mathcal{W}, g)$ and $\operatorname{grad}_{g_k, \theta_k} V$ its $k$-th block. At round $t$, $\mathcal{S}^t$ is the participating set, $\mathcal{N}_k^t = \mathcal{N}_k \cap \mathcal{S}^t$, $S_k^t = \bigcup_{j \in \mathcal{N}_k^t} S_{kj} \subseteq \mathcal{M}_k$, $\bar\omega_k^t = \sum_{j \in \mathcal{N}_k^t} \omega_{kj}$, and $z_j^t = \mathcal{T}_j(\theta_j^t)$. The coordinatewise quantities are: $\Omega_m^t = \sum_{j \in \mathcal{N}_k^t(m)} \omega_{kj}$, $\nu_m^t = \sum_{j \in \mathcal{N}_k^t(m)} \omega_{kj} \|z_j^{t,(m)} - Z_k^{t,(m)}\|^2$, and $C_k^t = \frac{\lambda}{2} \sum_{m \in S_k^t} \nu_m^t$.

## D.1. Standing Assumptions

We work under Assumptions 4.1–4.3 (geometry/regularity, $L$-smoothness w.r.t. retraction of both $V$ and every round-$t$ surrogate $\tilde J_k^t$, and lower boundedness with the optional compact sublevel set), and—for the stochastic results—Assumptions 4.8 (unbiased, variance-$\sigma^2$ gradients) and 4.9 (summable prototype drift $\Delta_T$). We also adopt the coordinatewise masked-Euclidean structure of Eq. (43), under which $\pi_S \circ \pi_{S'} = \pi_S$ for $S \subseteq S'$.

**Second-Order Retraction** The $L$-smoothness of Assumption 4.2 is a quadratic upper bound on the pullback $V \circ \mathrm{Retr}$ (resp. $\tilde{J}_k^t \circ \mathrm{Retr}$). Mere $C^1$ regularity of $F_k$ and $\mathcal{T}_k$ does *not* furnish such a bound; the standard sufficient regularity, which we make explicit, is that $\mathrm{Retr}$ is a *second-order retraction* on $(\mathcal{W}, g)$ and that $F_k(\cdot; \mathcal{I}_k)$ and $\mathcal{T}_k$ are *twice* continuously differentiable. On the compact sublevel set (Assumption 4.3(ii)) the pullback then has a bounded Riemannian Hessian, and $L$ may be taken as the supremum of its operator norm over that set; this is what licenses the "inherited from the differentiability of $F_k, \mathcal{T}_k$" clause of Assumption 4.2. (Alternatively, the inequality may be posited as a primitive hypothesis; the second-order retraction is precisely what makes it derivable from the $C^2$ data rather than assumed.)

## D.2. Exact Potential and Stationarity of the Reference Solver

**Exact Potential.** Proposition 4.4 restates that $V$ of Eq. (3) is an exact potential for the game Eq. (2); its proof is given in Proposition B.9. We use only its corollary, the blockwise gradient alignment $\mathrm{grad}_{g_k, \theta_k} J_k(\theta_k; W_{-k}) = \mathrm{grad}_{g_k, \theta_k} V(W)$ (Lemma B.12), which lets us analyze $V$ directly.

**Per-Step Decrease.** The reference solver is the simultaneous retracted update of Eq. (9): at each iteration *all* blocks move along their own Riemannian gradient,

$$W^{t+1} = \left( \mathrm{Retr}_{\theta_k^t}\big( -\eta_t \, \mathrm{grad}_{g_k, \theta_k} V(W^t) \big) \right)_{k=1}^{K}, \tag{51}$$

which is the single product-manifold step $W^{t+1} = \mathrm{Retr}_{W^t}(-\eta_t \, \mathrm{grad}\, V(W^t))$. Lemma C.3 gives, for $0 < \eta_t \leq 1/L$,

$$V(W^{t+1}) \leq V(W^t) - \frac{\eta_t}{2} \left\| \mathrm{grad}\, V(W^t) \right\|_g^2 = V(W^t) - \frac{\eta_t}{2} \sum_{k=1}^{K} \left\| \mathrm{grad}_{g_k, \theta_k} V(W^t) \right\|_{g_k}^2. \tag{52}$$

We stress that Eq. (52) is obtained from the *joint* step (51), in which no block is held fixed; this is the all-block form required for consistency with the main text.

**Theorem D.1** (Stationarity Convergence). *Under Assumptions 4.2–4.3, the iterates of the simultaneous update (51) with $0 < \eta_t \leq 1/L$ satisfy: (i) $\{V(W^t)\}$ is non-increasing; (ii) $\sum_{t=0}^{\infty} \eta_t \|\mathrm{grad}\, V(W^t)\|_g^2 \leq 2\big(V(W^0) - V_{\inf}\big) < \infty$; (iii) if $\inf_t \eta_t > 0$ then $\|\mathrm{grad}\, V(W^t)\|_g \to 0$, hence $\|\mathrm{grad}_{g_k, \theta_k} V(W^t)\|_{g_k} \to 0$ for every $k$ simultaneously; (iv) under the compact sublevel set (Assumption 4.3(ii)) and continuity of $\mathrm{grad}\, V$ (Assumption 4.1), every limit point of $\{W^t\}$ is a first-order Nash equilibrium.*

*Proof.* (i) is immediate from Eq. (52), whose right side is $\leq V(W^t)$. (ii): summing Eq. (52) over $t = 0, \ldots, T-1$ telescopes the left side, $\sum_{t=0}^{T-1} \frac{\eta_t}{2} \|\mathrm{grad}\, V(W^t)\|_g^2 \leq V(W^0) - V(W^T) \leq V(W^0) - V_{\inf}$, using $V \geq V_{\inf}$ (Assumption 4.3(i)); let $T \to \infty$. (iii): if $\inf_t \eta_t = \eta > 0$, then $\eta \sum_t \|\mathrm{grad}\, V(W^t)\|_g^2 \leq 2(V(W^0) - V_{\inf}) < \infty$, so the summand $\to 0$; the block identity $\|\mathrm{grad}\, V\|_g^2 = \sum_k \|\mathrm{grad}_{g_k, \theta_k} V\|_{g_k}^2$ forces *every* block gradient to vanish in the limit. (iv): by (i) the iterates remain in the compact sublevel set, so a convergent subsequence $W^{t_\ell} \to W^\infty$ exists; continuity of $\mathrm{grad}\, V$ and (iii) give $\mathrm{grad}\, V(W^\infty) = 0$, *i.e.* $W^\infty$ is a first-order Nash equilibrium (Definition B.10, via Lemma B.12). $\square$

*Remark* D.2 (Reference Solver vs. GeoEvo). Theorem D.1 describes the idealized solver that evaluates the exact $\mathrm{grad}\, V$ and therefore needs per-step neighbor synchronization. GeoEvo replaces this exact coupling by stale broadcast prototypes under partial participation; the relevant federated guarantees are the surrogate-stationarity rate (Theorem D.7) and the lift to $V$ (Proposition D.9).

## D.3. Optimal Surrogate Approximation and the Error-Bound Lemma

Recall the exact per-pair coupling and the single-prototype surrogate coupling,

$$\Phi_k^t(z) \triangleq \frac{\lambda}{2} \sum_{j \in \mathcal{N}_k^t} \omega_{kj} \, d_{\mathcal{Z}}^2\big( \pi_{S_{kj}}(z), \pi_{S_{kj}}(z_j^t) \big), \tag{53}$$

$$\widetilde{\Phi}_k^t(z) \triangleq \frac{\lambda}{2} \bar{\omega}_k^t \, d_{\mathcal{Z}}^2\big( \pi_{S_k^t}(z), \pi_{S_k^t}(Z_k^t) \big), \tag{54}$$

so that the client surrogate is $\tilde{J}_k^t(\theta_k) = w_k F_k(\theta_k; \mathcal{I}_k) + \widetilde{\Phi}_k^t(\mathcal{T}_k(\theta_k))$ (Eq. (12)). We write $\Phi_k^t$ for what was informally called $\Phi_k^{t,\mathrm{true}}$: it is the genuine per-pair object, *not* the union-mask proxy.

**Proposition D.3** (Variational Optimality of the Prototype). *The Fréchet prototype $Z_k^t$ of Eq. (11) minimizes the exact per-pair coupling: $\Phi_k^t(Z_k^t) \le \Phi_k^t(z)$ for all $z \in \mathcal{Z}$, with $\Phi_k^t(Z_k^t) = C_k^t$. Coordinatewise, $Z_k^{t,(m)}$ is the weighted Fréchet mean of the neighbors possessing $m$.*

*Proof.* By the bias–variance decomposition Lemma C.8, $\Phi_k^t(z) = \frac{\lambda}{2} \sum_{m \in S_k^t} \Omega_m^t \|z^{(m)} - Z_k^{t,(m)}\|^2 + C_k^t$. The first non-negative term is minimized, and equals 0, exactly at $z^{(m)} = Z_k^{t,(m)}$ for every $m \in S_k^t$; hence $Z_k^t$ minimizes $\Phi_k^t$ with value $C_k^t$. The coordinatewise weighted-mean form is Eq. (45). $\qquad\square$

We now formalize Assumption 4.7: rather than assuming a bound $\delta_k^t$, we derive a two-sided bound and identify its two distinct sources.

**Lemma D.4** (Surrogate Error Bound). *Let $R_k^t \triangleq \sup\{ d_{\mathcal{Z}}(\pi_{S_k^t}(\mathcal{T}_k(\theta_k)), \pi_{S_k^t}(Z_k^t)) : W \in \mathcal{L}(V(W^0)) \} < \infty$ be the structural radius over the compact sublevel set (Assumption 4.3(ii)), and $\Omega_k^{t,\min} \triangleq \min_{m \in S_k^t} \Omega_m^t$. Then for all admissible $\theta_k$,*

$$\left| \Phi_k^t(\mathcal{T}_k(\theta_k)) - \widetilde{\Phi}_k^t(\mathcal{T}_k(\theta_k)) \right| \le \delta_k^t, \qquad \delta_k^t \triangleq \underbrace{C_k^t}_{\text{dispersion}} + \underbrace{\frac{\lambda}{2}\left(\bar{\omega}_k^t - \Omega_k^{t,\min}\right)(R_k^t)^2}_{\text{per-pair}\rightarrow\text{union-mask collapse}}. \tag{55}$$

*The mask-collapse term vanishes iff overlaps are homogeneous ($S_{kj} = S_k^t$ for all $j \in \mathcal{N}_k^t$, hence $\Omega_m^t = \bar{\omega}_k^t$); the dispersion term vanishes iff all neighbors agree on every shared coordinate ($\nu_m^t = 0$).*

*Proof.* Write $D_m \triangleq \|z^{(m)} - Z_k^{t,(m)}\|^2$ with $z = \mathcal{T}_k(\theta_k)$, so $\sum_{m \in S_k^t} D_m = d_{\mathcal{Z}}^2(\pi_{S_k^t}(z), \pi_{S_k^t}(Z_k^t)) \le (R_k^t)^2$. By Lemma C.8 and Eq. (54),

$$\Phi_k^t(z) - \widetilde{\Phi}_k^t(z) = \left(\frac{\lambda}{2}\sum_m \Omega_m^t D_m + C_k^t\right) - \frac{\lambda}{2}\bar{\omega}_k^t \sum_m D_m = C_k^t - \frac{\lambda}{2}\sum_{m \in S_k^t}\left(\bar{\omega}_k^t - \Omega_m^t\right)D_m. \tag{56}$$

Since $0 \le \Omega_m^t \le \bar{\omega}_k^t$, the subtracted term lies in $\left[0, \frac{\lambda}{2}(\bar{\omega}_k^t - \Omega_k^{t,\min})\sum_m D_m\right] \subseteq \left[0, \frac{\lambda}{2}(\bar{\omega}_k^t - \Omega_k^{t,\min})(R_k^t)^2\right]$, using $\sum_m (\bar{\omega}_k^t - \Omega_m^t)D_m \le (\bar{\omega}_k^t - \Omega_k^{t,\min})\sum_m D_m$. Hence the difference lies in $\left[C_k^t - \frac{\lambda}{2}(\bar{\omega}_k^t - \Omega_k^{t,\min})(R_k^t)^2, C_k^t\right]$, and its absolute value is at most $C_k^t + \frac{\lambda}{2}(\bar{\omega}_k^t - \Omega_k^{t,\min})(R_k^t)^2 = \delta_k^t$. $\qquad\square$

*Remark D.5* (Explicit Decomposition of the Surrogate Error $\delta_k^t$). Earlier drafts bounded $\widetilde{\Phi}_k^t$ against a union-mask object, capturing only neighbor compression ($C_k^t$) and silently ignoring the change of mask from the edge-specific $\pi_{S_{kj}}$ to the union $\pi_{S_k^t}$. Lemma D.4 compares $\widetilde{\Phi}_k^t$ against the genuine per-pair coupling $\Phi_k^t$ (Eq. (53)) and isolates the missing geometry as the mask-collapse term $\frac{\lambda}{2}(\bar{\omega}_k^t - \Omega_k^{t,\min})(R_k^t)^2$, which is non-negative by the masked-distance monotonicity (Lemma C.5) and the weight ordering $\Omega_m^t \le \bar{\omega}_k^t$. Both sources are explicit, and each has a clean "vanishing" regime, so $\delta_k^t$ is a derived quantity rather than an opaque assumption.

## D.4. Stochastic Convergence to Surrogate Stationarity

We use the participating-average surrogate $\widetilde{J}^t \triangleq \frac{1}{|\mathcal{S}^t|}\sum_{k \in \mathcal{S}^t} \widetilde{J}_k^t(\theta_k^t)$ and $\widetilde{J}_{\inf} \triangleq \inf_t \mathbb{E}[\widetilde{J}^t]$, and write $\langle X_k \rangle_t \triangleq \frac{1}{|\mathcal{S}^t|}\sum_{k \in \mathcal{S}^t} X_k$.

**Lemma D.6** (Filtered Riemannian-SGD Descent of the Surrogate). *Fix round $t$, client $k \in \mathcal{S}^t$ and the message $(Z_k^t, S_k^t, \bar{\omega}_k^t)$. Let $\theta_k^{\text{base}}$ be the unbiased stochastic retracted step on $\widetilde{J}_k^t$ with $\eta \le 1/L$ (Assumptions 4.2, 4.8), and let $\theta_k^{t+1}$ be selected by Eq. (17) from a pool containing $\theta_k^{\text{base}}$. Then*

$$\mathbb{E}\left[\widetilde{J}_k^t(\theta_k^{t+1}) \,\middle|\, \theta_k^t\right] \le \widetilde{J}_k^t(\theta_k^t) - \frac{\eta}{2}\left\|\text{grad}_{g_k,\theta_k^t}\widetilde{J}_k^t\right\|_{g_k}^2 + \frac{L\eta^2}{2}\sigma^2. \tag{57}$$

*Proof.* By $L$-smoothness of $\widetilde{J}_k^t$ w.r.t. Retr and the step $\theta_k^{\text{base}} = \text{Retr}_{\theta_k^t}(-\eta \widehat{\text{grad}}_{g_k,\theta_k^t}\widetilde{J}_k^t)$,

$$\widetilde{J}_k^t(\theta_k^{\text{base}}) \le \widetilde{J}_k^t(\theta_k^t) - \eta\langle\text{grad}_{g_k,\theta_k^t}\widetilde{J}_k^t, \widehat{\text{grad}}_{g_k,\theta_k^t}\widetilde{J}_k^t\rangle_{g_k} + \frac{L\eta^2}{2}\left\|\widehat{\text{grad}}_{g_k,\theta_k^t}\widetilde{J}_k^t\right\|_{g_k}^2.$$

Taking $\mathbb{E}[\cdot \mid \theta_k^t]$ and using unbiasedness and $\mathbb{E}\|\widehat{\mathrm{grad}}_{g_k,\theta_k^t}\tilde{J}_k^t\|_{g_k}^2 = \|\mathrm{grad}_{g_k,\theta_k^t}\tilde{J}_k^t\|_{g_k}^2 + \mathrm{Var} \leq \|\mathrm{grad}_{g_k,\theta_k^t}\tilde{J}_k^t\|_{g_k}^2 + \sigma^2$,

$$\mathbb{E}[\tilde{J}_k^t(\theta_k^{\mathrm{base}}) \mid \theta_k^t] \leq \tilde{J}_k^t(\theta_k^t) - \eta\Big(1 - \tfrac{L\eta}{2}\Big)\|\mathrm{grad}_{g_k,\theta_k^t}\tilde{J}_k^t\|_{g_k}^2 + \tfrac{L\eta^2}{2}\sigma^2 \leq \tilde{J}_k^t(\theta_k^t) - \tfrac{\eta}{2}\|\mathrm{grad}_{g_k,\theta_k^t}\tilde{J}_k^t\|_{g_k}^2 + \tfrac{L\eta^2}{2}\sigma^2,$$

where $\eta \leq 1/L$ gives $1 - \tfrac{L\eta}{2} \geq \tfrac{1}{2}$. Finally, the selection rule retains $\theta_k^{\mathrm{base}}$ in the candidate pool and takes the exact $\arg\min$ of $\tilde{J}_k^t$, so $\tilde{J}_k^t(\theta_k^{t+1}) \leq \tilde{J}_k^t(\theta_k^{\mathrm{base}})$ for *every* noise realization; taking expectations preserves Eq. (57). Thus, the monotone filter never degrades the pure-SGD bound. □

**Theorem D.7** (Non-Convex Rate to Surrogate Stationarity)**.** *Under Assumptions 4.2, 4.8, 4.9, the GeoEvo iterates with* $\eta \leq 1/L$ *satisfy, with* $C = 2$,

$$\min_{t<T} \mathbb{E}\Big[\big\langle \|\mathrm{grad}_{g_k}\tilde{J}_k^t(\theta_k^t)\|_{g_k}^2 \big\rangle_t\Big] \leq \frac{C\big(\mathbb{E}[\tilde{J}^0] - \tilde{J}_{\inf} + \Delta_T\big)}{\eta T} + CL\eta\sigma^2. \tag{58}$$

*Proof.* Average Eq. (57) over $k \in \mathcal{S}^t$ and rearrange:

$$\frac{\eta}{2}\mathbb{E}\big\langle \|\mathrm{grad}_{g_k}\tilde{J}_k^t(\theta_k^t)\|^2 \big\rangle_t \leq \mathbb{E}\big\langle \tilde{J}_k^t(\theta_k^t)\big\rangle_t - \mathbb{E}\big\langle \tilde{J}_k^t(\theta_k^{t+1})\big\rangle_t + \frac{L\eta^2}{2}\sigma^2. \tag{59}$$

The first term is $\mathbb{E}[\tilde{J}^t]$. For the second, the round-$(t+1)$ surrogate differs from the round-$t$ surrogate at the same iterate by the prototype re-anchoring, so $\big\langle \tilde{J}_k^t(\theta_k^{t+1})\big\rangle_t \geq \tilde{J}^{t+1} - \xi_t$, where $\xi_t \triangleq \big\langle \tilde{J}_k^{t+1}(\theta_k^{t+1}) - \tilde{J}_k^t(\theta_k^{t+1})\big\rangle$ is the per-round drift bounded in aggregate by Assumption 4.9: $\sum_t \mathbb{E}[\xi_t] \leq \Delta_T$, the drift is controlled by $\sum_t \|\pi_{S_k^t}(Z_k^t) - \pi_{S_k^{t-1}}(Z_k^{t-1})\|$ via Lipschitzness of $d_{\mathcal{Z}}^2$. Substituting into Eq. (59) and summing $t = 0, \ldots, T-1$ telescopes:

$$\frac{\eta}{2}\sum_{t=0}^{T-1} \mathbb{E}\big\langle \|\mathrm{grad}_{g_k}\tilde{J}_k^t(\theta_k^t)\|^2 \big\rangle_t \leq \mathbb{E}[\tilde{J}^0] - \mathbb{E}[\tilde{J}^T] + \Delta_T + \frac{TL\eta^2}{2}\sigma^2 \leq \mathbb{E}[\tilde{J}^0] - \tilde{J}_{\inf} + \Delta_T + \frac{TL\eta^2}{2}\sigma^2. \tag{60}$$

Dividing by $\tfrac{\eta}{2}T$ and bounding the average from below by the minimum over $t < T$ gives Eq. (58) with $C = 2$. □

**Corollary D.8** (Step-Size Schedule)**.** *With* $\eta = \Theta(1/\sqrt{T})$, *the third term of Eq. (58) is* $\mathcal{O}(1/\sqrt{T})$ *and the first is* $\mathcal{O}\big((1 + \Delta_T)/\sqrt{T}\big)$. *Hence: (i) if the drift is summable,* $\Delta_T = \mathcal{O}(1)$, *the rate is* $\mathcal{O}(1/\sqrt{T})$; *(ii) if* $\Delta_T = o(\sqrt{T})$, *the bound is* $o(1)$ *(asymptotic surrogate stationarity); (iii) if* $\Delta_T = \Theta(\sqrt{T})$, *the bound saturates at an* $\mathcal{O}(1)$ *stationarity neighborhood. Regime (i) is the clean* $\mathcal{O}(1/\sqrt{T})$ *statement of Theorem 4.10; the others make explicit that vanishing stationarity requires the prototype drift to be controlled, which we recommend stating alongside the rate.*

The selection filter is thus a *safe-exploration* mechanism: by Lemma D.6 it never degrades the worst-case Riemannian-SGD bound, while the NES/PSO candidates enable escape from disconnected basins.

### D.5. Lifting to the Exact Potential: Expected Dissipation of $V$

We now lift the surrogate descent guarantees to the global exact potential $V$. The appendix C provides the per-round, per-client structural half via the majorization residual $\rho_k^t$ (Proposition C.10). Here we incorporate three additional real-world factors: partial participation ($\mathcal{S}^t$), prototype staleness across local steps (bounded by $\Delta_T$), and stochastic gradient noise. The result is an expected dissipation inequality for $V$, which under appropriate step-size and drift conditions yields the $\mathcal{O}(1/\sqrt{T})$ stationarity rate of Theorem 4.10.

Let $g_k^t(\theta) \triangleq w_k F_k(\theta; \mathcal{I}_k) + \Phi_k^t(\mathcal{T}_k(\theta))$ be the exact per-pair block over the participating neighbors $\mathcal{N}_k^t$. At the round-$t$ initialization these neighbors sit exactly at their uploads $z_j^t$, so $g_k^t$ coincides with the participating part of the $\theta_k$-block of $V$ at $W^t$. Under full participation ($\mathcal{N}_k^t = \mathcal{N}_k$) this is the entire block and $\mathrm{grad}_{g_k}g_k^t(\theta_k^t) = \mathrm{grad}_{g_k,\theta_k}V(W^t)$; under partial participation the omitted edges $\{(k,j) : j \in \mathcal{N}_k \setminus \mathcal{N}_k^t\}$ contribute a further bounded term $\beta_k^t \triangleq \|\mathrm{grad}_{g_k,\theta_k}V(W^t) - \mathrm{grad}_{g_k}g_k^t(\theta_k^t)\|_{g_k}^2$, which we carry explicitly in the residual below ($\beta_k^t = 0$ under full participation, and is $\mathcal{O}$ (fraction of missing neighbor mass) in general).

**Proposition D.9** (Per-Round Dissipation of $V$ up to an $\mathcal{O}(\delta + \Delta)$ Residual)**.** *Under the assumptions of Theorem D.7 and the error bound Lemma D.4, the exact per-pair block obeys, for each* $k \in \mathcal{S}^t$,

$$\mathbb{E}\big[g_k^t(\theta_k^{t+1}) \mid \theta_k^t\big] \leq g_k^t(\theta_k^t) - \frac{\eta}{2}\big\|\mathrm{grad}_{g_k,\theta_k^t}\tilde{J}_k^t\big\|_{g_k}^2 + \frac{L\eta^2}{2}\sigma^2 + 2\delta_k^t. \tag{61}$$

*Consequently, summing the participating blocks and absorbing the inter-round neighbor movement into the drift budget $\Delta_T$, the exact potential dissipates in expectation up to a per-round residual of order $\sum_{k \in \mathcal{S}^t}(\delta_k^t + \beta_k^t) + \Delta_T$; with $\eta = \Theta(1/\sqrt{T})$ and $\Delta_T = \mathcal{O}(1)$,*

$$\min_{t<T} \mathbb{E}\left[\langle \|\text{grad}_{g_k,\theta_k} V(W^t)\|^2_{g_k}\rangle_t\right] = \mathcal{O}\left(\frac{1}{\sqrt{T}}\right) + \mathcal{O}(\bar{\delta} + \bar{\beta}), \qquad \bar{\delta} \triangleq \limsup_t \max_{k \in \mathcal{S}^t} \delta_k^t, \;\; \bar{\beta} \triangleq \limsup_t \max_{k \in \mathcal{S}^t} \beta_k^t, \qquad (62)$$

*i.e. the iterates reach an $\mathcal{O}(\bar{\delta} + \bar{\beta})$ first-order-stationary neighborhood of $V$. The residual vanishes iff the compression is asymptotically exact ($\delta_k^t \to 0$, achieved under homogeneous overlaps $\Omega_k^{t,\min} = \bar{\omega}_k^t$ and vanishing neighbor dispersion $C_k^t \to 0$), participation is asymptotically full ($\beta_k^t \to 0$), and the drift is summable.*

*Proof.* **Per-round block inequality (Eq. (61)).** By the two-sided error bound, $g_k^t \leq \tilde{J}_k^t + \delta_k^t$ and $\tilde{J}_k^t \leq g_k^t + \delta_k^t$ pointwise (Lemma D.4, since $g_k^t - \tilde{J}_k^t = \Phi_k^t - \widetilde{\Phi}_k^t$). Hence, using the filtered descent Eq. (57),

$$\mathbb{E}[g_k^t(\theta_k^{t+1}) \mid \theta_k^t] \leq \mathbb{E}[\tilde{J}_k^t(\theta_k^{t+1}) \mid \theta_k^t] + \delta_k^t$$
$$\leq \tilde{J}_k^t(\theta_k^t) - \frac{\eta}{2}\|\text{grad}_{g_k}\tilde{J}_k^t(\theta_k^t)\|^2_{g_k} + \frac{L\eta^2}{2}\sigma^2 + \delta_k^t$$
$$\leq g_k^t(\theta_k^t) - \frac{\eta}{2}\|\text{grad}_{g_k}\tilde{J}_k^t(\theta_k^t)\|^2_{g_k} + \frac{L\eta^2}{2}\sigma^2 + 2\delta_k^t,$$

which is Eq. (61); the last line uses $\tilde{J}_k^t(\theta_k^t) \leq g_k^t(\theta_k^t) + \delta_k^t$.

**Lift to $V$.** Since $V$ is $L$-smooth (Assumption 4.2), the simultaneous round step makes $\mathbb{E}[V(W^{t+1}) - V(W^t)]$ the sum of participating block changes minus the over-counted within-$\mathcal{S}^t$ edge interactions, the latter absorbed by the drift mechanism $\Delta_T$ via Lipschitzness of $d_{\mathcal{Z}}^2$ (Assumption 4.9). Summing the block inequality over $\mathcal{S}^t$ and over rounds and dividing by $\frac{\eta}{2}T$ as in Theorem D.7 yields Eq. (58) plus the residual $\frac{2}{T}\sum_t \langle \delta_k^t \rangle_t$. Finally, $\text{grad}_{g_k}\tilde{J}_k^t(\theta_k^t) = \text{grad}_{g_k,\theta_k} V(W^t) + e_k^t$, where $e_k^t$ collects the gradient compression gap ($\|e_k^t\|^2_{g_k} \leq \kappa\,\delta_k^t$, $\kappa = \mathcal{O}(L)$, via $\|\nabla h\|^2 \leq 4M(h - \inf h)$ for $M$-smooth $h$) and the participation gap $\beta_k^t$, inflating the residual by $\mathcal{O}(\langle \delta_k^t + \beta_k^t \rangle_t)$. Thus $\eta = \Theta(1/\sqrt{T})$ and $\Delta_T = \mathcal{O}(1)$ give Eq. (62), with the stated vanishing conditions (Lemma D.4, plus asymptotically full participation). $\square$

### D.6. Monotone Surrogate Descent of Evolutionary Proposals

**Proposition D.10** (Evolutionary Proposals Preserve Monotonicity). *Fix round $t$ and fields $\{Z_k^t\}$, and let $\theta_k'$ be any proposal (gradient-based, NES, or PSO). If the client applies the acceptance rule Eq. (17), which retains $\theta_k^{\text{cur}}$ in the pool, then $\tilde{J}_k^t(\theta_k^{t+1}) \leq \tilde{J}_k^t(\theta_k^t)$. Under Lemma D.4, the exact per-pair coupling contribution obeys $g_k^t(\theta_k^{t+1}) \leq g_k^t(\theta_k^t) + 2\delta_k^t$.*

*Proof.* The surrogate inequality is the monotone-acceptance Proposition C.18: the $\arg\min$ over a finite pool containing $\theta_k^{\text{cur}}$ cannot exceed $\tilde{J}_k^t(\theta_k^{\text{cur}})$, independent of how $\theta_k'$ is generated. For the lift, chain the two-sided bound:

$$g_k^t(\theta_k^{t+1}) \leq \tilde{J}_k^t(\theta_k^{t+1}) + \delta_k^t \leq \tilde{J}_k^t(\theta_k^t) + \delta_k^t \leq g_k^t(\theta_k^t) + 2\delta_k^t.$$

$\square$

*Remark* D.11 (Saddle Avoidance in Practice). The stochastic exploratory nature of the NES/PSO proposals, combined with the monotone filter, lets GeoEvo discover descent directions in regions of negative curvature where a pure first-order step stalls. Because acceptance is governed solely by $\tilde{J}_k^t$, this exploration is variance-controlled and never violates the per-round guarantees above.

*Remark* D.12 (Exploration at No Cost to the Rate). Since the unbiased base step $\theta_k^{\text{base}}$ always remains in the candidate pool, accepting any NES/PSO proposal can only further decrease $\tilde{J}_k^t$; the worst-case surrogate-stationarity rate of Theorem D.7 is therefore preserved *exactly*, while a successful exploratory move strictly improves the per-step objective. The sole slack incurred when lifting this guarantee to the exact per-pair objective $g_k^t$ is the additive $2\delta_k^t$, which is controlled by Lemma D.4 and vanishes under homogeneous overlaps and vanishing neighbor dispersion. Hence, evolutionary exploration is *safe by construction*: it can escape disconnected basins yet never degrades the monotone descent or the convergence rate established above.

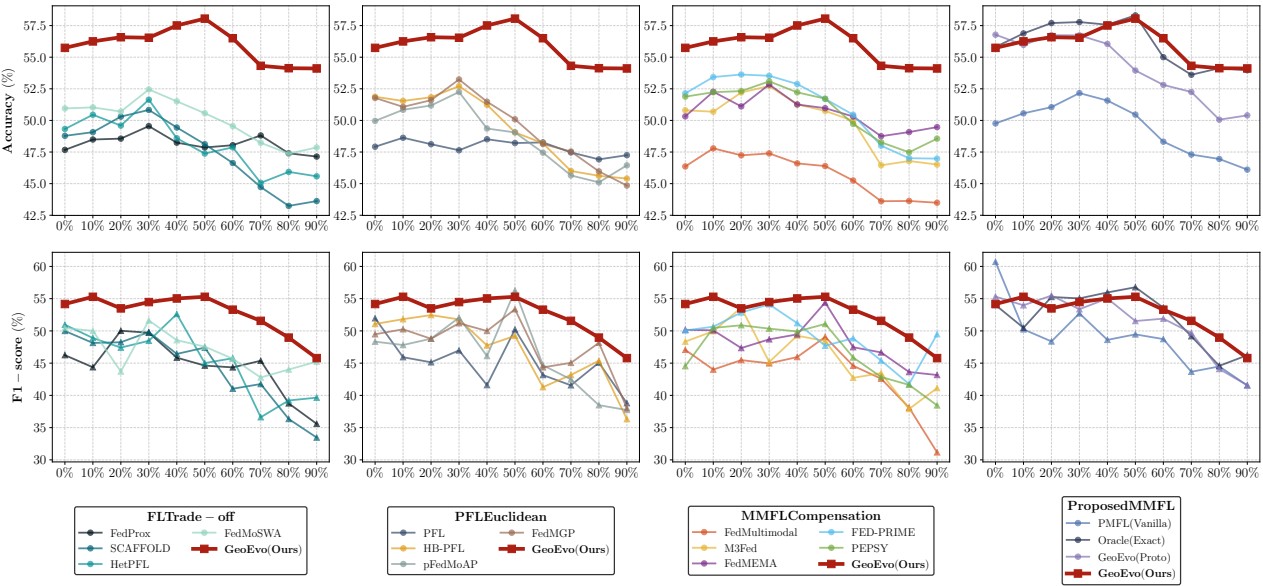

*Figure 3.* Comparative performance of representative methods on UPMC Food-101 dataset under binomial modality simulation.

# E. Experiments

## E.1. Experimental Setting

### E.1.1. DATASETS

We evaluate our method across diverse tasks spanning multiple domains. For emotion recognition (ER), we use CREMA-D (Cao et al., 2014), which contains audio-visual clips from 91 actors performing 12 sentences in six emotions (neutral, happy, anger, disgust, fear, sad). For social media (SM) crisis classification, we use CrisisMMD (Alam et al., 2018), a dataset of tweets with paired image and text data from seven major natural disasters, annotated for disaster impact such as property damage, injuries, and fatalities. For large-scale classification, we use UPMC Food-101 (Wang et al., 2015), a noisy bimodal dataset of image-text pairs from 101 food categories, serving as a benchmark for robust alignment under high-dimensional modality incompleteness.

### E.1.2. EVALUATION METRICS

To ensure a rigorous and context-sensitive assessment, we adopt a tailored evaluation strategy where the choice of metrics aligns with the specific characteristics of each multimodal learning task. For the emotion recognition task, where class distribution is often imbalanced, we employ Accuracy (ACC), Unweighted Average Recall (UAR), and the F1 score. These metrics collectively quantify overall correctness, per-class sensitivity, and the harmonic balance between precision and recall, respectively. Social media crisis classification and food category recognition (UPMC Food-101) are evaluated using Accuracy and the F1 score, providing a standard measure of classification effectiveness in these contexts.

### E.1.3. PARTITION AND SIMULATION

To simulate realistic federated conditions, we construct non-IID client partitions and modality-missing patterns tailored to each dataset. Speaker-dependent emotion data (CREMA-D) is partitioned by speaker ID, capturing natural individual variations. For CrisisMMD and UPMC Food-101, we generate synthetic non-IID splits using a Dirichlet distribution $\mathrm{Dir}(\alpha)$ with $\alpha \in \{0.05, 0.1, 0.5, 1.0\}$ to control heterogeneity levels, where smaller values of $\alpha$ induce more severe distribution skew while larger values yield more uniform distributions.

Two modality-missing patterns are implemented: (i) Proportional simulation preserves the original dataset-specific missing rates, reflecting natural availability biases; and (ii) Binomial simulation applies fixed Bernoulli probabilities per modality, generating uniformly random missingness across all samples. These complementary designs enable evaluation under both realistic and worst-case missing-modality scenarios.

### E.1.4. IMPLEMENTATION DETAILS

Following FedMultimodal (Feng et al., 2023), we employ efficient pre-trained backbone models for feature extraction: MobileNetV2 (Howard et al., 2017) for vision, MobileBERT (Sun et al., 2020) for text, and MFCCs (Davis & Mermelstein, 1980) for audio. For the large-scale UPMC Food-101, we adopt the ViLT (Kim et al., 2021) encoder to handle high-dimensional alignment. Our GeoEvo employs a lightweight multimodal architecture consisting of: (i) modality-specific encoders (RNN-only for video and text, Conv-RNN for others); (ii) an attention-based fusion module; and (iii) a two-layer dense classifier. The fusion module is implemented as a single-layer structured self-attention mechanism. For CREMA-D, we use 4 attention heads with a hidden dimension of 64 and 128, respectively. For CrisisMMD, we employ a fused-attention variant that operates on concatenated features, configured with 6 heads and a hidden dimension of 64 to capture structured cross-modal interactions better. All experiments share the following fixed hyperparameters: batch size of 16, dropout rate of 0.2, ReLU activation, and the AdamW optimizer with linear learning rate decay. These consistent settings ensure fair and reproducible comparisons across all evaluations.

## E.2. Comparative Experimental Analysis

Table 1, Figure 2 and 3 present a comprehensive evaluation of GeoEvo against state-of-the-art baselines under heterogeneous modality conditions. The results unequivocally validate our premise: treating modality incompleteness as a structural identity via geometry-aware optimization yields superior robustness compared to traditional strategies.

**Limitations of Euclidean-Based Aggregation.** Standard FL approaches, such as FedProx and SCAFFOLD, exhibit significant performance degradation as modality heterogeneity intensifies. Specifically, at a 70% missing rate, SCAFFOLD's accuracy plummets to 47.14%, accompanied by high variance. This decline underscores the fundamental limitation of enforcing a single global objective across clients with incongruent modality combinations. Similarly, while Euclidean-constrained PFL methods like HB-PFL and FedMGP attempt to mitigate this by personalizing parameters, they fail to account for the geometric curvature induced by missing modalities. Consequently, these methods suffer from "Euclidean rigidity," where the optimization trajectory is forcibly averaged in a flat space rather than adapting to the intrinsic Riemannian submanifolds of the local data. This is evidenced by the sharp drop in FedMGP's performance from 51.60% (30% missing) to 47.10% (70% missing), highlighting their inability to preserve structural integrity under severe data sparsity.

**The Ceiling of Compensation-Oriented MMFL.** In the realm of multimodal FL, methods that rely on imputation or forced alignment, such as FedMultimodal and $M^3$Fed, face a distinct performance ceiling. FedMultimodal, for instance, achieves only 44.82% accuracy at the 70% missing rate. This suboptimal adaptation suggests that compensatory mechanisms—such as zero-padding or latent hallucination—often introduce noise that distorts the joint representation rather than recovering true semantics. Unlike these approaches, which strive to repair the data, GeoEvo adopts an *Identity-Aware* paradigm. GeoEvo naturally nullifies updates along unobserved dimensions, allowing the optimization to remain grounded in observed reality. This shift from artificial compensation to geometry-aware adaptation explains why GeoEvo maintains a decisive lead over FED-PRIME and PEPSY, particularly in high-missing regimes where minimizing semantic distortion is critical.

**Synergy of Evolutionary Dynamics and Robustness.** The proposed GeoEvo demonstrates remarkable resilience, establishing a new state-of-the-art with an accuracy of **54.67%** under the extreme 70% missing scenario—outperforming the strongest baseline by a margin of 5.32%. This stability is attributed to the synergistic co-design of our algorithm: NES drive curvature-adaptive local exploration, while structure-guided PSO facilitates knowledge transfer across shared modalities without direct data exchange. The integration of these components within a potential game formulation enforces monotone surrogate descent that drives the global potential into a controlled dissipation band, guiding the system toward a geometry-aware Nash equilibrium. Notably, even as environmental uncertainty escalates, GeoEvo's performance variance remains minimal compared to the erratic fluctuations observed in baselines like FedMGP ($\pm$ 4.48), confirming the stabilizing effect of our closed-loop policy optimization.

**Component Efficacy and Oracle Approximation.** The ablation study further elucidates the necessity of our dual-force mechanism. The vanilla PMFL, lacking geometric adaptation, lags significantly behind the full model. The introduction of NES in *GeoEvo (Prototype)* yields substantial gains by navigating the local manifold curvature, yet it is the integration of PSO in *GeoEvo (Ours)* that unlocks the full potential of cross-client knowledge transfer, pushing accuracy from 52.05% to 54.67% in the 70% missing setting. Most strikingly, GeoEvo (Ours) achieves performance parity with the *Oracle (Exact)* baseline—and the exact-potential reference solver. This proximity indicates that our geometry-aware approach successfully recovers nearly all theoretically accessible information from incomplete data, effectively bridging the gap between rigid aggregation and geometry-aware personalization.

