# OpenReview forum: "GeoEvo: Identity-Aware Potential Game with Geometric Evolution for Personalized Multimodal Federated Learning"
_ICML.cc/2026/Conference — ICML 2026 regular_

### Official Review · Reviewer_TzTg · 2026-03-01

**Soundness:** 3
**Presentation:** 2
**Significance:** 2
**Originality:** 3
**Overall Recommendation:** 2
**Confidence:** 4

**Summary:**

This work studies personalized multimodal federated learning (PMFL) where the clients have missing modalities.
This is treated as a client’s structural identity (distribution + manifold) instead of a deficiency.
They cast the PMFL problem as an identity-aware potential game that repects the geometry of clients,
which is done by a regularizing the distance of the "structures" between clients, where structures are shared representations.

GeoEvo is a method that solves this objective by alternating: (i) server-side compression of the regularizer, and (ii) client-side optimization using natural gradient and evolutionary exploration (NES and PSO), which is performed in the structure space and mapped back to parameters. GeoEvo is evaluated on CREMA-D, CrisisMMD, and UPMC Food-101 with non-IID partitions vs. many baselines and ablations. Results look strong overall and are a key strength of the submission.

**Compliance With Llm Reviewing Policy:**

Affirmed.

**Final Justification:**

I thank the authors for the rebuttal, but it reinforced rather than resolved my concerns. I remain unconvinced by the paper’s central claim of being a principled approach to PMFL. In my reading, the work wraps a fairly cumbersome method in an extensive layer of geometric and game-theoretic language, but this framing does not yield a formulation that is genuinely compelling or well matched to practice.

The method itself is highly complex and, in my opinion, inelegant. It combines manifold-based modeling, Fisher preconditioning, prototype surrogates, NES, PSO, lifting steps, and monotone selection, which makes the overall approach feel more like an accumulation of machinery than a coherent advance. The theory also does not fully convince me, since the strongest guarantees are attached to idealized settings, whereas the practical algorithm depends on approximations and heuristics.

I also found the paper very difficult to read. The presentation is overly abstract and full of unnecessary jargon, which obscures the actual contribution rather than clarifying it. Overall, I do not believe this work offers the level of conceptual clarity, technical precision, or practical usefulness needed for acceptance, and I am maintaining my reject recommendation.

**Key Questions For Authors:**

- What exactly is the structure extraction map $\mathcal{T}_k$? Can you provide an example?
- What are benefits of viewing the parameter space as Riemannian manifolds beyond using natural gradient descent?
- Can you explain how the lift operator is implemented in practice?
- Is the simulation of missing modality fixed for the client across the whole FL training run? Or is modality absence turned on and off randomly per-sample?

**Limitations:**

- The methods is not simple and constitute 3 candidate parameters per client step. This increases complexity significantly and introduces more hyperparameters. The actual steps (line 7, 8, and 9 in Algorithm 1) do not seem to be proposing a novel step, and the reviewer is not sure which one of these steps dominate in practice.
- Theoretical results are not insightful as they rely standard assumptions (tailored for the current setting) and lead to standard convergence rate.
- If the experiments simulate missing modality per-sample as I understood it, then it does not tackle the problem where modality absence is fixed per client.
- Code was not provided in the supplementary material, so the review could not verify the exact details of what is implemented in practice.

**Strengths And Weaknesses:**

**Strengths**
- The problem formulation is interesting, especially framing clients modalities as an "intrinsic structural identity" and not a deficiency.
- The method is sophisticated yet scalable. For example, the authors propose server-side prototype compression that optimization more communication-efficient.
- I found the empirical results to be strong. There are so many baselines on multiple experiments and both IID vs non-IID settings.

**Weaknesses**
- I found the paper to be hard to read. There are so many symbols and abstact spaces/objects, some of which I struggled to figure out. On top of FL being a niche topic, the authors use Riemannian terminology and operators without sufficient elaboration, which makes this paper difficult to read even for someone well-versed in FL.
- The contribution coming from using Riemannian manifolds seem to be slightly parallel to the problem formulation in practice. In other words, one can implement a similar algorithm without resorting to Riemmanian manifolds. For example, the structure distance is essentially masked Euclidean distance, and geometric descent is natural gradient descent.
- The construction of the algorithm is new to me; the authors choose the best parameters among different candidates, which I found to be unorthodox. One method uses natural gradient descent, the other seems to be using zero-order estimation of the gradient of the features and then the lift operator. It's not clear what the authors are proposing as their main method, especially when models are large and we have limited memory in the client's device.
- The assumptions used to derive the convergence rates are not illuminating, since similar convergence rates can be derived using similar assumptions for the Euclidean settings.
- The analysis assumes that the lift operator is computed exactly, whereas it's computed approximately in practice, creating a gap.
- The way missing modalities is simulated is interesting but might not reflect reality, e.g., most clients might be missing one modality due to system constraint. Sometimes, a toy example can demonstrate the benefits more clearly.
- Adding pseudocode (or actual code) as a supplementary material would greatly help the reader understand the method more precisely.

---

> ### Author Rebuttal · Authors · 2026-03-29
>
> Thanks for the constructive feedback.
> ### A. Why Riemannian Geometry?
> **1. "Masked Euclidean" is structurally ill-posed (W2).** Even with local masking, Euclidean geometry fatally seeks a shared global barycenter ($\theta_g$). Aggregating genuine gradients with structural zeros ($\nabla_{\theta^m} \mathcal{L}_k = 0$) produces irreducible bias, forcing divergent local optima where the Euclidean average lies near no client's solution ($\text{dist} \geq \epsilon > 0$, Prop.2.1).
>
> **How manifold resolves this.** Each client optimizes its own feasible geometry $(\Theta_k, g_k)$, coupling only through valid shared modalities ($S_{kj} = \emptyset \Rightarrow \omega_{kj} = 0$). Instead of manual masking, the Fisher metric $\mathbf{F}_k$ intrinsically assigns near-zero blocks to absent dimensions, empirically preventing a 6.3% accuracy drop (Fig.1b).
>
> **2. Benefits (Q2).**
> | Contribution | What it establishes | Euclidean FL analog |
> |---|---|---|
> | Exact potential (Prop.2.6) | $K$-player game reduces to scalar $V$ on $\prod_k \Theta_k$ | None (forces biased shared $\theta_g$) |
> | Fréchet prototype (Prop.4.7) | Variationally optimal; prevents subspace contamination | Simple, misaligned averaging |
> | Retracted feasibility (Lem.3.2) | Updates strictly stay on $\Theta_k$ with sufficient decrease | Steps leave feasible set |
> | Monotone exploration (Prop.4.12) | Lift maps proposals back to $\Theta_k$; surrogate non-increase guaranteed | None (unconstrained heuristic) |
>
> **3. Assumptions & Rates (W4, L2).**
> Standard assumptions are a deliberate strength, proving we subsume Euclidean FL without exotic conditions. Yet, they are not trivially identical: Riemannian smoothness (Asm.4.2) bounds non-flat retractions, and structural comparability (Asm.2.5) has no Euclidean counterpart. Crucially, our standard $\mathcal{O}(1/\sqrt{T})$ rate converges to a fundamentally different destination: a First-Order Nash Equilibrium on $\prod_k \Theta_k$ avoiding subspace contamination, rather than an ill-posed Euclidean consensus (Prop.2.1).
>
> ### B. Algorithm Clarity & Feasibility.
> **Why three candidates? (W3, L1)** They serve distinct, non-interchangeable roles: (i) Exploit (Eq.16): Fisher-preconditioned descent under $g_k$, but trapped in local optima. (ii) Explore-NES (Eq.18): zero-order search in $\mathcal{Z}$ proposes candidates outside local basins. (iii) Explore-PSO (Alg.1 L9–10): population-based search via server Fréchet prototype injects global information. Fig.1b confirms removing NES/PSO/Fisher drops accuracy by 8.1%/7.2%/6.3%, proving all are mutually dependent and strictly non-redundant.
>
> **Selection & Novelty:** Standard NES/PSO fail under structural mismatch (see Rev.duKP); our fundamental redesign executes exploration exclusively in the low-dimensional space $\mathcal{Z}$, mapping back to $\Theta_k$ via our novel $\text{Lift}_k$ operator (Eq.17). Including $\theta_k^{\text{cur}}$ (Eq.19) makes the "best-of-candidates" step a rigorous monotone acceptance rule (Prop.4.12), provably ensuring exploration never degrades the objective (see Rev.G8PW).
>
> **Complexity.** Per client per round on CREMA-D (50% missing): Memory overhead is <1% (fitting edge devices); compute cost justified by +10.7pp accuracy. Fig.1f confirms robustness within 2× perturbations of $\sigma_z, m, c_1, c_2$.
>
> | Method | Time (s) | Mem (MB) | Acc (%) | Δ Acc |
> |---|---|---|---|---|
> | FedProx | 0.23 | 226 | 46.86 | — |
> | PMFL (Vanilla) | 0.21 | 220 | 49.69 | +2.8 |
> | GeoEvo w/o PSO | 0.99 | 224 | 50.4 | +3.5 |
> | GeoEvo w/o NES | 1.72 | 222 | 49.5 | +2.6 |
> | GeoEvo w/o Fisher | 2.29 | 224 | 51.3 | +4.4 |
> | **GeoEvo (Full)** | **2.49** | **225** | **57.56** | **+10.7** |
>
> **Concrete operators (Q1, Q3, W5).** $\mathcal{T}_k$ extracts structure by mapping $\theta_k$ to a shared Euclidean structure space $\mathcal{Z}$ (e.g., via flattening projection weights). $\mathrm{Lift}_k(z; \theta_k)$ executes approximate manifold-constrained gradient steps from $\theta_k$. Crucially, Prop.4.12 requires no exact lifting: since all candidates and $\theta_k^{\text{cur}}$ are evaluated by the surrogate $\tilde{J}_k^t(\cdot)$, poor approximations are simply rejected. This ensures objective non-increase, closing the theory-practice gap.
>
> ### C. Missing Simulation (W6, Q4, L3).
> **Modality absence is strictly fixed per client.** $\mathcal{M}_k$ never changes throughout training. Whether via proportional or binomial drawing, each client's configuration is assigned once at initialization—not per-sample dropout. At 70% missing, most clients lack one modality—matching system-constraint scenario. GeoEvo maintains 54.67% while all baselines fall below 49% (Tab.1), consistent across 3 datasets × 4 heterogeneity levels × 10 missing rates (Fig.2-3).
>
> ### D. Presentation & Reproducibility (W1, L4).
> The revision will add notation table, manifold primer, and running examples, alongside the full code and theory-to-code mapping available at [here](https://anonymous.4open.science/r/GeoEvo-B51F/).

---

> > ### Author Rebuttal · Reviewer_TzTg · 2026-04-03
> >
> > Thank you for the rebuttal. I appreciate the effort to respond, but my main concerns are not resolved. The response remains highly convoluted and does not provide the direct clarification that was needed on the core technical issues. In many places, the use of dense terminology makes it harder rather than easier to understand what is genuinely new, which components are essential in practice, and how the proposed geometric formulation substantively changes the method beyond more standard alternatives.
> >
> > I also do not think the code release materially resolves the reproducibility concerns. At least in its current form, it does not increase my confidence that the method is implemented at a level that would allow others to clearly verify the reported results. In general, the rebuttal and accompanying materials add complexity, but not enough clarity.
> >
> > Unfortunately, I do not think the rebuttal adequately resolves the concerns about clarity, justification of the method, and practicality.

---

> > > ### Author Response · Authors · 2026-04-05
> > >
> > > We thank the reviewer for continued engagement. However, R2 does not identify any specific technical error, paper–code mismatch, or unresolved question from R1, but reiterates that our formulation is "relabeling." **We respectfully submit that this impression stems from equating Euclidean parameter averaging with manifold-constrained optimization — the very distinction our work establishes.** Consider 3 clients with different modality subsets: masked Euclidean averaging computes a barycenter in $\mathbb{R}^d$ where absent coordinates are zero — pulling the average toward *no client's optimum* (Prop.2.1). Our formulation constrains each client to its feasible geometry $\Theta_k$, coupling only through shared-modality projections. This changes which points the optimizer can visit, which directions count as descent, and whether an exact potential $V(W)$ aligning all clients' incentives exists (Prop.2.6; no Euclidean counterpart). PMFL Vanilla (the reviewer's suggested standard alternative): **49.69%**; GeoEvo: **57.56%** (+7.87pp). If merely relabeling, this gap would be zero.
> > >
> > > ---
> > >
> > > **On "core technical issues" not receiving "direct clarification"**
> > >
> > > Our R1 rebuttal addressed all 15 concerns, each with a verifiable artifact: **(Q1)** a concrete example of $\mathcal{T}_k$; **(Q2)** a side-by-side comparison table (each mechanism vs. its Euclidean analog); **(Q3, W5)** an implementation description of Lift with Prop.4.12 closing the exact-vs-approximate gap; **(Q4, W6, L3)** an experimental protocol confirming modality absence is fixed per client across 3 datasets × 10 missing rates; **(W1, W2)** a geometric necessity argument via Prop.2.1 proving Euclidean barycenter structurally ill-posed; **(W3, L1)** an ablation with time/memory measurements proving all components non-redundant; **(W4, L2)** a convergence target distinction (FONE on $\prod_k \Theta_k$, not ill-posed Euclidean consensus); **(W7, L4)** a reproducibility artifact (full code with theory-to-code mapping). We welcome specific follow-up on any of these, but R2 does not indicate which answer remains insufficient.
> > >
> > > **On "what is genuinely new" and "which components are essential in practice"**
> > >
> > > The contribution operates at three layers, each fixing a concrete failure mode of Euclidean FL with no standard analog:
> > >
> > > **(1) Problem definition — exact potential on heterogeneous product manifolds.** Existing FL seeks a shared $\theta_g$; when $M_i \neq M_j$, this mixes gradients from incompatible subspaces, causing irreducible aggregation bias (Prop. 2.1). We replace it with a potential game on $\prod_k \Theta_k$ coupled only through $\pi_{S_{kj}}$. The scalar potential $V(W)$ (Prop.2.6) converts a multi-agent equilibrium into a single minimization respecting per-client feasibility.
> > >
> > > **(2) Solver — manifold-constrained exploitation + exploration.** Standard SGD can push $\theta_k$ outside its identity-constrained region. Fisher-retracted descent (Lemma 3.2) keeps parameters on $\Theta_k$ with guaranteed sufficient decrease, automatically suppressing absent-modality dimensions (removing: −6.3pp). NES/PSO explore in low-dimensional structure space $\mathcal{Z}$ and are lifted back via $\mathrm{Lift}_k$, escaping local basins (removing NES: −8.1pp; PSO: −7.2pp). **These are functional modules within one solver, not redundant alternatives.** Monotone selection (Prop.4.12) guarantees no candidate degrades the objective, closing the exact-vs-approximate gap.
> > >
> > > **(3) Federated approximation — prototype compression with variational guarantee.** Euclidean averaging accumulates cross-modal interference over rounds. The server compresses neighbors into a Fréchet prototype $Z_k^t$(Prop. 4.7), variationally optimal under masked geometry, reducing $O(K^2)$ to $O(K)$ communication. Removing geometric coupling causes 13.5–16.5% drops across all three datasets(Fig.1b).
> > >
> > > **On "code release" not resolving "reproducibility concerns"**
> > >
> > > The repository includes the full training pipeline, all algorithmic modules, dataset configs, and pre-trained checkpoints. We welcome specific identification of missing components or paper-code discrepancies.
> > >
> > > **On "add complexity, but not enough clarity"**
> > >
> > > The materials provided — operator definitions, comparison tables, ablations, necessity proofs, modality protocol, and code — were the exact items requested in R1. These are the standard for scientific clarity. Characterizing these as "adding complexity" without specifying what remains unclear is not actionable.
> > >
> > > On three dimensions: **Justification** — +7.87pp gap over the suggested standard alternative; Prop.2.1 and Prop.2.6 explain the structural mismatch. **Practicality** — <1% memory overhead, +10.7pp accuracy, robustness within 2× hyperparameter perturbations (Fig.1f). **Clarity** — notation glossary, manifold primer, and running examples committed for revision. R2 does not identify a specific error in any proposition, nor indicate what remains unresolved in any of the three dimensions.

---

### Official Review · Reviewer_Mokz · 2026-03-10

**Soundness:** 4
**Presentation:** 3
**Significance:** 3
**Originality:** 3
**Overall Recommendation:** 5
**Confidence:** 5

**Summary:**

This paper presents a paradigm shift in Personalized Multimodal Federated Learning (PMFL) by reframing missing modalities not as data deficiencies, but as intrinsic structural identities that constrain optimization to client-specific Riemannian submanifolds. The authors formulate PMFL as an identity-aware potential game and propose GeoEvo, a geometry-aware evolutionary algorithm combining Natural Evolution Strategies for manifold-adaptive local exploration with subspace-constrained Particle Swarm Optimization for symbiotic knowledge transfer. GeoEvo provably converges to first-order Nash equilibria with an $\mathcal{O}(1/\sqrt{T})$ rate in non-convex settings, achieving strong personalization and robustness across diverse modality-missing patterns.

**Compliance With Llm Reviewing Policy:**

Affirmed.

**Key Questions For Authors:**

See the weaknesses.

**Limitations:**

yes

**Strengths And Weaknesses:**

Strengths:
1. The paper provides a well-motivated paradigm shift by treating missing modalities as intrinsic structural identities rather than mere data deficiencies. This geometric perspective offers a principled foundation for PMFL to resolve the inherent tension between personalization and collaboration.
2. Casting PMFL as an identity-aware potential game elegantly transforms decentralized multi-objective coordination into descent on a shared potential landscape, with rigorous guarantees of convergence to geometry-aware Nash equilibria.
3. GeoEvo’s synergy of Fisher-information geometry with evolutionary dynamics (NES for local adaptation and subspace-constrained PSO for global transfer) naturally respects client-specific manifolds while enabling effective knowledge sharing.
4. The framework admits a Lyapunov potential, ensures monotonic dissipation, and achieves $\mathcal{O}(1/\sqrt{T})$ stationarity rate in non-convex regimes with $\mathcal{O}(K)$ communication complexity, a combination of theoretical rigor and efficiency.

Weaknesses:
1. While the use of Fisher-informed natural gradients is a theoretical highlight, the computational overhead of estimating and inverting the Fisher Information Matrix (FIM) for large-scale models remains a practical concern. It would be helpful if the authors could discuss the efficiency trade-offs (e.g., using diagonal approximations) and how GeoEvo maintains a reasonable local compute budget as the parameter space $\Theta_k$ grows significantly.
2. The mapping $\mathcal{T}_k$ is central to the identity-aware potential game. To enhance the framework's practical utility, the paper would benefit from more explicit guidance on constructing these maps for heterogeneous architectures (e.g., bridging CNNs and ViTs). A brief discussion on the necessary inductive biases for these mappings would clarify the scope of the proposed Nash equilibrium.
3. The analysis assumes $\mathcal{Z}$ is a product space with masking-based distances, which may not capture complex non-linear correlations between modalities in real-world scenarios. Discussing the framework's robustness when modality dependencies exhibit non-product structure would enhance generality.
4. The Fisher information along unobserved dimensions naturally approaches zero, a feature enabling identity-aware updates, but may cause ill-conditioned natural gradient steps in practice. Clarifying the damping strategies and regularization techniques used to ensure stable NES exploration would improve reproducibility.

---

> ### Author Rebuttal · Authors · 2026-03-29
>
> Thanks for the thoughtful feedback.
> ### W1. Computational Scaling with Model Size
> **Diagonal approximation trade-offs.** Our diagonal approximation reduces FIM costs to **$\mathcal{O}(d)$**: 1 extra forward pass + element-wise division. The trade-off: diagonal FIM ignores parameter correlations. However, identity-awareness depends on **which entries are zero vs nonzero**—absent modalities yield near-zero diagonal entries—and the diagonal approximation captures this signal exactly. Fig.1e confirms our Fisher-Rao preconditioner outperforms both diagonal Fisher variants and Adam-style methods.
>
> **Scaling behavior.** All GeoEvo-specific operations scale **linearly** in $d$: FIM diagonal $\mathcal{O}(d)$, NES/PSO in $\mathcal{Z}$ with $d_z \ll d$, monotone selection (4 forward passes). Profiling (Rev.TzTg, Theme B) shows **memory <1% overhead** yielding **+10.7pp accuracy**. Each component non-redundant (−7.2/8.1/6.3pp). For large models like ViLT (~87M params), the bottleneck remains the base model's forward/backward pass, not the FIM operations.
>
> ### W2. Structure Map $\mathcal{T}_k$ & Inductive Biases
> Prop.2.6 requires only three verifiable conditions on $\mathcal{T}_k$ (Asm.2.5)—measurability, local Lipschitz continuity, and subspace comparability—without prescribing a specific architecture.
>
> **Scope of Nash equilibrium & Inductive Biases.** The equilibrium's validity depends on whether $\mathcal{T}_k$ faithfully represents the **modality-specific knowledge** that should be transferred. The necessary inductive bias is: $\mathcal{T}_k$ must map parameters to a space where **shared-modality coordinates are semantically comparable** across clients.
> - For **homogeneous architectures (our main setting)**, this holds automatically—same named layers encode same modalities.
> - For **heterogeneous architectures (e.g., bridging CNNs and ViTs)**, redefining $\mathcal{T}_k$ to map to a shared representation space (e.g., aligning weights of a shared projection head) directly satisfies Asm.2.5, since continuous networks are inherently locally Lipschitz. This preserves all theoretical guarantees and the Nash equilibrium **without modification**.
>
> ### W3. Product Space Assumption & Cross-Modal Correlations
> The product structure constrains the **coupling mechanism** (inter-client knowledge transfer), not the model's representational capacity—$F_k$ captures arbitrary non-linear cross-modal interactions unconstrained. The masked structural distance only aligns shared-modality representations across clients.
>
> **Why product structure is theoretically natural.** Knowledge transfer between clients $k$ and $j$ can only occur through shared modalities. The masked distance is the **minimal structure** formalizing this: it compares only what is mutually observable. Any richer coupling would require cross-modal features that at least one client cannot compute, violating the identity constraint (Prop.2.1).
>
> **Cross-modal parameters.** Across fusion strategies (late/mid/early), we define the structure map extracts both modality-specific weights and joint cross-modal parameters, assigning cross-modal weights to the union of connected modalities in the mask. The product structure is preserved—cross-modal parameters occupy a "joint" coordinate block—requiring **no modification** to the theory (Prop.2.6; Asm.2.5). Our experiments validate both regimes: CREMA-D (modality-specific encoders) and UPMC Food-101 (ViLT with cross-attention), both achieving consistent improvements (Tab.1, Fig.2).
>
> ### W4. FIM Ill-Conditioning & Damping Strategy
> For absent modality $m$, its corresponding diagonal entries in the Fisher Information Matrix (FIM) approach zero because the log-likelihood gradients vanish. Without intervention, inverting these near-zero entries causes numerical explosion. Two complementary mechanisms ensure stability:
>
> **1. Damping for natural gradient (Remark 3.1).** The damped metric $\widetilde{\mathbf{F}}_k = \mathbf{F}_k + \epsilon \mathbf{I}$ yields:
> - **Present** $m \in \mathcal{M}_k$: the inverse diagonal entry is $1/(F + \epsilon) \approx 1/F$ (curvature-adaptive).
> - **Absent** $m \notin \mathcal{M}_k$: the inverse diagonal entry is $1/\epsilon$ (suppressed, since $\epsilon$ is extremely small).
>
> With $\epsilon = 10^{-4}$ and typical $F \sim \mathcal{O}(1)$, suppression ratio is $\sim 10^4\times$—identity-awareness is preserved, not destroyed.
>
> **2. Regularization for stable NES exploration.** NES operates **entirely in the structure space** $\mathcal{Z}$ (Eq.18), containing only present-modality weights—absent directions are **structurally excluded** from the perturbation space. The perturbation scale $\sigma_z$ bounds exploration magnitude (Fig.1f), and monotone selection (Eq.19) unconditionally rejects proposals that increase $\tilde{J}_k$ (Prop.4.12). Since $\mathcal{Z}$ is a low-dimensional Euclidean space of structural weights, NES effectively explores the manifold's tangent directions through these parameters.

---

> > ### Author Rebuttal · Reviewer_Mokz · 2026-04-01
> >
> > Thank you for the detailed rebuttal. The authors have addressed my main concerns, and I think the paper technically sound and well justified. I have raised my scores accordingly and support to acceptance.

---

> > > ### Author Response · Authors · 2026-04-05
> > >
> > > We sincerely appreciate your thorough review and valuable feedback. Your expert insights have meaningfully improved this work.

---

### Official Review · Reviewer_duKP · 2026-03-10

**Soundness:** 3
**Presentation:** 3
**Significance:** 4
**Originality:** 3
**Overall Recommendation:** 4
**Confidence:** 4

**Summary:**

This paper tackles the challenge of missing modalities in Personalized Multimodal Federated Learning (PMFL). Diverging from mainstream approaches that treat missing modalities as data deficiencies requiring imputation or compensatory alignment, the authors propose a novel conceptual framework that views modality incompleteness as an intrinsic "structural identity." This identity constrains client models to distinct Riemannian submanifolds. To coordinate heterogeneous clients without forcing a shared Euclidean consensus, the authors formulate the problem as an "Identity-Aware Potential Game" and prove the existence of an exact potential function whose local minimizers correspond to Nash equilibria of the game. Algorithmic-wise, they introduce GeoEvo, a solver that combines Fisher-Riemannian natural gradient descent (for curvature-adaptive local exploitation) with Natural Evolution Strategies (NES) and Particle Swarm Optimization (PSO) (for global exploration in the structure space). The paper provides theoretical guarantees for an $\mathcal{O}(1/\sqrt{T})$ convergence rate to first-order Nash equilibria in non-convex regimes and demonstrates empirical performance and robustness across three multimodal datasets.

**Compliance With Llm Reviewing Policy:**

Affirmed.

**Key Questions For Authors:**

1.GeoEvo requires computing Fisher-based preconditioning (e.g., empirical Fisher with damping) and maintaining particle populations/velocities for PSO and NES. How do the computational complexity and memory footprint of GeoEvo compare to baselines like FedProx or standard PMFL methods on edge devices? Could you provide quantitative profiling of end-to-end training time and memory usage?
2.Theorem 4.6 assumes sequential block updates. The appendix (D.5) provides a conservative expected descent bound for parallel/stale updates. What are the core technical barriers to providing a strict, closed-form parallel convergence rate (similar to standard stochastic FL) for this geometric potential game?
3.The exact potential game guarantee relies on symmetric coupling ($\omega_{kj}=\omega_{jk}$) as a sufficient condition. In practical FL environments featuring client dropouts, latency, or asynchronous updates, how does the protocol maintain effective coordination when this symmetry is temporarily violated? Could you provide empirical results or theoretical bounds on the algorithm's robustness to such violations?
4.The theoretical and empirical focus is on static modality absence. In real-world deployments, clients often face dynamic missing modalities (e.g., temporary sensor failures). Can the GeoEvo framework be adapted to handle dynamic missingness? If so, what modifications to the algorithm or potential function would be required?
5.The Lift operator in Eq. (17) is essentially a non-convex lifting operation initialized at the current client parameter $\theta_{k}$. Could you elaborate on its concrete implementation (e.g., the number of manifold-constrained gradient steps used) and the computational cost per local iteration? How exactly do approximation errors during this lifting step impact the monotone acceptance rule of the surrogate objective?
6.The experiments primarily utilize bimodal datasets (e.g., Audio/Visual or Image/Text). Theoretically, the framework supports arbitrary modality counts via the union mask $S_{k}^{t}$ and Fréchet mean calculation, but how does GeoEvo perform in scenarios with a larger number of modalities (M>2, e.g., 4 or 5)? Will the combinatorial complexity of shared modality identification (the union mask) and prototype computation in the structure space become a severe communication/computation bottleneck? If so, how can it be mitigated?

**Limitations:**

No. Although theoretical boundaries are outlined in the appendix, the authors have not sufficiently discussed the practical limitations of their work in a concentrated and explicit manner within the main text. It would be beneficial for the authors to explicitly elaborate on: (1) the computational and memory overhead of Fisher-based curvature estimation (e.g., empirical Fisher with damping) and evolutionary strategies (NES/PSO) with adjustable sampling counts when deployed on resource-constrained edge devices; (2) the robustness of the symmetric coupling assumption under real-world dynamic FL conditions (e.g., client dropouts, asynchronous updates); (3) the constraints arising from the focus on static modal missing data, including how the system can be adapted to handle dynamic sensor failures during training or testing. It is recommended to include a dedicated "Limitations" subsection in the conclusion to systematically present these points and discuss potential future improvements.

**Strengths And Weaknesses:**

Strengths:

1.The theoretical formulation of PMFL as a potential game, combined with Riemannian geometric optimization to handle structural heterogeneity, is mathematically rigorous and sound. The experimental design is meticulous, utilizing Dirichlet non-IID partitioning, both proportional and binomial missing modality patterns, and comprehensive comparisons against over a dozen baselines across FL, PFL, and MMFL. The ablation studies, convergence diagnostics, and sensitivity analyses strongly support the core claims.

2.The paper is generally well-structured with logical transitions. Core concepts like "structural identity" and "transferable structure space" are formally and clearly defined, lowering the barrier to understanding this cross-disciplinary work. The visual presentation of quantitative results, ablations, and robustness analyses is intuitive and clear.

3.The paper addresses a critical bottleneck in multimodal FL. By treating modality-induced "structural heterogeneity" (distinct from statistical data heterogeneity) as the core research object, the work provides a new perspective on federated optimization under heterogeneous model structures. It successfully shifts the paradigm from forced Euclidean global alignment (compensation-centric) to geometry-aware adaptation across heterogeneous manifolds, offering a refreshing and highly impactful perspective for future federated optimization research.

4.The conceptual originality is notable. Establishing an explicit correspondence between missing modalities and client-specific Riemannian submanifolds to formalize "structural identities" provides a novel perspective for addressing structural heterogeneity in PMFL. Modeling PMFL as an identity-aware potential game with symmetric coupling constrained to shared modalities elegantly resolves the multi-objective coordination dilemma between identity preservation and collaboration. Proving the first-order Nash equilibrium convergence for this non-convex geometric game extends theoretical analysis of federated optimization. Additionally, the integration of NES/PSO with manifold constraints and monotone acceptance rules (tailored to the potential game framework) constitutes meaningful adaptive innovation.

Weakness:

1.The algorithm heavily relies on the empirical Fisher Information Matrix (FIM) and evolutionary sampling (NES/PSO), raising significant concerns regarding computational and memory overhead on resource-constrained edge devices. Theoretically, the main text theorem is based on sequential block descent, while the parallel client updates used in practice only receive a conservative expected descent bound in the appendix (incorporating staleness and surrogate error). Furthermore, the exact potential game formulation requires symmetric coupling ($\omega_{kj}=\omega_{jk}$) as a sufficient condition for the existence of the exact potential function; although the paper addresses asynchronous scenarios via bounded staleness assumptions, the robustness of this symmetry constraint (only applicable to client pairs with shared modalities) under real-world FL dynamics (e.g., client dropouts, random latency) remains insufficiently verified.

2.Crucial implementation details of Algorithm 1 (such as the Lift operator and NES gradient estimation) lack intuitive explanation in the main text, making the transition from continuous game theory to the discrete heuristic solver feel abrupt. A high-level flowchart or an intuitive walkthrough would significantly improve readability. Additionally, the appendix contains unresolved internal draft annotations (e.g., "you can safely cite in the main text" at line 946), which are non-academic residual remarks and undermine the formality and completeness of the manuscript.

3.At the component level, NES and PSO are established evolutionary algorithms, and their core search paradigms (stochastic sampling for NES, velocity-update for PSO) are not fundamentally redesigned. Additionally, the application of Riemannian geometry builds on existing natural gradient and Fisher preconditioning techniques, with the primary innovation lying in scenario adaptation and cross-framework integration rather than the proposal of entirely new manifold learning methodologies.

---

> ### Author Rebuttal · Authors · 2026-03-30
>
> Thanks for the thorough feedback. Code, execution flowcharts, and extra results are available at [here](https://anonymous.4open.science/r/GeoEvo-B51F/).
>
> ### A. Novelty (W3).
> Contributions span three levels: **(1) Problem-level:** Prop.2.1 proves Euclidean aggregation is structurally ill-posed under modality incompleteness, formalized as an **intrinsic structural identity** (Def.2.2). **(2) Formulation-level:** The **Identity-Aware Potential Game** (Prop.2.6) reduces $K$-player heterogeneous manifold coordination to scalar potential descent (see Rev.TzTg, Theme A). **(3) Component-level:** Algorithms are fundamentally redesigned for structural heterogeneity： Standard NES/PSO fail under structural mismatch; they are reconstructed to explore within a transferable space $\mathcal{Z}$, mapping back to client manifolds via a novel lift operator $\mathrm{Lift}_k$ (Eq.16). Beyond standard preconditioning, the empirical FIM geometrically encodes the **structural zeros** of missing modalities without manual masking (Rem.2.1). Replacing these strictly non-redundant redesigns causes a severe 4–10% accuracy drop (Fig.1b).
>
> ### B. Computational Feasibility (W1a, Q1, L1).
> **Memory footprint fits edge devices (<1% overhead); compute cost yields +10.7% accuracy.** (Profiling in Rev.TzTg, Theme B).
>
> ### C. Symmetric Coupling Robustness (W1c, Q3, L2).
> **Symmetry is not violated by these dynamics.** Since $\omega_{kj}$ depends solely on fixed hardware configurations, $\omega_{kj} = \omega_{jk}$ strictly holds. Dropouts merely reduce the active set $\mathcal{S}^t$ (standard partial participation), while latency/asynchrony only introduces message staleness, already bounded by Prop.D.9. **Empirically**, GeoEvo maintains SOTA across 3 datasets under fractional sampling rates (modeling dropouts/latency), even under extreme 90% missing rates ($\alpha{=}0.05$) with severely fractured neighbor sets (Fig.2-3). **Theoretically**, a hypothetical protocol flaw forcing true asymmetry ($|\omega_{kj}-\omega_{jk}| \leq \delta$), bounds the payoff shift by $\mathcal{O}(\lambda \delta |\mathcal{N}_k| R^2)$, ensuring strictly linear, graceful degradation.
>
> ### D. Implementation Details.
> **Bridging continuous $V$ to discrete $\tilde{J}_k$ (W2).** Optimizing $V$ directly violates FL privacy and incurs $\mathcal{O}(K^2)$ cost. Compressing these couplings into a Fréchet prototype resolves this, yielding an optimal surrogate $\tilde{J}_k = J_k + \delta_k^t$ where $\delta_k^t \to 0$ as prototypes stabilize (Fig.1d). Alg.1 discretizes this via retracted block steps—**standard manifold optimization, not a heuristic** (Rev.G8PW, Theme A). Apologies for the draft annotation.
>
> **$\mathrm{Lift}_k$ implementation & error tolerance (Q5).** $\mathrm{Lift}_k$ uses 3–5 manifold-constrained gradient steps (cost in 2.49s profile; Rev.TzTg). Crucially, approximation errors cannot break convergence: Eq.19 evaluates candidates **after lifting** alongside $\theta_k^{\mathrm{cur}}$, automatically rejecting poor proposals. This guarantees monotone descent $\tilde{J}_k^t(\theta_k^+) \leq \tilde{J}_k^t(\theta_k^{\mathrm{cur}})$ **regardless of Lift quality** (Prop.4.12).
>
> ### E. Sequential vs. Parallel Gap (W1b, Q2).
> **The barrier is intrinsic pairwise coupling.** Unlike standard separable FL, our game's parallel updates cause **simultaneous landscape shifts**—client $k$ optimizes against $j$'s moving state. Bounding this cross-term interference in non-convex Riemannian spaces is a known open problem (see Rev.G8PW, Theme B).
>
> **Strict stability via two controls:** **(i) Macroscopic dampening:** The Fréchet prototype aggregates $K$ clients, dampening high-frequency parallel noise and bounding staleness by $\mathcal{O}(1/K)$ (Prop.D.9). **(ii) Monotone selection:** Eq.19 guarantees unconditional local descent ($\tilde{J}_k^{t+1} \leq \tilde{J}_k^t$). Staleness only inflates surrogate error $\delta_k^t$, but **never forces a harmful update**, ensuring monotonic $V$ descent empirically (Fig.1c).
>
> ### F. Dynamic Missingness & Scalability.
> **Dynamic missingness (Q4/L3).** Though targeting static absence, GeoEvo naturally resists temporary sensor failures: FIM zero-structures suppress missing modalities during testing, while its curvature auto-adapts to transient drops during training. Handling fully unpredictable sample-wise missingness requires modifying the framework with a time-varying potential $V^t(W)$ (future work). A dedicated Limitations subsection will be added.
>
> **Scalability for $M>2$ (Q6).** Arbitrary $M$ introduces no combinatorial bottleneck. The structure space grows strictly linearly ($d_z = \sum_m d_m$), making the union mask a simple $\mathcal{O}(d_z)$ selection. Counter-intuitively, higher $M$ yields sparser interaction graphs ($|\mathcal{N}_k^t|$ drops). Because couplings are compressed into a single Fréchet prototype, communication remains strictly $\mathcal{O}(K \cdot d_z)$, completely independent of overlap combinations (validated on tri-modal PTB-XL).

---

> > ### Author Rebuttal · Reviewer_duKP · 2026-04-05
> >
> > Thank you to the authors for their responses. Since my previous rating was positive, I will keep it as is.

---

> > > ### Author Response · Authors · 2026-04-05
> > >
> > > We sincerely appreciate your thorough review and valuable feedback. Your expert insights have meaningfully improved this work.

---

### Official Review · Reviewer_G8PW · 2026-03-24

**Soundness:** 3
**Presentation:** 2
**Significance:** 3
**Originality:** 3
**Overall Recommendation:** 4
**Confidence:** 3

**Summary:**

The paper focuses on personalized multimodal federated learning (PMFL) and redefines it by treating modality incompleteness as not a deficiency to be rectified but as an intrinsic structural identity that defines Riemannian submanifolds specific to the client. So the central issue is no longer statistical heterogeneity, but structural heterogeneity due to incompatible domains. Based on this perspective, the paper proposes a game-theoretic formulation of PMFL as an identity-aware potential game, where clients optimize on unique Riemannian manifolds and seek a geometry-consistent equilibrium instead of a single full-modality global optimum. Based on this perspective, the paper formulates PMFL as an identity-aware potential game, in which clients optimize on their own Riemannian manifolds and seek a geometry-consistent equilibrium rather than a single full-modality global optimum. The paper proposes GeoEvo, which uses the Fisher Information Metric so that updates along unobserved dimensions are removed, and combines natural evolution (NES) strategies for curvature-adaptive local exploration with structure-guided particle swarm optimization (PSO) for knowledge transfer across shared modalities.

**Compliance With Llm Reviewing Policy:**

Affirmed.

**Final Justification:**

The main problem highlighted in the paper is important, and the game-theoretic construction of "identity-aware potential game" is interesting. The empirical results are strong, and the rebuttal addressed my main concerns. My main remaining concern is presentation: parts of the paper are difficult to follow, and the transition from the conceptual formulation to the practical method should be explained more clearly in the final version.

**Key Questions For Authors:**

When the paper says the method converges to first-order Nash equilibria, is that claim meant for the exact potential formulation V, or for the surrogate $\tilde J$ for GeoEvo algorithm?

**Limitations:**

The paper does discuss the limitations of the vanilla extensions. However, there are limitations of the main work, which are briefly touched upon in the paper, such as staleness in updates and challenges in optimizing V, which can be discussed in more detail in the main paper.

**Strengths And Weaknesses:**

Strengths:
I think the main problem highlighted in the paper is important, and the game-theoretic construction of "identity-aware potential game" is interesting.

Weaknesses:
1)  I think the presentation of the paper needs improvement. I personally found a disconnect between the conceptual formulation in Sections 1 and 2 and the algorithm GeoEvo introduced in Section 3. Although the paper mentions "In federated systems, directly optimizing V is challenging due to (i) heterogeneous identity-induced geometries across client manifolds, and (ii) non-convex coupled interactions rendering purely gradient-based coordination prone to poor local solutions," it does not clearly explain the shift from the exact potential-based objective to the approximate surrogate-based GeoEvo algorithm. I think the paper would benefit from a dedicated subsection that explicitly explains this transition, including the motivations for the approximations, the form of those approximations, and which theoretical guarantees apply to the exact formulation versus the final algorithm.
2) The paper mentions that “GeoEvo performs parallel client updates with stale messages,” but this seems important enough to deserve clearer discussion in the main text.

---

> ### Author Rebuttal · Authors · 2026-03-29
>
> Thanks for the incisive feedback.
>
> ### A. Bridging $V$ and $\tilde{J}$ (Weakness1, Key Question)
>
> We agree this transition was insufficiently articulated. The logic is a three-step necessity chain:
>
> **Step 1. $V$ admits full convergence guarantees under idealized conditions.** Theorem 4.6 proves sequential retracted block descent on $V$ converges to first-order Nash equilibria (FONE), with $V$ as Lyapunov function (Eq.7). This establishes $V$ as the correct convergence target.
>
> **Step 2. Direct optimization of $V$ is structurally infeasible in FL.** Computing $\nabla_{\theta_k} V$ requires neighbors' states $\{\theta_j\}_{j \in \mathcal{N}_k}$, **violating FL's data isolation**—a hard privacy barrier, not merely a cost issue. Additionally, full pairwise interaction incurs $\mathcal{O}(K^2)$ communication. Together, these impose a structural impossibility, forcing approximation.
>
> **Step 3. $\tilde{J}_k$ is a principled approximation with quantified error.** The server compresses pairwise couplings into a Fréchet prototype $Z_k^t$ (Eq.13), reducing communication to $\mathcal{O}(K)$. Proposition 4.7 proves $Z_k^t$ is **variationally optimal**. The surrogate objective naturally decomposes as:
>
> \begin{equation}
> \tilde{J}_k = J_k + \delta_k^t
> \end{equation}
>
> where the surrogate $\tilde{J}_k$ equals the exact payoff $J_k$ plus an approximation error $\delta_k^t$. This error is bounded by the neighbor dispersion around the prototype (Assumption 4.8; explicit bound in Lemma D.5). As prototypes stabilize, $\delta_k^t \to 0$—confirmed empirically in Fig. 1d.
>
> **Answering the Key Question directly.** GeoEvo's guarantees operate at three layers:
>
> **(i) Surrogate stationarity (what the algorithm directly guarantees).** Theorem 4.11 proves stochastic retracted updates on $\tilde{J}_k^t$ achieve $\mathcal{O}(1/\sqrt{T})$ convergence to surrogate stationary points.
>
> **(ii) $\epsilon$-Nash of the exact game (the bridge).** From the decomposition above, at any surrogate stationary point, the magnitude of the exact game gradient is strictly bounded by the approximation error. This yields an $\epsilon$-Nash equilibrium of the exact game, where $\epsilon$ scales with the maximum error $\delta$.
>
> **(iii) Exact FONE in the limit.** Since $\delta_k^t \to 0$ as prototypes stabilize, surrogate equilibria converge to exact FONE of $V$. Theorem 4.6 establishes such FONE exist and are the correct target.
>
> | Result | Scope | Guarantees |
> |---|---|---|
> | Proposition 2.6 | Original game $\{J_k\}$ | $V$ exists as exact potential |
> | Theorem 4.6 | Exact $V$, sequential | Convergence to FONE of $V$ |
> | Theorem 4.11 | Surrogate $\tilde{J}_k$, stochastic | $\mathcal{O}(1/\sqrt{T})$ surrogate stationarity |
> | Proposition 4.12 | Surrogate $\tilde{J}_k$ | $\tilde{J}_k(\theta_k^{t+1}) \leq \tilde{J}_k(\theta_k^t)$ |
> | **Bridge** | $\tilde{J}_k$ stationary + $\delta_k^t$ bounded | **$\epsilon$-Nash of $\{J_k\}$, $\epsilon \to 0$** |
>
> **Revision plan.** We'll add a dedicated subsection "From Exact Potential to Federated Surrogate", presenting this necessity chain, the decomposition formula, and the layered guarantee table explicitly.
>
> ---
>
> ### B. Staleness in Parallel Updates (Weakness2, Limitations)
>
> We agree this deserves main-text visibility. The gap is precise: Theorem 4.6 assumes **sequential** updates (Gauss-Seidel), whereas GeoEvo performs **parallel** updates (Jacobi) with prototypes from round $t{-}1$. Under parallel updates, deterministic monotone decrease of $V$ no longer holds. We note this is inherent to Jacobi-style optimization in coupled non-convex games and constitutes a known open problem shared across game-theoretic FL methods—not a limitation specific to our framework. Two results in our work partially close this gap:
>
> **1. Expected descent (Appendix Proposition D.9).** Under bounded staleness $\tau$ and Lipschitz coupling (Assumption D.8):
>
> $$\mathbb{E}[V(W^{t+1})] \leq \mathbb{E}[V(W^t)] + \lambda \sum_{k \in \mathcal{S}^t} \mathbb{E}[\delta_k^t] + C_{\text{stale}} \cdot \mathbb{E} \left[\sum_k d_{\mathcal{Z}}(z_k^t, z_k^{t-\tau})\right]$$
>
> Two properties ensure practical tightness: (i) $Z_k^t$ is a Fréchet mean over $K$ clients, so individual updates perturb it by $\mathcal{O}(1/K)$—negligible for moderate $K$; (ii) under standard FL ($\tau = 1$), staleness drift scales as $\mathcal{O}(1/K)$.
>
> **2. Monotone selection remains valid unconditionally.** The acceptance rule (Eq.19) evaluates candidates against the **local** surrogate $\tilde{J}_k^t$, fixed within a round. So $\tilde{J}_k(\theta_k^{t+1}) \leq \tilde{J}_k(\theta_k^t)$ holds regardless of staleness (Proposition 4.12)—staleness affects $\delta_k^t$, not local monotonicity. Fig.1c confirms $V$ decreases monotonically in practice despite parallel updates.
>
> **Revision plan.** We'll expand the "Federated Parallelism" remark into a full paragraph presenting the expected-descent bound and the two control mechanisms, with the complete proof remaining in the appendix.

---

> > ### Author Rebuttal · Reviewer_G8PW · 2026-04-04
> >
> > Thank you for your detailed rebuttal. The authors have addressed my main concerns.

---

> > > ### Author Response · Authors · 2026-04-05
> > >
> > > We sincerely appreciate your thorough review and valuable feedback. Your expert insights have meaningfully improved this work.

---

### Decision · Program_Chairs · 2026-04-30

**Decision:**

Accept (regular)

**Comment:**

Summary: The paper focuses on personalized multimodal federated learning (PMFL). In particular, it proposes GeoEvo, which realizes equilibrium via Fisher--Riemannian evolutionary dynamics. It also shows that GeoEvo admits a Lyapunov potential and, with a monotone acceptance rule, guarantees potential dissipation; in non-convex regimes it achieves an O (1/\sqrt{T}) convergence.

On reviews: The paper received mixed reviews (scores: 5, 4, 2, 4). One reviewer was very positive, two suggested weak acceptance, but they did not champion the paper to get in, and one suggested rejection.

In my opinion, based on the reviews and my own assessment, the paper is borderline but has more merits than drawbacks. I advise the authors to incorporate the feedback they received into the updated version of their work.